TECHNICAL REPORT
# Mendelian imputation of parental genotypes improves estimates of direct genetic effects

Alexander I. Young [1,2,3,4 ✉], Seyed Moeen Nehzati[2,3], Stefania Benonisdottir[1], Aysu Okbay[5], Hariharan Jayashankar[6], Chanwook Lee[7], David Cesarini[6,8,9], Daniel J. Benjamin [3,4,6], Patrick Turley [2,10] and Augustine Kong [1 ✉]

Effects estimated by genome-wide association studies (GWASs) include effects of alleles in an individual on that individual (direct genetic effects), indirect genetic effects (for example, effects of alleles in parents on offspring through the environment) and bias from confounding. Within-family genetic variation is random, enabling unbiased estimation of direct genetic effects when parents are genotyped. However, parental genotypes are often missing. We introduce a method that imputes missing parental genotypes and estimates direct genetic effects. Our method, implemented in the software package snipar (single-nucleotide imputation of parents), gives more precise estimates of direct genetic effects than existing approaches. Using 39,614 individuals from the UK Biobank with at least one genotyped sibling/parent, we estimate the correlation between direct genetic effects and effects from standard GWASs for nine phenotypes, including educational attainment ($r = 0.739$, standard error (s.e.) = 0.086) and cognitive ability ($r = 0.490$, s.e. = 0.086). Our results demonstrate substantial confounding bias in standard GWASs for some phenotypes.

GWASs have found thousands of associations between genetic variants and human phenotypes[1] and enabled the prediction of phenotypes using polygenic indices (PGIs)[2]. GWASs typically estimate the linear association between a phenotype and a single-nucleotide polymorphism (SNP) by regression of individuals' phenotypes onto the number of copies of an allele (genotype) that they carry.

Multiple phenomena contribute to the associations found by GWASs[3], referred to here as 'population effects', as they reflect the genotype–phenotype association in the population, including the causal effects of alleles carried by an individual on that individual, called direct genetic effects; effects of alleles in relative(s) through the environment, called indirect genetic effects (IGEs) or genetic nurture[4]; and confounding due to population stratification and assortative mating (AM), which lead to correlations between the SNP and other genetic and environmental factors. Although methods such as principal-component (PC) analysis and linear mixed models (LMMs) adjust for population stratification[5,6], residual confounding often persists in GWAS summary statistics[7,8]. By modeling the effects of many genome-wide SNPs, LMMs can reduce confounding due to AM[9]. However, unless all of the heritability is captured, some confounding due to AM will remain. Depending on the regression design (e.g., PC adjustment, LMM, LMM and PC adjustment), the amount of confounding due to population stratification and AM can differ[6,9].

Decomposing the population effects estimated by GWASs into the different components is important for interpreting and applying GWAS results. For example, IGEs and AM can lead to spurious inference of disease causes in Mendelian randomization[10], and residual stratification in GWASs of height resulted in inflated

signals of selection[7,11]. These problems and others can be remedied with stratification-free estimates of direct genetic effects.

Offspring genotype varies around the expectation given the genotype of the mother and father due to random segregation of genetic material in the parents during meiosis. Thus, analysis of parent–offspring trios can be used to obtain unbiased estimates of direct genetic effects[12–15]. However, parental genotypes are often missing. In the absence of parental genotypes, genetic differences between siblings can be used to estimate direct genetic effects, at the cost of lower power and potential bias due to IGEs from siblings[7,9,16,17]. Here, we show that by treating parental genotypes as missing data and imputing them based on Mendelian laws, we can perform a unified analysis of different data types, including sibling pairs and parent–offspring pairs. Compared to existing approaches[7,9,17,18], our method increases power and identifiability while retaining unbiased estimates of parameters and sampling variance.

Our imputation approach is similar to methods used in animal breeding to impute ungenotyped members of a pedigree[19–21], which are typically designed for breeding applications in large, complex pedigrees. In contrast, our method is designed for estimation of direct effects from sets of nuclear families that are approximately unrelated to one another, the kind of data typical in human genetics. We provide the methods for imputing missing parental genotypes in a software package, snipar (single-nucleotide imputation of parents), that also infers identity-by-descent (IBD) segments shared between siblings and performs genome-wide association and PGI analyses.

We apply our methods to UK Biobank (UKB) data. Among our findings are results showing that, for educational attainment (EA)

[1]Big Data Institute, Li Ka Shing Centre for Health Information and Discovery, University of Oxford, Oxford, UK. [2]Center for Economic and Social Research, University of Southern California, Los Angeles, CA, USA. [3]UCLA Anderson School of Management, Los Angeles, CA, USA. [4]Human Genetics Department, UCLA David Geffen School of Medicine, Los Angeles, CA, USA. [5]Department of Economics, School of Business and Economics, Vrije Universiteit Amsterdam, Amsterdam, the Netherlands. [6]National Bureau of Economic Research, Cambridge, MA, USA. [7]Department of Economics, Harvard University, Cambridge, MA, USA. [8]Department of Economics, New York University, New York, NY, USA. [9]Research Institute of Industrial Economics (IFN), Stockholm, Sweden. [10]Department of Economics, University of Southern California, Los Angeles, CA, USA. ✉e-mail: alextisyoung@gmail.com; augustine.kong@bdi.ox.ac.uk

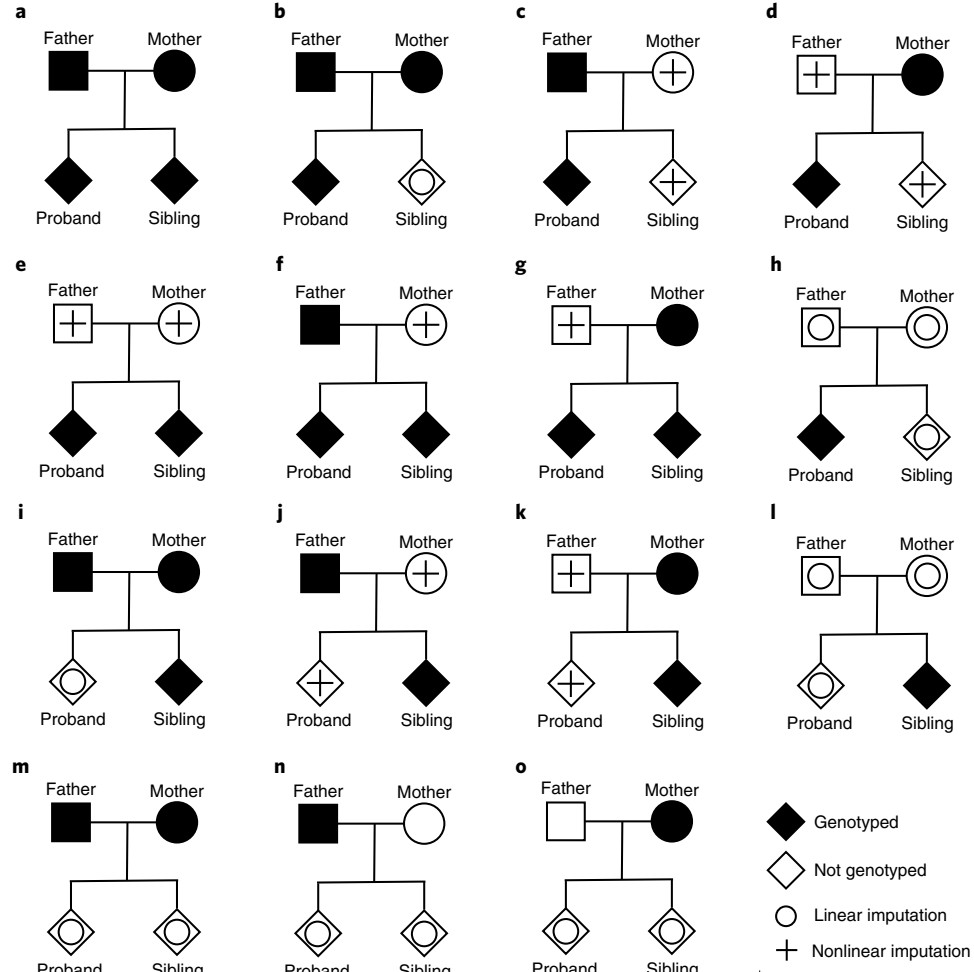

**Fig. 1 | Mendelian imputation for 15 different missing genotype cases.** Displayed are cases where one offspring is phenotyped (the proband), but our framework can handle cases where both offspring are phenotyped. We distinguish between imputations that are linear functions of observed genotypes and imputations that are nonlinear functions of observed genotypes, and thereby add information for estimating the parameters of models (1) and (2). There are seven cases (c–g, j and k) where nonlinear imputations are possible, but we note that in some of these cases (c–e, j and k), the resulting variance-covariance matrix of the observed and imputed genotypes is not of full rank (Extended Data Fig. 1), implying that the full parameter vector of model (2) cannot be identified based on data of that type alone. For example, in case e, the imputed paternal and maternal genotypes are the same, but the imputed sum of paternal and maternal can be used to estimate the parameters of model (3). We detail how to combine information from different data types below. Although we show the case of two offspring here to simplify exposition, our imputation method and software (snipar) can handle any number of genotyped offspring (Methods and Supplementary Note Sections 3 and 5).

and cognitive ability, effects estimated from standard GWASs provide inaccurate estimates of direct genetic effects.

## Results

**Single-locus model.** We consider a model for the effect of a SNP on the phenotypes of two siblings. (We consider a model for two siblings for the purposes of exposition, but our missing-data framework can handle any number of siblings, with or without phenotype observations.) Let $Y_{ij}$ be the phenotype of sibling $j$ in family $i$. Then

$$Y_{ij} = \delta g_{ij} + \alpha_p g_{p(i)} + \alpha_m g_{m(i)} + \epsilon_{ij}; \qquad (1)$$

where $g_{ij}$ is the genotype of sibling $j$ in family $i$, $\delta$ is the direct effect of the SNP and $g_{p(i)}$ and $g_{m(i)}$ are the genotypes of the father and mother in family $i$. SNPs are assumed to be biallelic with alleles '0' and '1', and genotypes are counts of the allele '1' with frequency $f$. Sibling genotypes are conditionally independent of environmental effects

given parental genotypes. Therefore, estimates of direct effects from fitting model (1) are unbiased[12]. We refer to $\alpha_p$ and $\alpha_m$ as 'nontransmitted coefficients' (NTCs), as they are the expected coefficients on the alleles not transmitted to the proband in a regression of proband phenotype on proband genotype and nontransmitted alleles[4]. The NTCs capture IGEs from relatives, in addition to confounding due to population stratification and AM[3,4]. The residual $\epsilon_{ij}$ is uncorrelated with $g_{ij}$, $g_{p(i)}$, and $g_{m(i)}$, but $\epsilon_{i1}$ and $\epsilon_{i2}$ may be correlated. Note that standard GWAS methods that regress proband phenotype onto proband genotype estimate the 'population effect', $\beta = \delta + (\alpha_p + \alpha_m)/2$, which is the direct effect plus the average NTC, $\alpha = (\alpha_p + \alpha_m)/2$.

We also consider a model that adds IGEs from siblings:

$$Y_{i1} = \delta g_{i1} + \eta_s g_{i2} + (\alpha_p - \eta_s/2) g_{p(i)} + (\alpha_m - \eta_s/2) g_{m(i)} + \epsilon_{i1};$$

$$Y_{i2} = \delta g_{i2} + \eta_s g_{i1} + (\alpha_p - \eta_s/2) g_{p(i)} + (\alpha_m - \eta_s/2) g_{m(i)} + \epsilon_{i2};$$

$$(2)$$

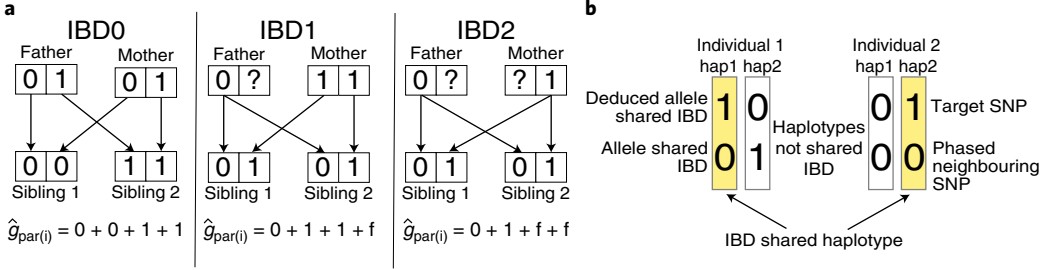

**Fig. 2 | Imputation of missing parental genotypes. a,** Imputation from sibling pairs (Fig. 1e). Given knowledge of the IBD state of the siblings' alleles (alleles coded by '0' and '1'), the sum of the maternal and paternal genotypes can be imputed $\left(\hat{g}_{par(i)}\right)$. If the siblings do not share any alleles IBD, then all four parental alleles are observed (IBD0). If the siblings share one allele IBD, then three parental alleles are observed (IBD1). If the siblings share both alleles IBD, then two parental alleles are observed (IBD2). When parental alleles are unobserved, we impute them with the frequency of allele 1, $f$. The IBD state between siblings changes with the recombination events that occurred during meiosis in the parents and can be inferred (Methods and Supplementary Note Section 9). **b,** Shows how phased data can be used to determine which allele is shared between two individuals who share one allele IBD at a SNP where both are heterozygous. This applies to sibling pairs in IBD1 and parent–offspring pairs, who always shared one allele IBD. A neighboring SNP that has been phased with the target SNP and is homozygous for one individual and heterozygous for the other is used to resolve the uncertainty. For the individual on the left, the 0 allele must be the allele shared with the other individual at the neighboring SNP; thus, through the phased haplotype '1-0' (hap1), it is determined that the 1 allele is the shared allele at the target SNP.

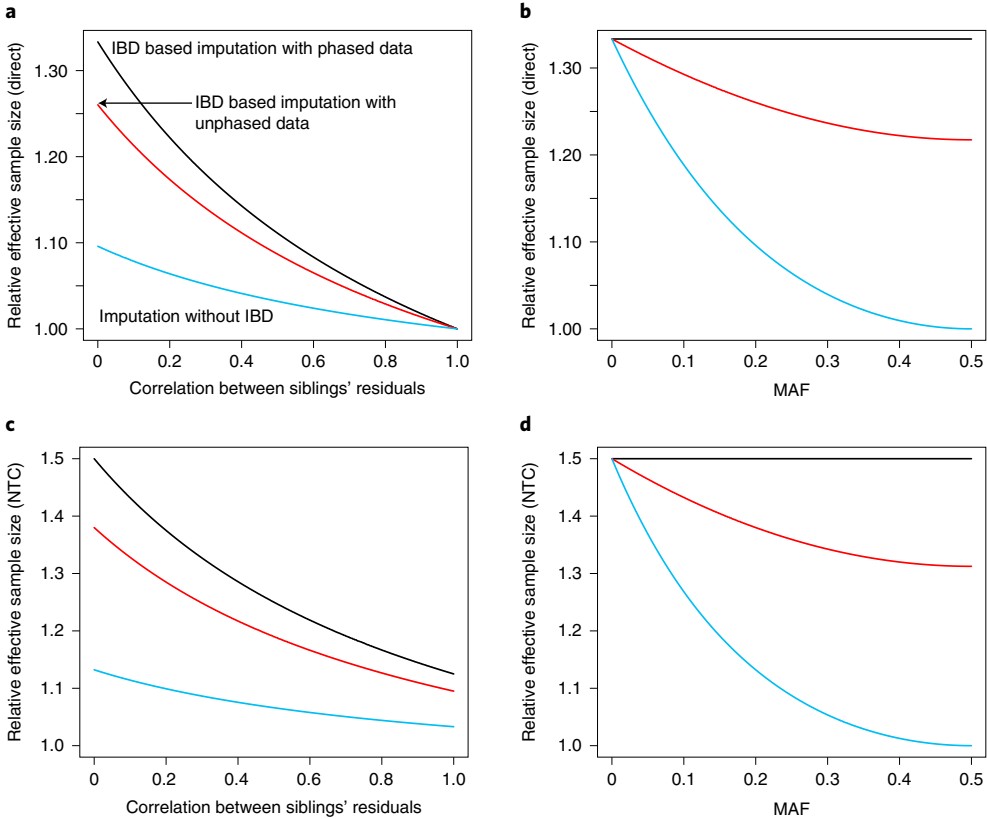

**Fig. 3 | Relative efficiency for estimation of direct effects and NTCs from sibling pairs using different imputation methods.** We compare the theoretical effective sample size for estimation of direct genetic effects and average NTCs from three imputation methods: one that does not use IBD segments (blue)[39], one that uses IBD and unphased data (red) and one that uses IBD and phased data (black). Effective sample size is measured relative to that from using sibling genotypes alone without any imputation and assuming that we have a sample of independent families with two genotyped and phenotyped siblings in each family (Supplementary Note Section 4). **a,** Effective sample size for estimation of the direct genetic effect when MAF is 20% as a function of correlation between siblings' residuals. **b,** Effective sample size for estimation of direct genetic effects as a function of MAF when the correlation between siblings' residuals is zero. (Results follow a similar pattern for other sibling correlations.) For imputation from unphased data, when both siblings are heterozygous and share one allele IBD, which allele is shared IBD cannot be determined (Fig. 2b), so the imputation averages over the two possibilities. When phased data are used, the observed parental alleles can be determined, so the relative efficiency does not depend upon MAF. **c,** The same as for a, but for average NTC. **d,** The same as for b, but for average NTC.

**Table 1 | Examples of regressions and expected regression coefficients for different data types**

| Observed genotypes | $Y_{i1}$ regressed on | $E(\hat{\theta})$ |
|---|---|---|
| Proband (Fig. 1h) | $g_{i1}$ | $\delta + (\alpha_p + \alpha_m + \eta_s)/2$ |
| Sibling pairs (Fig. 1e) | $g_{i1}$ | $\delta$ |
| | $g_{i2}$ | $\eta_s$ |
| | $\hat{g}_{par(i)}$ | $(\alpha_p + \alpha_m - \eta_s)/2$ |
| Father–child pairs (Fig. 1c) | $g_{i1}$ | $\delta$ |
| | $g_{p(i)}$ | $\alpha_p$ |
| | $\hat{g}_{m(i)}$ | $\alpha_m$ |
| Mother–child pairs (Fig. 1d) | $g_{i1}$ | $\delta$ |
| | $\hat{g}_{p(i)}$ | $\alpha_p$ |
| | $g_{m(i)}$ | $\alpha_m$ |
| Trios (Fig. 1b) | $g_{i1}$ | $\delta$ |
| | $g_{p(i)}$ | $\alpha_p$ |
| | $g_{m(i)}$ | $\alpha_m$ |
| Quads (Fig. 1a) | $g_{i1}$ | $\delta$ |
| | $g_{i2}$ | $\eta_s$ |
| | $g_{p(i)}$ | $\alpha_p - \eta_s/2$ |
| | $g_{m(i)}$ | $\alpha_m - \eta_s/2$ |

In the first column, we give the data type in terms of the observed genotypes in the nuclear family, referencing the relevant panel of Fig. 1; in the second column, we give an example of a regression that could be performed using that data type and parental genotypes imputed from the observed genotypes; in the third column, we give the expected column vector of regression coefficients from performing the regression. $Y_{i1}$ is the phenotype of sibling 1 in family $i$; $g_{ij}$ is the genotype of sibling $j$ in family $i$; $g_{p(i)}$ is the paternal genotype, and $\hat{g}_{p(i)}$ is the imputed paternal genotype; $g_{m(i)}$ is the maternal genotype, and $\hat{g}_{m(i)}$ is the imputed maternal genotype; $\hat{g}_{par(i)}$ is the imputed sum of maternal and paternal genotypes; $\delta$ is the direct effect; $\eta_s$ is the indirect sibling effect; and $\alpha_p$ and $\alpha_m$ are, respectively, the paternal and maternal NTCs.

**Table 2 | Summary of data types in the 'White British' UKB subsample**

| Data type | N (probands) |
|---|---|
| Proband and sibling(s) but no parents (Fig. 1e) | 35,197 |
| Proband and parent (Fig. 1c, d) | 3,216 |
| Proband, parent and sibling(s) (Fig. 1f, g) | 312 |
| Proband and both parents (Fig. 1b) | 832 |
| Proband, both parents and sibling (Fig. 1a) | 62 |
| Total | 39,619 |

each SNP as fixed effects in an LMM, which includes a family-level random effect, thereby accounting for phenotypic correlations between siblings (Methods).

In Supplementary Note Section 2, we prove general theoretical properties of multiple regression using this type of imputation: where unobserved covariates are replaced with their expectations given the observed covariates. We prove that estimates remain unbiased and consistent, and that the empirical sampling variance-covariance matrix of the estimates is an unbiased estimate of the true sampling variance-covariance matrix.

**Estimating direct effects using parent–offspring pairs.** Consider a sample of families where the genotype of the proband and its mother have been observed but the father's genotype is unobserved (Fig. 1d). If we impute the father's genotype as $\hat{g}_{p(i)} = E[g_{p(i)}|g_{i1}, g_{m(i)}]$ (Methods), then our theoretical results imply that by performing the regression

$$Y_{i1} = \delta g_{i1} + \alpha_p \hat{g}_{p(i)} + \alpha_m g_{m(i)} + \epsilon_{i1},$$

we obtain unbiased and consistent estimates of $\delta$, $\alpha_p$, and $\alpha_m$. If no imputation is performed, then it is impossible to obtain unbiased estimates of $\delta$ without making an additional assumption, such as $\alpha_p = \alpha_m$. When using phased data, the effective sample size for estimation of direct genetic effects relative to complete observation of parental genotypes (Fig. 1a,b) is approximately equal to 1/2 (Supplementary Note Section 5.3). With unphased data, this increases from a minimum of 1/6 when minor allele frequency (MAF) is 0.5 to a maximum of 1/2 as MAF approaches 0 (Extended Data Fig. 2).

**Imputation from siblings increases power.** Many previous analyses regressed phenotypic differences between siblings onto genotypic differences[7,9,17]. In model (2), this corresponds to:

$$Y_{i1} - Y_{i2} = (\delta - \eta_s)(g_{i1} - g_{i2}) + \epsilon_{i1} - \epsilon_{i2}.$$

This method yields unbiased estimates of $\delta - \eta_s$.

When genotypes are imputed from sibling data (Fig. 1e), we have no information on differences between maternal and paternal genotypes, only their sum. We can express model (2) as (Supplementary Note Section 1)

$$\begin{aligned} Y_{i1} &= \delta g_{i1} + \eta_s g_{i2} + \alpha' g_{par(i)} + \epsilon'_{i1}; \\ Y_{i2} &= \delta g_{i2} + \eta_s g_{i1} + \alpha' g_{par(i)} + \epsilon'_{i2}; \end{aligned} \quad (3)$$

where $\alpha' = (\alpha_p + \alpha_m - \eta_s)/2$, and for some $\epsilon'_{i1}$, $\epsilon'_{i2}$ that are uncorrelated with the siblings' genotypes and $g_{par(i)}$. By imputing $g_{par(i)}$ as the conditional expectation given $g_{i1}$ and $g_{i2}$ and the IBD state of the siblings (Fig. 2a), we can obtain unbiased estimates of the parameter vector $(\delta, \eta_s, \alpha')$ by regression of phenotype jointly onto proband, sibling and imputed parental genotype. Note that by performing

where $\eta_s$ is the IGE from the sibling. Because both proband and sibling genotypes are conditionally independent of environment given parental genotype, estimates of $\delta$ and $\eta_s$ from fitting model (2) are unbiased[4]. Because a parental allele has a 1/2 chance of being passed onto a sibling, the NTCs include one half of the indirect sibling effect, but this is removed from the coefficients on the parental genotypes in model (2) due to inclusion of the sibling genotype.

**Imputing missing genotypes in a nuclear family.** Genotypes in the complete-data model (2) that are unobserved are treated as missing data and imputed based on Mendelian laws. Imputations that are linear functions of observed genotypes do not add information for estimation of the parameters of models (1) and (2). However, there are seven cases out of the $2^4 - 1 = 15$ complete-missing data patterns (Fig. 1 and Extended Data Fig. 1) where nonlinear imputations are possible. These seven cases can be divided into three equivalence classes (up to symmetry): genotyped sibling pairs (Fig. 1e), genotyped parent–offspring pairs (Fig. 1c,d,j,k) and genotyped sibling pairs with one genotyped parent (Fig. 1f,g). When two or more siblings are genotyped, we use the IBD state of the siblings to determine which parental alleles have been observed (Fig. 2a). We provide a method to infer the IBD states of siblings in snipar (Methods and Supplementary Note Section 9). In certain cases, phased genotypes are required to determine which parental alleles have been observed (Fig. 2b), although imputation can proceed without phased data at the cost of lower accuracy. See Methods and Supplementary Note Sections 3 and 5 for further details.

*Estimating effects using imputed genotypes.* We replace unobserved parental genotypes with their imputed values to estimate direct effects and NTCs. We estimate the direct effect and NTCs of

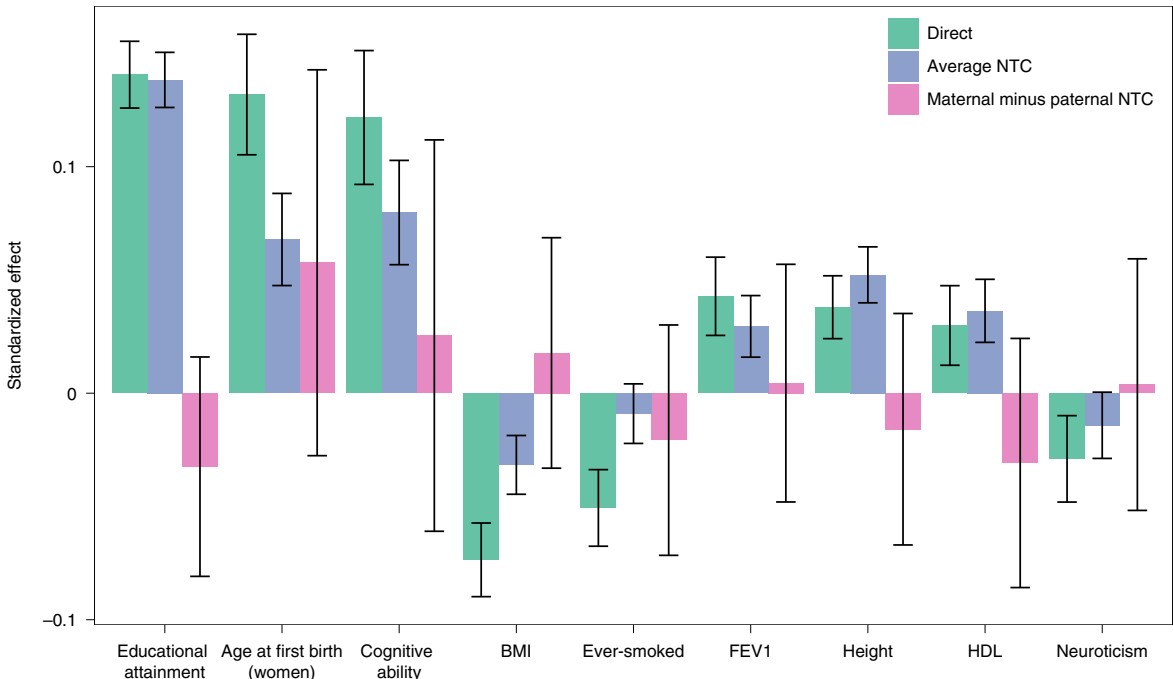

**Fig. 4 | Decomposition of the EA PGI effect assuming no indirect sibling effects.** The standardized effect estimate (standard deviation (s.d.) change in phenotype per s.d. change in PGI) is given along with the 95% confidence interval. Estimates were derived from a sample of 39,619 individuals with imputed and/or observed parental genotypes. Effects were estimated by joint regression of individuals' phenotypes onto their own PGI, and their mother's and father's (imputed or observed) PGIs (Methods). We give the average of the estimated maternal and paternal NTCs, adjusted for bias due to imputation in the presence of AM (Methods), as the 'average NTC' here, and the difference between maternal and paternal NTCs as 'maternal minus paternal'. Phenotypes were adjusted for 40 genetic PCs before analysis. HDL, high-density lipoprotein cholesterol; BMI, body mass index; FEV1, forced expiratory volume in one second.

the imputation, we are able to separately estimate $\delta$ and $\eta_s$, whereas the sibling difference approach can only estimate $\delta - \eta_s$. This shows how Mendelian imputation enables identification of more complex models than approaches that do not perform imputation.

Although our method is able to distinguish IGEs through siblings from direct genetic effects, more precise estimates of direct genetic effects can be obtained by assuming $\eta_s = 0$, at the cost of some bias if $\eta_s \neq 0$. Letting $r$ be the correlation of the siblings' residuals (which will be approximately equal to their phenotypic correlation when the SNP explains a small fraction of the phenotypic variance, as is typical for complex human traits), then the bias in estimates of $\delta$ is $-[(1 + 2r)/(2 + r)]\eta_S$ (Supplementary Note Section 4.3). This is smaller than the bias from regression on differences in sibling genotype, which is $-\eta_s$. For the following results, $\eta_s = 0$ is assumed unless otherwise stated.

Using parental genotypes imputed from phased data increases the effective sample size for estimation of $\delta$ by a factor of $1 + \frac{1-r}{3(1+r)} \geq 1$ relative to using genetic differences between siblings (Supplementary Note Section 4). This has a maximum of 4/3 at $r = 0$ (Fig. 3). We confirmed the theoretical result using simulated data (Extended Data Fig. 3). For estimation of the average NTC, $\alpha = (\alpha_p + \alpha_m)/2$, the effective sample size is increased by a factor of $1 + \frac{1-r/2}{2(1+r)} \geq 1.125$, which has a maximum of 1.5 when $r = 0$. For estimation of both the direct effect and the average NTC, the gain is somewhat lower when using unphased data, depending upon MAF and $r$ (Fig. 3). Using parental genotypes imputed from phased data always gives more precise estimates of direct effects and average NTCs than using unphased data or genetic differences between siblings (Supplementary Note Section 4).

**Combining different missing data types.** The different missing data patterns in Fig. 1 enable estimation of different linear transformation

of the parameters of model (2), $(\delta, \eta_s, \alpha_p, \alpha_m)$, with examples in Table 1. Although not all data types enable identification of $(\delta, \eta_s, \alpha_p, \alpha_m)$, they can contribute to an overall estimate of $(\delta, \eta_s, \alpha_p, \alpha_m)$ when the combination of data types enables identifiability. If genotypes are observed or imputed as outlined above, then a single regression that combines all the data types together gives consistent estimates of the full parameter vector provided that the resulting regression design matrix is not collinear (Supplementary Note Section 2). This is the approach we adopt for applications to real data (Methods). Alternatively, a form of multivariate meta-analysis can be used (Supplementary Note Section 6).

**Imputing missing parental genotypes in the UKB.** We applied our methods to the 'White British' subsample of the UKB[22] (Methods). Using KING[23], we identified a sample of 39,619 individuals for which parental genotypes were observed or could be imputed (Table 2). We inferred IBD segments for sibling pairs using snipar (Methods). We validated the IBD inference using 31 families with two siblings and both parents genotyped, finding that the IBD states were correct 99.65% of the time (Supplementary Table 1). The IBD sharing statistics of the siblings were close to theoretical expectations (Extended Data Fig. 4).

Using snipar, we imputed missing parental genotypes from phased haplotypes for 1,586,010 SNPs, the union of the genotyping array SNPs and HapMap3 SNPs with MAF > 1%. We found that there was negligible bias in the imputed genotypes (Methods).

We tested the performance of our method in realistic simulations based on genetic data from the UKB 'White British' sample (Supplementary Note Section 12.2). We simulated traits affected by AM, parental IGEs, vertical transmission[24] (where a phenotype of the parent(s) affects the phenotype of the offspring through the environment), vertical transmission and AM[25] and population

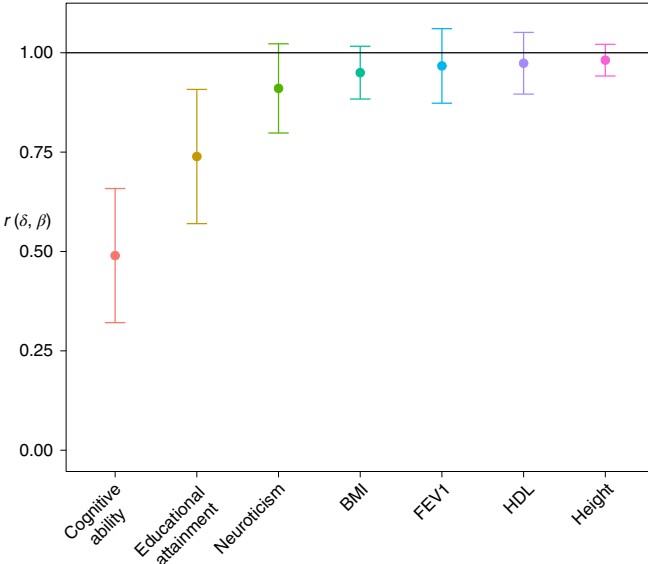

**Fig. 5 | Estimates of genome-wide correlation between direct and population effects, $r(\delta, \beta)$.** The estimate is given along with the 95% confidence interval. Direct effects are causal effects due to inheritance of alleles; population effects are estimated by standard GWASs and include direct effects, indirect effects from relatives and confounding due to population stratification and AM. We estimated the correlation between direct and population effect estimates using summary statistics derived from a sample of 39,619 individuals from the 'White British' subsample of the UKB where parental genotypes were imputed, using the developed methods, or observed (Methods). Phenotypes were adjusted for 40 genetic PCs before analysis. We do not show the results for age at first birth in women and ever-smoked here due to their large standard errors (see Supplementary Table 5).

stratification. We did not detect any bias in estimates of direct effects across the simulated traits (Supplementary Table 2).

**Direct and indirect effects of an education PGI.** We analyzed the effects of an EA PGI on nine phenotypes in the UKB (Methods). By using observed and imputed parental PGIs, we obtained unbiased estimates of the indirect sibling effects ($\eta_s$) of the EA PGI (Extended Data Fig. 5 and Supplementary Table 3), which were not statistically significant for any phenotype ($P > 0.05$, two-sided $Z$-test). Assuming $\eta_s = 0$, we obtained more precise estimates of direct effects and average NTCs (Fig. 4 and Supplementary Table 4).

Statistically significant estimates of direct effects were obtained for all phenotypes; and statistically significant average NTCs ($P < 0.05$, two-sided $Z$-test) were obtained for all phenotypes other than ever-smoked and neuroticism. Estimates of maternal minus paternal NTCs were not statistically significant for any of the phenotypes ($P > 0.05$, two-sided $Z$-test), although power for this analysis was limited.

Across the phenotypes, the direct/(direct + average NTC) ratio was similar to a previous analysis of an EA PGI in Icelandic data[4], except for EA, where here the ratio was 0.50 (compared to 0.70 in Iceland). The fraction of variance explained by the direct effect is the square of this ratio, implying that only 25% of the variance explained by the EA PGI is due to its direct effect alone (compared to 49% in Iceland).

**Genome-wide association analyses for nine phenotypes.** We estimated the direct effects, NTCs and population effects of 1,586,010 SNPs with MAF > 1% on nine phenotypes (Methods). Phenotypes

were adjusted for 40 genetic PCs before SNP effects were estimated. In our PGI analyses (above), we found no evidence for substantial IGEs from siblings. Therefore, to increase precision, we estimated effects assuming $\eta_s = 0$.

At these sample sizes, power is limited for analysis of direct effects and NTCs of individual SNPs. We therefore focused on estimating the genome-wide correlation between direct and population effects, $r(\delta, \beta)$ (Methods). This measures the degree to which population effects, as estimated by standard GWAS, reflect direct genetic effects. We also estimated $r(\delta, \alpha)$, the genome-wide correlation between direct effects and average NTCs. To estimate the correlations, we used a moment-based estimator that adjusts for the known sampling variance-covariance matrix of the estimates (Supplementary Note Section 11).

We estimated $r(\delta, \beta)$ and $r(\delta, \alpha)$ for the phenotypes simulated from genetic data from the UKB 'White British' subsample (Supplementary Table 2). For phenotypes affected by direct effects and parental IGEs in a random-mating population, $r(\delta, \alpha)$ is the correlation between direct effects and average parental IGEs, which our simulation results confirmed.

A plausible model for IGEs is vertical transmission[24]. We simulated a phenotype affected by vertical transmission for 20 generations of random mating, reaching an approximate equilibrium. For this phenotype, $r(\delta, \beta) = 0.953$ (s.e. = 0.009).

When there is population stratification or AM, average NTCs (and therefore population effects) capture effects due to other genetic and environmental factors with which the SNP is correlated due to nonrandom mating, in addition to IGEs from relatives. We simulated 20 generations of AM for the same vertical transmission phenotype model, reaching an approximate equilibrium. For this phenotype, $r(\delta, \beta) = 0.883$ (s.e. = 0.009). For a phenotype affected by direct effects and a random environmental component (no indirect effects or vertical transmission), $r(\delta, \beta) = 0.949$ (s.e. = 0.008) after 20 generations of AM. We also simulated a phenotype affected by direct effects and population stratification, for which $r(\delta, \beta) = 0.917$ (s.e. = 0.007). These results show that population stratification, AM, and vertical transmission, along with their interactions, can lead to $r(\delta, \beta)$ substantially below 1.

Across the nine phenotypes, $r(\delta, \beta)$ was not statistically distinguishable from 1 ($P > 0.05$, one-sided $Z$-test for $r(\delta, \beta) < 1$) except for EA ($r(\delta, \beta) = 0.739$, s.e. = 0.086, $P = 1.2 \times 10^{-3}$) and cognitive ability ($r(\delta, \beta) = 0.490$, s.e. = 0.086, $P = 1.6 \times 10^{-9}$) (Fig. 5). We also estimated $r(\delta, \alpha)$ (Supplementary Table 5), finding negative correlations (discussed below) for cognitive ability ($r(\delta, \alpha) = -0.588$, s.e. = 0.094, $P = 3.1 \times 10^{-10}$, two-sided $Z$-test for $r(\delta, \alpha) \neq 0$) and neuroticism ($r(\delta, \alpha) = -0.421$, s.e. = 0.190, $P = 0.027$), and a positive correlation for height ($r(\delta, \alpha) = 0.666$, s.e. = 0.270, $P = 0.014$). For height, the results are similar to simulation results for a phenotype affected by direct effects and AM (Supplementary Table 2), consistent with previous analyses showing that, for height, AM is strong and parental IGEs are weak[4,26].

**Evidence for residual stratification.** To investigate whether residual population stratification that persists after adjustment for PCs[7,8] contributes to the low correlations between direct and population effects for EA and cognitive ability, we adjusted those phenotypes for birth coordinates and the location where each individual was assessed, in addition to PCs (Methods). This increased the estimated correlation for EA to $r(\delta, \beta) = 0.791$ (s.e. = 0.066), an increase of 0.053 (s.e. = 0.045; $P = 0.124$ from a one-sided $Z$-test for an increase); and increased the estimated correlation for cognitive ability to $r(\delta, \beta) = 0.568$ (s.e. = 0.088), an increase of 0.078 (s.e. = 0.064; $P = 0.113$).

PCs based on rare variants or IBD sharing capture recent population structure better than PCs based on common variants[8], which we used to adjust the phenotypes in our primary analysis. To better

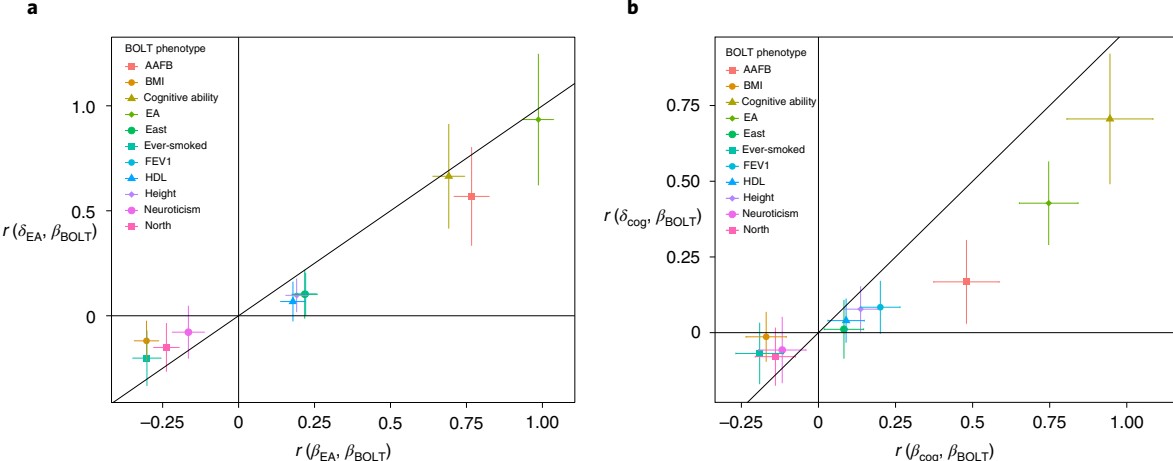

**Fig. 6 | Correlations between population effects from an independent sample and direct and population effects on EA and cognitive ability.** We computed correlations between summary statistics from our main analysis of 39,619 individuals from the 'White British' subsample of the UKB, where parental genotypes were imputed or observed, and population effects estimated by BOLT-LMM in an independent sample of 276,419 unrelated individuals from the 'White British' subsample of the UKB (Methods). **a**, Correlation between the population effects on EA estimated in our primary analysis and population effects for nine phenotypes and birth coordinates estimated in the unrelated sample, $r\left(\beta_{EA}, \beta_{BOLT}\right)$, on the x axis; and the correlation between the direct effects on EA estimated in our primary analysis and the population effects in the unrelated sample, $r\left(\delta_{EA}, \beta_{BOLT}\right)$, on the y axis. **b**, Correlation between the population effects from the unrelated sample and direct effects on cognitive ability, $r(\delta_{cog}, \beta_{BOLT})$, on the y axis; and the correlation between the population effects in the unrelated sample and population effects on cognitive ability from our primary analysis, $r\left(\beta_{cog}, \beta_{BOLT}\right)$, on the x axis. The vertical error bars give the 95% confidence intervals for the correlation with the direct effects, and the horizontal error bars give the 95% confidence intervals for the correlation with population effects. We give numerical results along with P values for differences between correlations with direct and population effects in Supplementary Table 7. AAFB, age at first birth (in women); north, north birth coordinate; east, east birth coordinate.

adjust for recent structure, we adjusted EA and cognitive ability for the top 40 PCs of the IBD relatedness matrix in addition to the 40 common variant PCs used originally (Methods). This increased the estimated correlation for EA to $r(\delta, \beta) = 0.785$ (s.e. = 0.076), an increase of 0.048 (s.e. = 0.019; $P = 5.7 \times 10^{-3}$ from a one-sided Z-test for an increase); and increased the estimated correlation for cognitive ability to $r(\delta, \beta) = 0.621$ (s.e. = 0.058), an increase of 0.131 (s.e. = 0.041; $P = 6.5 \times 10^{-4}$).

To further investigate the contribution of residual stratification, we computed correlations between genetic associations with birth coordinates (adjusted for PCs) and direct effects, average NTCs, and population effects (Methods and Supplementary Table 6). The correlations with birth coordinates reflect the degree to which SNP effects on phenotypes, after adjustment for PCs, are correlated with the geographic structure in the population[27]. Estimated correlations between birth coordinates and direct effects were attenuated toward zero relative to correlations between birth coordinates and average NTCs and population effects. Furthermore, correlations between birth coordinates and average NTCs and population effects tended to line up along a south-east to north-west axis (Extended Data Fig. 6), likely reflecting the phenotypes' correlations with socioeconomic status, genetic structure in the UK population and geographic variation in socioeconomic status across the United Kingdom[27,28].

**Correlations across phenotypes for direct and population effects.**
We sought to test whether correlations between population effects on EA and cognitive ability and population effects on other phenotypes are inflated due to IGEs and confounding factors. To remove the influence of sampling correlations between effect estimates due to overlapping samples, we estimated population effects in a sample of 276,419 unrelated individuals who are unrelated (third degree or less) to the sample used in our primary analysis. We used BOLT-LMM to estimate population effects on the same nine phenotypes as in our primary analysis as well as north and east birth coordinates (Methods). We refer to the population effects estimated

in this sample as $\beta_{BOLT}$ for an unspecified phenotype and $\beta_{BOLT:EA}$ to refer to a specific phenotype (in this case EA). This enabled us to estimate, without needing to adjust for sampling correlations, correlations between direct effects on EA and cognitive ability ($\delta_{EA}$ and $\delta_{cog}$) from our primary analysis and population effects on nine phenotypes and north and east birth coordinates from the unrelated sample ($\beta_{BOLT}$), which we refer to as $r(\delta_{EA}, \beta_{BOLT})$ and $r(\delta_{cog}, \beta_{BOLT})$. We compared these to correlations between population effects on EA and cognitive ability ($\beta_{EA}$ and $\beta_{cog}$) from our primary analysis and population effects on nine phenotypes and north and east birth coordinates from the unrelated sample ($\beta_{BOLT}$), which we refer to as $r(\beta_{EA}, \beta_{BOLT})$ and $r(\beta_{cog}, \beta_{BOLT})$ (Supplementary Tables 5 and 7).

Across the phenotypes, population effects were more strongly correlated with population effects on EA and cognitive ability than with direct effects on EA and cognitive ability (Fig. 6), supporting the hypothesis that IGEs and confounding factors inflate correlations between population effects. The largest estimated difference was $r(\beta_{cog}, \beta_{BOLT:EA}) - r(\delta_{cog}, \beta_{BOLT:EA}) = 0.319$ (s.e. = 0.073), suggesting either shared confounding in population effects on EA and cognitive ability, or shared IGEs that are not highly correlated with direct effects on cognitive ability, or both.

## Discussion

We introduce Mendelian imputation as a tool to perform genetic association analyses. Conceptually, this is similar to multipoint linkage analysis performed with pedigrees[29], familial imputations[30,31] and methods in animal breeding[19]. However, our approach focuses on imputing missing parental genotypes in a nuclear family rather than in the large pedigrees typical in animal breeding. We found that our imputation method improves on a recent method developed for imputing missing genotypes in a nuclear family, AlphaFamImpute[21], both in terms of bias and $R^2$ between imputed and actual parental genotypes (Methods and Supplementary Table 8). The improvement in $R^2$ derives from the use of pre-phased genotypes, enabling resolution of which parental alleles have been observed in certain doubly

heterozygous cases (Fig. 2b). Our approach could be extended to use genotypes of other relatives of the missing parent(s). However, many data sets contain genotypes of siblings and/or parent–offspring pairs from families that have no known pedigree relation, and for these data sets, our approach provides close to optimal recovery of the genotypes of the missing parent(s).

Mendelian imputation, used appropriately, produces unbiased estimates of parameters along with valid sampling errors. This makes it rather unique among single imputation methods in situations where the amount of missing information is substantial[32]. These properties derive from the fact that missing genotypes are imputed as nonlinear functions of observed genotypes, using Mendelian laws, thereby adding information for parameter estimation without introducing noise. Mendelian imputation enables integrated analysis of different data types, maximizing power and enabling identification of models than cannot be identified without imputation.

We examined the degree to which GWAS estimates reflect direct effects by estimating the genome-wide correlation between direct and population effects, finding that population effects and direct effects are not highly correlated (<0.9) for EA and cognitive ability. We found evidence that this is in part due to recent structure in the population that is captured by PCs of the IBD relatedness matrix, but not by PCs computed from common variants[8]. Our simulation results (Supplementary Table 2) suggest that a combination of vertical transmission and AM[24,25] may also contribute to the low correlation between direct and population effects.

Another phenomenon that may contribute is ascertainment: If direct effects and NTCs are not very strongly correlated, but both are also correlated with ascertainment, then collider bias[33] could push the correlation estimate in the negative direction and reduce correlations between direct and population effects. Analysis of simulated phenotypes under ascertainment supports this hypothesis, where strong ascertainment reduced $r(\delta, \alpha)$ to $-0.264$ (s.e. = 0.091) for a phenotype with uncorrelated direct effects and parental IGEs (true $r(\delta, \alpha) = 0$), which may explain why we observed negative $r(\delta, \alpha)$ for cognitive ability and neuroticism, as observations for these phenotypes are ascertained for higher education and lower neuroticism[34,35].

If population effects are not highly correlated with direct effects, then this has implications for genetic prediction methods. For example, for prediction of differences between embryos, only direct effects are relevant, so selecting embryos using population effects would perform poorly compared to using direct effects[36,37] (assuming equal precision of direct and population effect estimates) and could introduce confounding related biases. Confounding related biases can also lead to spurious inferences in mendelian randomization[10] and studies of selection[7,11]. If NTCs are substantial and imperfectly correlated with direct genetic effects, then prediction accuracy could be increased by including predictors based on NTCs of parental genotypes in addition to predictors based on proband genotypes.

We found evidence that correlations between GWAS summary statistics on many of the nine phenotypes examined here and EA and cognitive ability are inflated by factors other than direct effects. Application of the methods developed here to larger sample sizes will enable us to estimate the relative contribution of direct effects, IGEs, and confounding factors to estimates of genetic correlations[38].

By analyzing an EA PGI, we observed a lower direct/(direct + average NTC) ratio for EA than was observed in Iceland[4]. This implies that the combined influence of IGEs, population stratification, and AM is stronger in the UK than in Iceland. The PGI was constructed from standard GWAS summary statistics, so the average NTCs of the PGI could reflect bias and/or IGEs in the original GWAS summary statistics. Future studies could examine prediction using PGIs constructed from direct effect estimates, which do not have these biases.

Collection of genetic data on close relatives is inevitable as sample sizes grow larger. However, samples of close relatives will remain much smaller than samples of distantly related individuals. We see data on unrelated individuals as one possible pattern of missing data in a framework for human genetic analysis that treats the nuclear family as the fundamental unit of analysis rather than the individual. By combining information from different missing data patterns, we will be able to construct a more accurate picture of the role of genetics in human phenotype variation.

## Online content

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

## Methods

**Imputation from sibling pairs.** Given the genotypes of a sibling pair (Fig. 1e) and the IBD state of the alleles (which alleles are shared by descent from the parents), the observed parental alleles can be determined (Fig. 2a).

Because parent-of-origin cannot be determined from sibling data, we impute the sum of maternal and paternal genotypes (Fig. 2). Let $g_{\mathrm{par(i)}} = g_{\mathrm{m(i)}} + g_{\mathrm{p(i)}}$ be the sum of the genotype of the mother ($g_{\mathrm{m(i)}}$) and the genotype of the father ($g_{\mathrm{p(i)}}$) in family $i$, and let $g_{i1}$ and $g_{i2}$ be the genotypes of the two siblings. We compute $\mathrm{E}\left[g_{\mathrm{par(i)}} | g_{i1}, g_{i2}, \mathrm{IBD}_i\right]$, where $\mathrm{IBD}_i$ is the IBD state of the two siblings. Because all four alleles are observed in IBD state 0, we have that

$$\mathrm{E}\left[g_{\mathrm{par(i)}} | g_{i1}, g_{i2}, \mathrm{IBD}_i = 0\right] = g_{i1} + g_{i2} = g_{\mathrm{par(i)}}.$$

When we do not observe a parental allele, we impute it using the population frequency of allele 1, $f$. Therefore,

$$\mathrm{E}\left[g_{\mathrm{par(i)}} | g_{i1}, g_{i2}, \mathrm{IBD}_i = 2\right] = g_{i1} + 2f.$$

When the siblings share one allele IBD, let $g_{i2}^{\neg s}$ be 1 if the allele in sibling 2 that is not shared IBD with sibling 1 is allele 1, and let $g_{i2}^{\neg s}$ be 0 otherwise. If we impute by

$$\mathrm{E}\left[g_{\mathrm{par(i)}} | g_{i1}, g_{i2}, \mathrm{IBD}_i = 1\right] = g_{i1} + g_{i2}^{\neg s} + f,$$

then the squared correlation between imputed and observed parental genotype is ¾. This is because we observe two parental alleles in IBD2, with probability ¼; three parental alleles in IBD1, with probability ½; and four parental alleles in IBD0, with probability ¼; giving an average of 3 observed parental alleles. The squared correlation is higher than based on best linear unbiased imputation, 2/3 (Supplementary Note Appendix D).

When in IBD1 (the siblings share one allele IBD), the alleles not shared are known unless both siblings are heterozygous. When both are heterozygous, information from neighboring phased SNPs can be used to resolve the uncertainty (Fig. 2b). However, without phased data, imputation can proceed by averaging over the two possibilities (shared allele is 0 versus shared allele is 1), giving

$$\mathrm{E}\left[g_{\mathrm{par(i)}} | g_{i1} = 1, g_{i2} = 1, \mathrm{IBD}_i = 1\right] = 1 + 2f,$$

at the cost of lower correlation with the unobserved parental genotype (Supplementary Note Section 3.2).

We generalize the above approach to imputing from genotype observations on three or more siblings (Supplementary Note 3.2.1). When $n_i$ siblings have been observed in family $i$, on average we observe $4\left(1 - 2^{-n_i}\right)$ parental alleles, so the imputation approaches full recovery of the combined parental genotype as the number of siblings increases.

**Imputation from parent–offspring pairs.** Consider imputing the genotype of a proband's father given observations of the proband and mother's genotypes (Fig. 1d). We impute as the expectation given the proband and mother's genotype: $\hat{g}_{\mathrm{p(i)}} = \mathrm{E}\left[g_{\mathrm{P(i)}} | g_{i1}, g_{\mathrm{m(i)}}\right]$. Given the proband's paternally inherited allele, one half of the paternal genotype is determined, and the expectation of the other half is given by $f$. The resulting squared correlation of the paternal genotype with the imputed paternal genotype is therefore 1/2, higher than with the best linear unbiased imputation, 1/3 (Supplementary Note Appendix D).

Similar to the sibling case, the paternally inherited allele of the proband is known unless both mother and proband are heterozygous, in which case phased data are needed to resolve the uncertainty (Fig. 2b). Without phased data, the unobserved paternal genotype can be imputed by averaging over the two possible inheritance patterns (Supplementary Note Section 5.1), giving $\mathrm{E}\left[g_{\mathrm{p(i)}} | g_{i1} = 1, g_{\mathrm{m(i)}} = 1\right] = 2f$. The loss of information relative to phased data increases with increasing heterozygosity.

We generalize the imputation procedure to incorporate situations where two or more siblings' genotypes and one parent's genotype have been observed (Fig. 1f, g). We leverage both IBD sharing between siblings and the observed parent to efficiently impute the missing parent's genotype, giving methods for both phased and unphased data (Supplementary Note Sections 5.1.1 and 5.2).

**Estimation of SNP effects.** Phenotype observations of siblings are correlated through both shared genetic factors and shared environmental factors. To obtain efficient estimates of SNP effects, the phenotypic correlations between siblings should be modeled. We implemented an LMM in snipar that achieves this by modeling the mean phenotype within each family as a random effect. Let $Y_{ij}$ be the phenotype of sibling $j$ in family $i$; then, assuming the overall mean of the phenotype is zero,

$$Y_{ij} = X_{ij}\theta + \mu_i + \epsilon_{ij}; \mu_i \sim N\left(0, \sigma_F^2\right); \epsilon_{ij} \sim N\left(0, \sigma_\epsilon^2\right);$$

where $X_{ij}$ are the mean-centered (observed or imputed) genotypes; $\theta$ is the corresponding vector of parameters; $\mu_i$ is the mean in family $i$, which we model as

a mean-zero normally distributed random effect with variance $\sigma_F^2$, independent for each family; and $\epsilon_{ij}$ is the residual for individual $j$ in family $i$, independent for each individual. This implies that, conditional on $X$, the phenotypic correlation of siblings is $\sigma_F^2/(\sigma_F^2 + \sigma_\epsilon^2)$.

The columns of X and $\theta$ depend upon the data type and model being estimated (Table 1). The default is for the columns of X to be the individual's genotype, the individual's father's imputed or observed genotype, and the individual's mother's imputed or observed genotype, with $\theta = \left[\delta, \alpha_p, \alpha_m\right]^T$. When only sibling genotypes are available, to prevent collinearity, the columns of X reduce to the individual's genotype and the imputed sum of maternal and paternal genotypes, and $\theta = \left[\delta, \alpha = \left(\alpha_p + \alpha_m\right)/2\right]^T$. We also provide an option in snipar to add the proband's siblings' genotypes to the regression to fit indirect effects from siblings. For estimation of the effects of genome-wide SNPs, snipar first infers the variance components $\sigma_F^2$ and $\sigma_\epsilon^2$ by maximum likelihood for a null model without any SNP effects, which can be done in O($n$) computations (Supplementary Note Section 10). We then fix the variance components at their maximum likelihood estimate for estimation of the SNP effects. Given the variance components, the estimate of $\theta$ can be obtained analytically in O($n$) computations.

**Effect of population structure.** In the results above and the main text, we have assumed random mating. When population structure is present, this leads to bias in the imputed parental genotypes. We analyze the consequences of this in Supplementary Note Section 7. In general, estimates of NTCs are biased by structure, with the bias increasing with Wright's $F_{st}$. Bias is introduced into estimates of direct effects when data types with different numbers of observed parental alleles are mixed together. For imputation from sibling pairs, the number of observed parental alleles differs with the IBD state of the siblings, introducing a bias into estimates of $\delta$ that is approximately equal to $F_{st}\alpha/2$ when $F_{st}$ is small. For relatively homogeneous samples, any such bias is therefore likely to be negligible at the individual SNP level. Further, SNPs with large values of $F_{st}$ will tend to be filtered out during quality control because they violate Hardy–Weinberg Equilibrium.

In Supplementary Note Section 7.2, we derive an alternative estimator for $\delta$ that splits the regression by the number of observed parental alleles, and we prove that this estimator is not biased by population structure. Although this estimator is more robust, it is less precise than the estimator described above, which performs a single regression using all individuals, irrespective of the number of parental alleles observed. However, the alternative estimator for $\delta$ is still more precise than the estimator based on genetic differences between siblings, having an effective sample size $1 + \frac{1-r}{6(1+r)} \geq 1$ times higher, with a maximum of 7/6 when $r = 0$.

**PGI analyses using imputed parental genotypes.** Consider a PGI composed of $L$ SNPs for the father in family $i$:

$$PGI_{p(i)} = \sum_{l=1}^{L} w_l g_{p(i)l}, \tag{5.1}$$

where $w_l$ is the weight of SNP $l$, and $g_{p(i)l}$ is the genotype of the father at SNP $l$. If the father is not genotyped, then the imputed PGI is:

$$\widehat{PGI}_{p(i)} = \sum_{l=1}^{L} w_l \hat{g}_{p(i)l}, \tag{5.2}$$

where $\hat{g}_{p(i)l}$ is imputed as described above. Assuming the $L$ SNPs are in linkage equilibrium, then theoretical results for single SNP analyses carry over.

In practice, linkage disequilibrium (LD) between some of the SNPs is expected. However, if many SNPs from across the genome contribute to the PGI, only a small fraction of the pairs of SNPs will have non-negligible correlations due to local LD, and the effect on the imputations and estimates would be negligible. However, for a phenotype with AM, contributing SNPs can become correlated regardless of their physical positions[3,40,41]. Because each SNP is imputed individually without conditioning on other SNPs that contribute to the PGI, the imputed PGIs are not exactly the conditional expectations given the observed PGIs. Consider a model for the association between a phenotype and a PGI:

$$Y_{i1} = \delta PGI_{i1} + \alpha(PGI_{p(i)} + PGI_{m(i)}) + \epsilon_{i1},$$

where $PGI_{ij}$ is the PGI of sibling $j$ in family $i$, and $PGI_{m(i)}$ is the PGI of the mother in family $i$. We show in the Supplementary Note Section 8 that using imputed parental PGIs in place of observed parental PGIs does not introduce bias to estimates of $\delta$, even when AM is present, when the number of SNPs, $L$, is large. However, a slight bias in estimates of NTCs is introduced. For example, if using parental genotypes imputed from sibling pairs with phased data, the estimate of $\alpha$ would be inflated by a factor of $(1 + r_{am})/(1 + r_{am}/2)$, where $r_{am}$ is the equilibrium correlation between maternal and paternal PGI. We note that, even with fully observed genotypes, AM implies that $\alpha$ captures confounding due to correlation between the parental PGI and the genetic component of the phenotype that would be uncorrelated with the PGI under random mating, as described previously[4,42].

**UKB sample.** We used the UKB sample that had been identified by UKB to have predominantly 'White British' ancestry[22]. We filtered out individuals identified by the UKB as having excess relatives, excess heterozygosity, or sex chromosome aneuploidy. We used the kinship coefficients computed by the UKB to identify individuals with a first-degree relative, where a first-degree relation is defined as a kinship coefficient of 0.177 and above[23]. We used KING[23] with the '–related –degree 1' options to infer the sibling and parent–offspring relations within that set of individuals (Table 2). We identified 157 duplicates/monozygotic twins and removed one from each pair from further analyses. There were 19,290 sibling pairs from 17,289 sibships, including 913 sibships of size greater than 2, with a maximum size of 6.

Haplotypes for the SNPs that were present on both the UKB Axiom and the UK BiLEVE genotyping arrays and that passed quality control were provided by the UKB[22]. Phasing was performed using SHAPEIT3 (ref. [43]) and the 1000 Genomes Phase 3 dataset[44] as a reference panel. This resulted in phased haplotypes for a set of 658,720 autosomal SNPs with an estimated switch error rate of 0.229%[22].

In addition, we used SHAPEIT2 with the –duohmm option (with -W 5 parameter) to phase 1.1 million HapMap3 SNPs with MAF > 1% from the imputed genotype data provided by the UKB. The 'duohmm' option takes advantage of parent–offspring relations to improve phasing. We merged the haplotypes provided by UKB with the haplotypes for HapMap3 SNPs using QCTOOL, giving haplotypes for 1,652,145 unique SNPs, 1,586,010 of which had MAF > 1%.

To compute the PCs of the IBD relatedness matrix, we used KING[23] with the –ibdseg option to infer IBD segments between all pairs in the 39,619 individuals from the 'White British' subsample of the UKB for which parental genotypes were observed or could be imputed, along with their first-degree relatives, giving a total sample of 44,553. The relatedness between two individuals based on IBD sharing was calculated as $(1/2) \times P(IBD1) + P(IBD2)$, where $P(IBD1)$ and $P(IBD2)$ are the fractions of the autosome shared in IBD1 and IBD2 segments respectively. We extracted the eigenvectors with the 40 largest eigenvalues from the resulting relatedness matrix.

**UKB phenotypes.** We analyzed EA; standing height (Data Field 50); body mass index (Data Field 21001); neuroticism score (Data Field 20127); whether an individual answered that they had ever smoked or not (Data Field 20160), encoded as a binary variable; cognitive ability, derived from a test of 'fluid intelligence' (Data Field 20016); high-density lipoprotein cholesterol (Data Field 30760); forced expiratory volume in one second (Data Field 3063); age at first live birth (in women) (Data Field 2754); and north (Data Field 129) and east (Data Field 130) birth coordinates. For EA, we converted the answers to the qualifications question (Data Field 6138) to years of education according to the method used in the most recent GWAS of EA[45]. For all phenotypes, we regressed out age, age², age³, sex and interactions between sex and age, age², and age³, along with the 40 genetic PCs provided by the UKB. For quantitative phenotypes, we normalized the phenotypes to have variance 1 separately in males and females.

To further investigate the impact of residual population stratification on EA and cognitive ability, we adjusted EA and cognitive ability for birth coordinates and the center where they were assessed (Data Field 54), in addition to the covariates listed above. To do this, we regressed the phenotype onto the covariates listed above and linear and nonlinear functions of north and east birth coordinates, assessment center coded as a categorical variable and interactions between assessment center and north and east birth coordinates and the squares and cubes of north and east birth coordinates. For the nonlinear functions of north and east birth coordinates, we used north coordinate, its square and cube; east coordinate, its square and cube; and all pairwise products between north coordinate and its square and cube, and east coordinate and its square and cube.

**IBD inference.** We developed a hidden Markov model (HMM), implemented in snipar, to infer IBD segments shared between siblings (Supplementary Note Section 9). The HMM models the joint distribution of a sibling pair's (unphased) genotypes at a SNP conditional on the IBD state. To account for LD between nearby SNPs, we weighted the contribution of siblings' genotypes at each SNP to the overall likelihood for the chromosome by the inverse of the LD score of the SNP. We calculated the LD scores using the LD Score Regression[46] software with a 1 centimorgan (cM) window. The probability of transitioning from one IBD state to another is inferred from the genetic distance between the SNPs. We account for genotyping errors, which requires a parameter $\gamma$ for the probability of a genotyping error. We smooth the IBD segments inferred by the HMM to remove short segments that are improbable based on their length in cM and whose neighboring segments have the same IBD state. This requires a parameter, $m$, the minimum allowed length (in cM) of an IBD segment that differs from its adjacent segments.

We optimized the parameters $\gamma$ and $m$ by using 31 families where two siblings and both parents are genotyped, and therefore the true IBD state can be inferred for many SNPs (Supplementary Note Section 9). We found that $(\gamma, m) = (10^{-4}, 0.01$ cM) gave the highest probability of inferring the true IBD state, 99.65%. We give the proportions of SNPs with inferred IBD states 0, 1 and 2 as a function of the true IBD state in Supplementary Table 1.

We compared this to IBD segments inferred by KING using the –ibdsegs option, which had an overall probability of inferring the true IBD state of 98.5%.

Our method therefore gave around a fourfold reduction in IBD errors compared to KING. Furthermore, we found that the distribution of IBD states inferred by KING diverged substantially from the theoretical expectation near the ends of chromosomes and centromeres, whereas the distribution of IBD states inferred by snipar was close to theoretical expectations from end to end (Extended Data Fig. 4).

**Imputation of missing parental genotypes.** Using the inferred IBD segments (above), we imputed missing parental genotypes from phased haplotypes for 1,586,010 SNPs, the union of the genotyping array SNPs and the HapMap3 SNPs with MAF > 1%. We examined the bias in the imputed parental genotypes by performing the imputation for families with both parental genotypes as if one or both parental genotypes were missing. If the imputation is unbiased, then the regression coefficient of the observed parental genotypes onto the imputed parental genotypes should be 1. This is because the covariance between the imputed parental genotypes and the observed parental genotypes should be equal to the variance of the imputed parental genotypes (Supplementary Note Section 2). Based on data from 31 families with two siblings and two parents genotyped, we obtained a regression coefficient of 0.9997 for regression of the sum of observed parental genotypes onto the imputed sum of parental genotypes. Based on data from 894 families with both parents genotyped, we set one parent's genotype as missing and imputed it from the remaining genotypes in the family, and we obtained a regression coefficient of 0.9989 for regression of the observed parent's genotype onto the imputed parent's genotype. These results show there is negligible bias in the imputed parental genotypes.

**Estimating direct and indirect effects of an education PGI.** We used summary statistics from a GWAS of EA[9] modified to remove the individuals in this study and their relatives, up to the third degree, from the summary statistics. We computed the PGI by applying LD-pred[47] to the summary statistics. We computed PGIs for individuals and their siblings and parents based on observed and imputed genotypes. We estimated the effects of the PGI by performing an LMM regression in snipar:

$$Y_{ij} = X_{ij}\theta + \mu_i + \epsilon_{ij}; \mu_i \sim N\left(0, \sigma_F^2\right); \epsilon_{ij} \sim N\left(0, \sigma_\epsilon^2\right);$$

where the columns of X were the intercept, the individual's PGI, the mean PGI of the siblings of the individual, the (imputed or observed) PGI of the individual's father, and the (imputed or observed) PGI of the individual's mother. Here, to account for the fact that the PGI might explain a substantial amount of phenotypic variance, the variance parameters $\sigma_F^2$ and $\sigma_\epsilon^2$ were estimated jointly with $\theta$. For the analysis assuming that the indirect effect from the sibling was zero, we dropped the sibling PGI from the regression and expanded the sample to include individuals with genotyped and/or imputed parents but without a genotyped sibling. We adjusted average NTC estimates for bias introduced by imputation when AM is present (Supplementary Note Section 8).

**GWASs in unrelated individuals.** We conducted GWASs using BOLT-LMM[48] in the sample of 'White British' UKB participants without third-degree or closer relatives also genotyped in the UKB[22]. This sample is therefore unrelated (less than third degree) from the sample of individuals with observed and/or imputed parental genotypes, who all have at least one first-degree relative also genotyped in the UKB. As in the related sample, we filtered out individuals identified by the UKB as having excess relatives, excess heterozygosity or sex chromosome aneuploidy, giving a sample size of 276,419. We applied BOLT-LMM to the 658,720 SNPs present on the UKB genotyping array.

In addition to the nine phenotypes used in the related sample, we also analyzed north and east birth coordinates. We adjusted the phenotypes for the same set of covariates as in our analysis of the related sample, including 40 genetic PCs. We estimated correlations between the summary statistics in the related sample and the summary statistics in the unrelated sample using the moment-based estimator (Supplementary Note Section 11) with the sampling correlation between the estimates set to zero.

**Reporting summary.** Further information on research design is available in the Nature Research Reporting Summary linked to this article.

## Data availability

Summary statistics for the direct effects, NTCs, and population effects of 1,586,010 SNPs on nine phenotypes can be downloaded from http://www.thessgac.org/data, subject to a terms of use to encourage responsible use of the data. Applications for access to the UKB data can be made on the UKB website (http://www.ukbiobank.ac.uk/register-apply/).

## Code availability

The code for IBD inference, imputation and genome-wide association and PGI analyses is available as a Python package, snipar[49], under an MIT license at

https://github.com/AlexTISYoung/snipar. Analyses were performed using Anaconda3 with Python 3.7.6 (https://repo.anaconda.com/archive/). Phasing was performed using SHAPEIT v2.r904 (https://mathgen.stats.ox.ac.uk/genetics_software/shapeit/shapeit.html), and haplotype merging was performed using QCTOOL v2.0.7 (https://www.well.ox.ac.uk/~gav/qctool_v2/). Relationship inference was performed using KING 2.2.4 (http://people.virginia.edu/~wc9c/KING). LD scores were computed using LDSC v1.0 (https://github.com/bulik/ldsc). Genome-wide association analyses in the sample of unrelated individuals were performed using BOLT v2.3.4 (https://alkesgroup.broadinstitute.org/BOLT-LMM/).

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

## Acknowledgements

The study was supported by funding from the Li Ka Shing Foundation, Open Philanthropy (010623-00001 and 2019-198171), the Netherlands Organisation for Scientific Research (VENI grant 016.Veni.198.058) and the National Institute on Aging/National Institutes of Health through grants R24-AG065184 and R01-AG042568 (to the University of California, Los Angeles), K99-AG062787-01 (to the Massachusetts General Hospital) and R00-AG062787 and R21-AG067585 (to the University of Southern California). We thank the UKB (application ID 11867).

## Author contributions

A.I.Y. and A.K. conceived and oversaw the study. A.I.Y. and A.K. derived the majority of the theoretical results, with contributions from S.M.N. and C.L. A.I.Y. developed the IBD inference algorithm. A.I.Y. and H.J. developed the method for computing correlations between effects. S.M.N. wrote the imputation code and developed the algorithm for imputation using more than two offspring. A.I.Y. wrote the code for genome-wide association and PGI analyses. A.O. and S.B. computed the EA PGI. A.I.Y. and S.M.N. analyzed the UKB data and performed the simulations. S.B., A.O., D.C., D.J.B. and P.T. provided important input and feedback on various aspects of the study design. All authors contributed to and critically reviewed the manuscript.

## Competing interests

The authors declare no competing interests.

## Additional information

**Extended data** is available for this paper at https://doi.org/10.1038/s41588-022-01085-0.

**Correspondence and requests for materials** should be addressed to Alexander I. Young or Augustine Kong.

a)

|  | $g_{i1}$ | $g_{i2}$ | $g_{p(i)}$ | $g_{m(i)}$ |
|---|---|---|---|---|
|  | 1 | 1/2 | 1/2 | 1/2 |
|  | 1/2 | 1 | 1/2 | 1/2 |
|  | 1/2 | 1/2 | 1 | 0 |
|  | 1/2 | 1/2 | 0 | 1 |

b)

|  | $g_{i1}$ | $\hat{g}_{i2}$ | $g_{p(i)}$ | $g_{m(i)}$ |
|---|---|---|---|---|
|  | 1 | 1/2 | 1/2 | 1/2 |
|  | 1/2 | 1/2 | 1/2 | 1/2 |
|  | 1/2 | 1/2 | 1 | 0 |
|  | 1/2 | 1/2 | 0 | 1 |

c)

|  | $g_{i1}$ | $\hat{g}_{i2}$ | $g_{p(i)}$ | $\hat{g}_{m(i)}$ |
|---|---|---|---|---|
|  | 1 | 1/2 | 1/2 | 1/2 |
|  | 1/2 | 3/8 | 1/2 | 1/4 |
|  | 1/2 | 1/2 | 1 | 0 |
|  | 1/2 | 1/4 | 0 | 1/2 |

d)

|  | $g_{i1}$ | $\hat{g}_{i2}$ | $\hat{g}_{p(i)}$ | $g_{m(i)}$ |
|---|---|---|---|---|
|  | 1 | 1/2 | 1/2 | 1/2 |
|  | 1/2 | 3/8 | 1/4 | 1/2 |
|  | 1/2 | 1/4 | 1/2 | 0 |
|  | 1/2 | 1/2 | 0 | 1 |

e)

|  | $g_{i1}$ | $g_{i2}$ | $\hat{g}_{p(i)}$ | $\hat{g}_{m(i)}$ |
|---|---|---|---|---|
|  | 1 | 1/2 | 1/2 | 1/2 |
|  | 1/2 | 1 | 1/2 | 1/2 |
|  | 1/2 | 1/2 | 3/8 | 3/8 |
|  | 1/2 | 1/2 | 3/8 | 3/8 |

f)

|  | $g_{i1}$ | $g_{i2}$ | $g_{p(i)}$ | $\hat{g}_{m(i)}$ |
|---|---|---|---|---|
|  | 1 | 1/2 | 1/2 | 1/2 |
|  | 1/2 | 1 | 1/2 | 1/2 |
|  | 1/2 | 1/2 | 1 | 0 |
|  | 1/2 | 1/2 | 0 | 3/4 |

g)

|  | $g_{i1}$ | $g_{i2}$ | $\hat{g}_{p(i)}$ | $g_{m(i)}$ |
|---|---|---|---|---|
|  | 1 | 1/2 | 1/2 | 1/2 |
|  | 1/2 | 1 | 1/2 | 1/2 |
|  | 1/2 | 1/2 | 3/4 | 0 |
|  | 1/2 | 1/2 | 0 | 1 |

h)

|  | $g_{i1}$ | $\hat{g}_{i2}$ | $\hat{g}_{p(i)}$ | $\hat{g}_{m(i)}$ |
|---|---|---|---|---|
|  | 1 | 1/2 | 1/2 | 1/2 |
|  | 1/2 | 1/4 | 1/4 | 1/4 |
|  | 1/2 | 1/4 | 1/4 | 1/4 |
|  | 1/2 | 1/4 | 1/4 | 1/4 |

i)

|  | $\hat{g}_{i1}$ | $g_{i2}$ | $g_{p(i)}$ | $g_{m(i)}$ |
|---|---|---|---|---|
|  | 1/2 | 1/2 | 1/2 | 1/2 |
|  | 1/2 | 1 | 1/2 | 1/2 |
|  | 1/2 | 1/2 | 1 | 0 |
|  | 1/2 | 1/2 | 0 | 1 |

j)

|  | $\hat{g}_{i1}$ | $g_{i2}$ | $g_{p(i)}$ | $\hat{g}_{m(i)}$ |
|---|---|---|---|---|
|  | 3/8 | 1/2 | 1/2 | 1/4 |
|  | 1/2 | 1 | 1/2 | 1/2 |
|  | 1/2 | 1/2 | 1 | 0 |
|  | 1/4 | 1/2 | 0 | 1/2 |

k)

|  | $\hat{g}_{i1}$ | $g_{i2}$ | $\hat{g}_{p(i)}$ | $g_{m(i)}$ |
|---|---|---|---|---|
|  | 3/8 | 1/2 | 1/4 | 1/2 |
|  | 1/2 | 1 | 1/2 | 1/2 |
|  | 1/4 | 1/2 | 1/2 | 0 |
|  | 1/2 | 1/2 | 0 | 1 |

l)

|  | $\hat{g}_{i1}$ | $g_{i2}$ | $\hat{g}_{p(i)}$ | $\hat{g}_{m(i)}$ |
|---|---|---|---|---|
|  | 1/4 | 1/2 | 1/4 | 1/4 |
|  | 1/2 | 1 | 1/2 | 1/2 |
|  | 1/4 | 1/2 | 1/4 | 1/4 |
|  | 1/4 | 1/2 | 1/4 | 1/4 |

m)

|  | $\hat{g}_{i1}$ | $\hat{g}_{i2}$ | $g_{p(i)}$ | $g_{m(i)}$ |
|---|---|---|---|---|
|  | 1/2 | 1/2 | 1/2 | 1/2 |
|  | 1/2 | 1/2 | 1/2 | 1/2 |
|  | 1/2 | 1/2 | 1 | 0 |
|  | 1/2 | 1/2 | 0 | 1 |

n)

|  | $\hat{g}_{i1}$ | $\hat{g}_{i2}$ | $g_{p(i)}$ | $\hat{g}_{m(i)}$ |
|---|---|---|---|---|
|  | 1/4 | 1/4 | 1/2 | 0 |
|  | 1/4 | 1/4 | 1/2 | 0 |
|  | 1/2 | 1/2 | 1 | 0 |
|  | 0 | 0 | 0 | 0 |

o)

|  | $\hat{g}_{i1}$ | $\hat{g}_{i2}$ | $\hat{g}_{p(i)}$ | $g_{m(i)}$ |
|---|---|---|---|---|
|  | 1/4 | 1/4 | 0 | 1/2 |
|  | 1/4 | 1/4 | 0 | 1/2 |
|  | 0 | 0 | 0 | 0 |
|  | 1/2 | 1/2 | 0 | 1 |

**Extended Data Fig. 1 | Variance-covariance matrices of observed and imputed genotypes.** Each matrix shows the variance-covariance matrix within the nuclear family given observed and imputed genotypes. The labels a) to o) correspond to those in Fig. 1. $g_{i1}$ denotes the proband's genotype, $g_{i2}$ denotes the proband's sibling's genotype, $g_{p(i)}$ denotes father's genotype and $g_{m(i)}$ denotes mother's genotype. A caret ('hat') over the 'g', such as $\hat{g}_{p(i)}$, indicates an unobserved genotype that is imputed (either linearly or non-linearly) using observed genotypes. Displayed are the variance-covariance matrices of the genotypes, normalized by the variance of an observed genotype, $2f(1-f)$ where $f$ is the allele frequency. This scaling means that the diagonal entry corresponding to an observed genotype is 1. Best linear unbiased imputations can be derived from applying formulae for multivariate Gaussian random-variables to a) (Supplementary Note Appendix D).

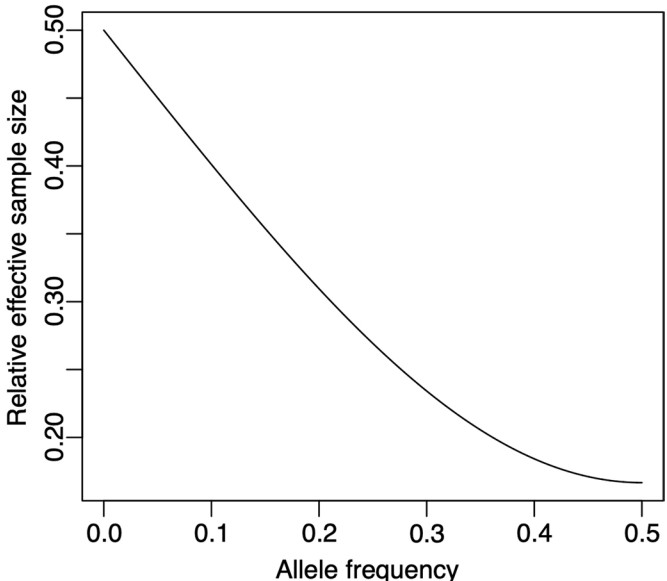

**Extended Data Fig. 2 | Relative effective sample size for estimating direct genetic effects when imputing a missing parent's genotype from a parent–offspring pair using unphased data.** We derive that the relative effective sample is $\frac{1-3f(1-f)}{2-2f(1-f)}$ compared to using fully observed parental genotypes (Supplementary Note Section 5.1). There is a penalty for heterozygosity when using unphased data since the allele that is shared with the observed parent cannot be determined when both parent and offspring are heterozygous. In contrast, the relative efficiency when using phased data is 1/2, independent of allele frequency.

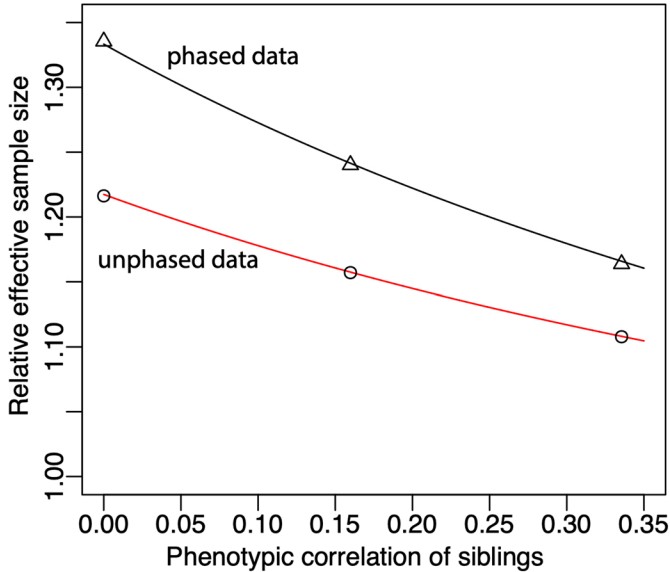

**Extended Data Fig. 3 | Confirmation of theoretical result for direct effects.** Here we compare theoretical predictions to results from simulated data for the effective sample size for estimation of direct genetic effects using parental genotypes imputed from both phased and unphased sibling genotype data relative to estimating direct genetic effects using genetic differences between siblings. For 3,000 independent families, we simulated three different traits affected by direct, paternal, and maternal effects of 1,000 SNPs with minor allele frequency 0.5 (Supplementary Note Section 12.1). The overall variance explained by the combined direct, paternal, and maternal effects varied between the simulations, leading to different correlations between siblings' phenotypes. We computed theoretical expectations based on formulae derived in Supplementary Note Section 4.3, which are drawn as the red curve (theoretical expectation for imputation from unphased data) and the black curve (theoretical expectation for imputation from phased data). The simulation results for unphased data are given by the black circles, and the simulation results for the phased data are given by black triangles. The relative effective sample size is given by the ratio between the sampling variance for estimation of direct effects when using differences between siblings to the sampling variance when using parental genotypes imputed from phased or unphased data.

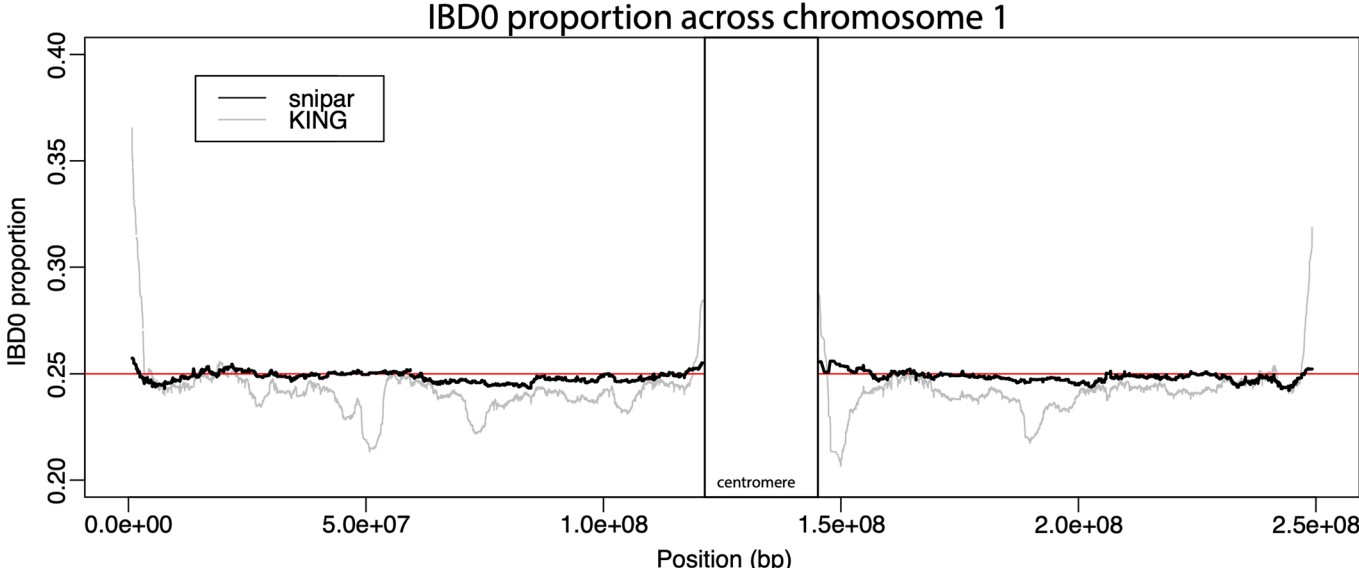

**Extended Data Fig. 4 | IBD0 proportion among sibling pairs across chromosome 1.** We show the proportion of sibling pairs, out of 19,290 pairs, that are called as IBD0 for each SNP with MAF>1% on chromosome 1 on the UK Biobank genotyping array. We compare the fraction of pairs that are called IBD0 by snipar (black line) and KING (gray line). The theoretical expectation according to Mendelian segregation is 0.25, indicated by the red horizontal line.

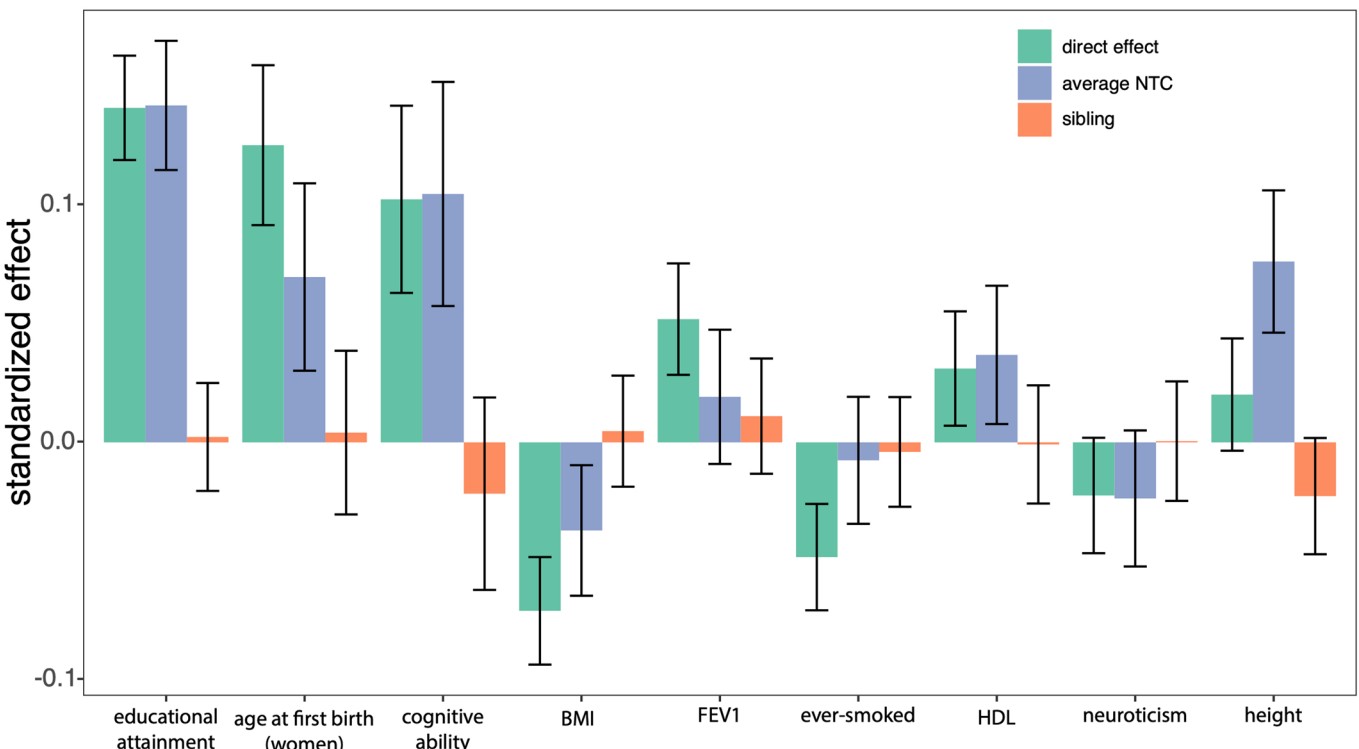

**Extended Data Fig. 5 | Indirect sibling effects of the EA polygenic index.** The standardized effect estimate (SD change in trait per SD change in PGI) is given along with the 95% confidence interval. Estimates were derived from a sample of 35,556 individuals from the 'White British' subsample of the UKB with at least one genotyped sibling and imputed and/or observed parental genotypes. Effects were estimated by joint regression of individuals' traits onto their own PGI, their siblings' PGI, and their mother's and father's (imputed/observed) PGI (Methods). We give the average of the maternal and paternal NTCs, adjusted for bias due to imputation in the presence of assortative mating (Methods), as the 'average NTC' here. Traits were adjusted for 40 genetic principal components prior to analysis. Trait abbreviations: HDL, high-density lipoprotein cholesterol; BMI, body mass index; FEV1, forced expiratory volume in one second.

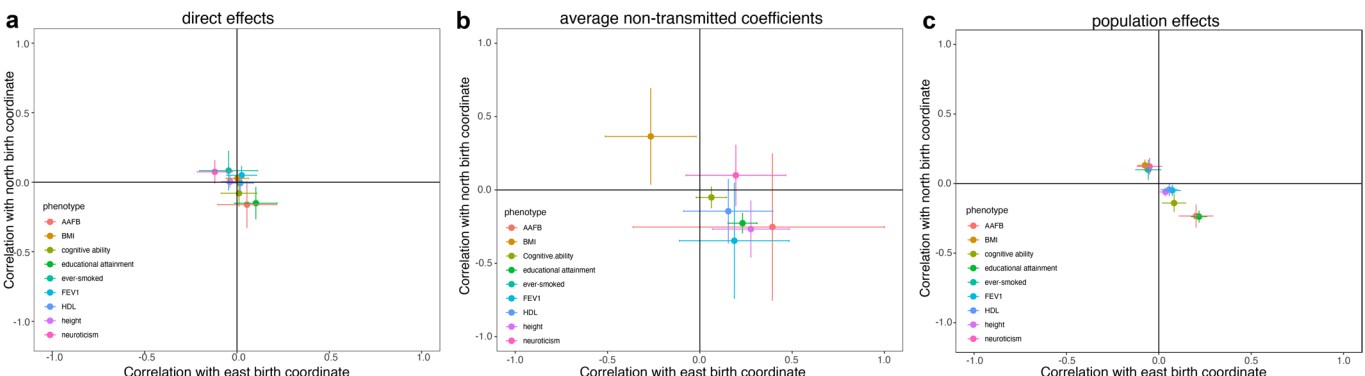

**Extended Data Fig. 6 | Correlations with north and east birth coordinates.** Associations between SNPs and north and east birth coordinates that persist after principal component adjustment were assessed by performing a genome-wide association study of north and east birth coordinates in a sample of unrelated individuals from the 'White British' subsample of the UK Biobank (Methods). We estimated genome-wide correlations between SNP associations with north and east birth coordinate and **a**) direct effects, **b**) average non-transmitted coefficients (NTCs), and **c**) population effects on 9 phenotypes (Methods). Point estimates are plotted as points, with error bars giving 95% confidence intervals. See Supplementary Table 6 for numerical results. Abbreviations: AAFB, age at first birth; BMI, body mass index; FEV1, forced expiratory volume in one second; HDL, high-density lipoprotein cholesterol.

# nature research

# Reporting Summary

Nature Research wishes to improve the reproducibility of the work that we publish. This form provides structure for consistency and transparency in reporting. For further information on Nature Research policies, see our Editorial Policies and the Editorial Policy Checklist.

## Statistics

For all statistical analyses, confirm that the following items are present in the figure legend, table legend, main text, or Methods section.

| n/a | Confirmed | |
|-----|-----------|---|
| ☐ | ☒ | The exact sample size ($n$) for each experimental group/condition, given as a discrete number and unit of measurement |
| ☒ | ☐ | A statement on whether measurements were taken from distinct samples or whether the same sample was measured repeatedly |
| ☐ | ☒ | The statistical test(s) used AND whether they are one- or two-sided<br>*Only common tests should be described solely by name; describe more complex techniques in the Methods section.* |
| ☐ | ☒ | A description of all covariates tested |
| ☐ | ☒ | A description of any assumptions or corrections, such as tests of normality and adjustment for multiple comparisons |
| ☐ | ☒ | A full description of the statistical parameters including central tendency (e.g. means) or other basic estimates (e.g. regression coefficient) AND variation (e.g. standard deviation) or associated estimates of uncertainty (e.g. confidence intervals) |
| ☐ | ☒ | For null hypothesis testing, the test statistic (e.g. $F$, $t$, $r$) with confidence intervals, effect sizes, degrees of freedom and $P$ value noted<br>*Give P values as exact values whenever suitable.* |
| ☒ | ☐ | For Bayesian analysis, information on the choice of priors and Markov chain Monte Carlo settings |
| ☒ | ☐ | For hierarchical and complex designs, identification of the appropriate level for tests and full reporting of outcomes |
| ☐ | ☒ | Estimates of effect sizes (e.g. Cohen's $d$, Pearson's $r$), indicating how they were calculated |

*Our web collection on statistics for biologists contains articles on many of the points above.*

## Software and code

Policy information about availability of computer code

| | |
|---|---|
| Data collection | No new data was collected in this study, so no software was used for data collection. |
| Data analysis | We used snipar v0.0.9 (https://doi.org/10.5281/zenodo.6484858) for IBD inference, imputation, genome-wide association and polygenic index analyses, and estimation of genome-wide correlations between effects. snipar is available as a Python package under an MIT license at https://github.com/AlexTISYoung/snipar. Analyses were performed using Anaconda3 with Python 3.7.6 (https://repo.anaconda.com/archive/). Phasing was performed using SHAPEIT v2.r904 (https://mathgen.stats.ox.ac.uk/genetics_software/shapeit/shapeit.html), and haplotype merging was performed using QCTOOL v2.0.7 (https://www.well.ox.ac.uk/~gav/qctool_v2/). Relationship inference was performed using KING 2.2.4 (http://people.virginia.edu/~wc9c/KING). LD scores were computed using LDSC v1.0 (https://github.com/bulik/ldsc). Genome-wide association analyses in the sample of unrelated individuals was performed using BOLT v2.3.4 (https://alkesgroup.broadinstitute.org/BOLT-LMM/). |

For manuscripts utilizing custom algorithms or software that are central to the research but not yet described in published literature, software must be made available to editors and reviewers. We strongly encourage code deposition in a community repository (e.g. GitHub). See the Nature Research guidelines for submitting code & software for further information.

## Data

Policy information about availability of data

All manuscripts must include a data availability statement. This statement should provide the following information, where applicable:
- Accession codes, unique identifiers, or web links for publicly available datasets
- A list of figures that have associated raw data
- A description of any restrictions on data availability

Summary statistics for the direct effects, NTCs, and population effects of 1,586,010 SNPs on 9 phenotypes can be downloaded from http://www.thessgac.org/data,

# Field-specific reporting

Please select the one below that is the best fit for your research. If you are not sure, read the appropriate sections before making your selection.

☒ Life sciences ☐ Behavioural & social sciences ☐ Ecological, evolutionary & environmental sciences

For a reference copy of the document with all sections, see nature.com/documents/nr-reporting-summary-flat.pdf

# Life sciences study design

All studies must disclose on these points even when the disclosure is negative.

| | |
|---|---|
| Sample size | We used all genotyped individuals in UK Biobank passing ancestry and quality filters with at least one genotyped first-degree relative: 39,619. |
| Data exclusions | In order to make our analyses comparable to existing genome-wide association studies, which generally use a genetically homogeneous population, we restricted our sample to individuals determined by UK Biobank to have predominantly White British ancestry. We filtered out individuals identified by UK Biobank as having excess relatives, excess heterozygosity, or sex chromosome aneuploidy. |
| Replication | Replication is not applicable to this study since we are describing a novel methodology. We demonstrate the methodology with application to UK Biobank data, and the results presented are valid for that dataset. Different results may be obtained by application of this methodology to different datasets/populations. |
| Randomization | Genetic material is randomized during meiosis, which we use to obtain estimates of direct genetic effects, i.e. to remove the influence of confounding factors. |
| Blinding | Blinding is not applicable since we did not compare different experimental groups. |

# Reporting for specific materials, systems and methods

We require information from authors about some types of materials, experimental systems and methods used in many studies. Here, indicate whether each material, system or method listed is relevant to your study. If you are not sure if a list item applies to your research, read the appropriate section before selecting a response.

## Materials & experimental systems

| n/a | Involved in the study |
|---|---|
| ☒ ☐ | Antibodies |
| ☒ ☐ | Eukaryotic cell lines |
| ☒ ☐ | Palaeontology and archaeology |
| ☒ ☐ | Animals and other organisms |
| ☒ ☐ | Human research participants |
| ☒ ☐ | Clinical data |
| ☒ ☐ | Dual use research of concern |

## Methods

| n/a | Involved in the study |
|---|---|
| ☒ ☐ | ChIP-seq |
| ☒ ☐ | Flow cytometry |
| ☒ ☐ | MRI-based neuroimaging |

