## [Peer Review File · Nature Genetics]

Peer Review Information

Manuscript Title: Mendelian imputation of parental genotypes improves estimates of direct genetic effects

Corresponding author name(s): Dr Alexander Young

Reviewer Comments & Decisions:

Decision Letter, initial version:
--

9th Mar 2021

Dear Dr. Young,

Your Technical Report, "Mendelian imputation of parental genotypes for estimation of direct and indirect genetic effects" has now been seen by 2 referees. You will see from their comments copied below that while they find your work of considerable potential interest, they have raised quite substantial concerns that must be addressed. In light of these comments, we cannot accept the manuscript for publication, but would be very interested in considering a revised version that addresses these serious concerns.

We hope you will find the referees' comments useful as you decide how to proceed. If you wish to submit a substantially revised manuscript, please bear in mind that we will be reluctant to approach the referees again in the absence of major revisions.

To guide the scope of the revisions, the editors discuss the referee reports in detail within the team, with a view to identifying key priorities that should be addressed in revision. As you will see from these comments, reviewer #1 thinks that there is much emphasis on the imputation part but little comment on the technical advance of the new method or comparisons with other methods. Reviewer #2 has a few concerns regarding the terminology and the interpretation of some of the results. We hope that you will find the prioritised set of referee points to be useful when revising your study.

If you choose to revise your manuscript taking into account all reviewer and editor comments, please highlight all changes in the manuscript text file. At this stage we will need you to upload a copy of the manuscript in MS Word .docx or similar editable format.

*2) If you have not done so already please begin to revise your manuscript so that it conforms to our Technical Report format instructions, available [here](http://www.nature.com/ng/authors/article_types/index.html). Refer also to any guidelines provided in this letter.

[REDACTED]

If you wish to submit a suitably revised manuscript we would hope to receive it within 6 months. If you cannot send it within this time, please let us know. We will be happy to consider your revision so long as nothing similar has been accepted for publication at Nature Genetics or published elsewhere. Should your manuscript be substantially delayed without notifying us in advance and your article is eventually published, the received date would be that of the revised, not the original, version.

Nature Genetics is committed to improving transparency in authorship. As part of our efforts in this direction, we are now requesting that all authors identified as 'corresponding author' on published papers create and link their Open Researcher and Contributor Identifier (ORCID) with their account on the Manuscript Tracking System (MTS), prior to acceptance. ORCID helps the scientific community achieve unambiguous attribution of all scholarly contributions. You can create and link your ORCID from the home page of the MTS by clicking on 'Modify my Springer Nature account'. For more information please visit please visit

<http://www.springernature.com/orcid>>www.springernature.com/orcid.

Thank you for the opportunity to review your work.

Sincerely,

Wei

Wei Li, PhD
Senior Editor
Nature Genetics
1 New York Plaza, 47th Fl.
New York, NY 10004, USA
www.nature.com/ng

Reviewers' Comments:

Reviewer #1:

Remarks to the Author:

This paper is a timely contribution to the reconciliation of genotype-phenotype associations estimated from population-based GWAS samples and within-family effects. It is exciting to see the return of families after recent emphasis on the study of 'unrelated' individuals.

The paper makes conducts three main analyses (1) imputation of parental and/or sibling genotypes based on nuclear families, (2) deconstruction of population-based GWAS estimates of SNP effects for 9 traits in the UKB into direct and indirect (sibling, maternal, paternal) effects, and (3) estimation of the direct and indirect effects of an education polygenic score on 9 traits in the UKB. The authors estimate a low correlation between GWAS and direct effects for EA, cognitive ability and HDL cholesterol. They also find EA polygenic score to have a significant direct effect on all traits (EA, age at first birth, cognitive ability, BMI, ever-smoked, FEV1, HDL, neuroticism, height) and indirect effect on all traits except ever-smoked and neuroticism.

There is much emphasis in the paper on the imputation part of the manuscript, but little comment on the advancement that this new method brings. There is extensive (slightly overwhelming) literature on imputation, and some related to the genotype imputation of ungenotyped individuals with pedigree links – mostly from livestock. Currently there is no comparison of the new method to other methods, e.g. F-impute and AlphaImpute. As I understand these programs use a rules-based approach and segregation analysis. Is this – in your terminology – equivalent to linear imputation or something else? Other authors have used linear models in complex pedigrees, e.g. Gengler, Mayeres & Szydlowski (2007) Animal and related references. Further, programs such as F-impute (and possibly beagle) use both pedigree and population-based information for imputation of SNP genotypes for ungenotyped individuals. As far as I can see, the method proposed by the authors does not utilise population-based haplotype information for imputation (as it resorts to the allele frequency in the population). Is the current 'IBD' method equivalent to a long-range phasing / haplotype approach? Accuracy of

imputation increases with the number of relatives in a family but there are no clear take-away messages regarding data structure. Greater context – in the introduction, up front – for this aspect of the study would help. The authors should give a clearer examples of when their method is superior, possible modification of Fig 3 to show the case with and without IBD information. The paper needs to better articulate the novelty of this approach, given many of the other approaches already developed for imputation. The relevant literature suggests that either population information (short haplotypes) or information on other close (genotyped) relatives (e.g. uncles, or cousins) may also be useful.

The subsequent analysis are novel, building on the authors previous publications in this area. My main concern is the high degree of collinearity between the sources of information (proband, parental, sibling) combined with a relatively small sample size. If there are four main effects (direct, sibling, maternal, paternal), there are 10 potential pairwise interactions? The authors take a number of steps throughout the manuscript to fix or remove certain parameters, maybe because trying to estimate all these correlated effects simultaneously results in high uncertainty. The methods are not clear to me, p. 20 for example, how are the SNP effects calculated. They seem to be estimated from GBLUP – what are the estimates of the variance components? The authors assume knowledge of LD score regression on p. 21, it should contain a brief explanation.

Is Supplementary Figure 4 the genetic correlation between direct and indirect-parental effects? There is not much comment on these results – are these 95% CI? There seems a significant negative correlation for cognitive ability, and positive correlation for height. Under what conditions can a negative (or positive) correlation be generated between a probands direct genetic value and its mid-parent value? The expectation should be zero? Can the authors clarify.

The main findings from the results, Fig 6, suggest a low correlation between the direct and population effects for education, cognitive ability and HDL. However, we want to know the correlation between the population effects and the true (unobserved) direct effects. Is this what was done in simulation, or the correlation between the inferred direct effects, it is not clear. Can the authors show that their method produces unbiased estimates of the direct effects? Is Fig. 6 the correlation between population effect (direct + avg. parental) and direct from this population? A more useful parameter would be the correlation with the population effect from the latest GWAS using unrelated individuals. Is the population effect bias from indirect effects decreased with increasing sample size?

Supplementary Note Table 8 makes it clearer as to which type of parameters can be estimated from different types of data and where the information is coming from. This is presented in the context of a meta-analysis but may be more useful (in some form) in the main text. It would make interpretation of the UKB results easier, as it is clear that there are few trios or parent-offspring pairs, thus little power to distinguish paternal / maternal effects.

Detailed comments:

Why are 'assortative mating' effects classified as confounding but not real/causal effects? (introduction, line 10 para 2). Assortative mating creates gametic phase disequilibrium, i.e. correlations between trait-increasing alleles across chromosomes. It is a real effect, and generates real genetic variance. If the analysis is one SNP at a time then the effects of the correlated SNP are captured but it is still a genetic effect. Traits under assortative mating, such as height or EA, have higher (true) genetic variance in the current population than that under random mating. I find the 'linear' and 'non-linear' description of imputation distracting. Non-linear means using haplotype information? – non-linear implies (to me) imputation that does not follow the normal rules

of inheritance.

There are well documented effects for assortative mating for human height, and yet the results in Fig 6 suggest a strong correlation between direct and population effects?

Why the low correlation for HDL? (Fig 6).

What is the frequency of the 15 different missing information imputation patterns in the UKB? A straight statement of these cases/number of families would be useful (statements on pg 11 are unclear). Most (>88%) are case e. Are the configurations for the ungenotyped proband (last 2 lines Fig 1) relevant? Top of page 11 sounds like only analysed sibling pairs but I don't think this was the case. The results should state the sample size for each analysis and be clearer as to what type of information is contributing to each estimate.

There seem missing co-efficients in the eq. on the bottom of page 7.

There are many places throughout the manuscript and supplementary material where the terms are not defined where they should be. e.g. 'PGS' (pg. 1, line 3 after introduction), 'f' and 'X' legend to Supplementary Figure 1 (both) and 2 ('f' only), η_s and δ in legend of Supplementary Figure 3, etc. Not an exhaustive list.

Ref [2] is incorrect. Should be Meuwissen, Hayes & Goddard.

Its seems that there is also no significant effect of the parental education polygenic score on FEV1 also (p. 14 Figure 7).

I'm not sure that being deceased is a precondition for multi-point linkage analysis (p. 14, discussion).

Methods: UKB Sample. No detail on number of SNP, MAF, chip etc etc for UKB. How did they phase their genotypes? What do you mean by 'family based gwas' – not explained. sample size? Did the mean and variance in IBD sharing match expectations for sib-pairs?

Reviewer #2:

Remarks to the Author:

This paper has two goals. The first is to show how genomic data of the nuclear family (sibs and parents) can be imputed using IBD information in a way that causes no biases in downstream analyses, even when the vast majority of the data is missing. This is important for the second goal, which is to estimate direct and indirect genetic influences in large biobanks (here, the UKB) that are not oversampled for close relatives. The authors demonstrate that their imputation method (instantiated in software SNIPar) produces unbiased estimates of these influences and that (perhaps more surprisingly) those estimates have the correct standard errors. By applying the proposed method to an educational attainment (EA) and cognitive ability (which I'll call "IQ") PGS, the authors observed a lower direct/(direct+indirect) effect ratio than was observed in their previous study with Iceland cohort. To explain these differences, the authors claimed that the combined influence of indirect effects, population stratification, and assortative mating is stronger in the UK than in Iceland.

This is an excellent idea and represents a tremendous amount of work on the authors' part. However, the manuscript really needs work in multiple areas. The three biggest problems are, first, parts of it are overly confusing, use poor terminology, and are inconsistent. More work needs to be put into clarity. Second, the surprisingly large and potentially important result that most of the population effect of EA and IQ are not direct, and that this may be due to stratification, is not adequately explored. Third, there are a lot of different influences on the results (e.g., assortative mating alone influences Mendelian imputation, the GWAS estimates, LDSC estimates, and the correlation between transmitted and non-transmitted PGSs). These various factors influence the results the authors

describe in complicated and sometimes competing ways. It would be nice to see a simulation that shows that the method works as advertised under simplifying scenarios, and to understand how big the biases can be under other scenarios in order to better gauge the relative importance of these factors. I describe these and other points below as they arose in the manuscript.

p.3 - alpha's referred to as "parental effects" is poor terminology, misleading, and confusing. Words matter. First, the caveat the authors provide after describing their choice of terminology could be easily forgotten or missed. Second, and more importantly, the alphas represent only those influences of parental genes mediated via the parental phenotype. Unless the trait is completely heritable (in which case there would be no genetic nurture), there must be additional parental influences that are environmental in nature and thus going into the epsilon terms in the model. Thus, "parental effects" as used by the authors are only a potentially small portion of the actual parental phenotype -> offspring phenotype influences. Part of the problem in my view is that the models used by the authors don't make this explicit (e.g., these parental phenotype -> offspring phenotype influences, as well as the sib -> sib influences, are buried in the epsilon term). At the least, I'd suggest authors describe "alpha" differently. They used to call it "genetic nurture," and while I can see that this is inflated by AM and its estimate can be biased by stratification, "genetic nurture" or "parental indirect genetic effects" or "residual parent-offspring genetic association" aren't great, but they all seem much preferable to the misleading "parental influences" term.

p.3 - when first introduced, authors should explain what "non-linear imputation" is and why it is relevant.

A few suggestions for interpretability of Figure 2:

- would be easier to understand if "?" replaced the values of the parental genotypes that are unknown, given that the values provided are just arbitrary numerical examples.
- at top, write "Stretches where siblings are IBD0", "Stretches where siblings are IBD1", etc.
- rewrite last sentence a bit. "...and IBD segments can be determined by finding segments of chromosomes where siblings' genotypes always have at least one allele the same (are IBS1 or IBS2), in which case the segment is IBD1, or always have both alleles the same (IBS2), in which case the segment is IBD2".

p. 6 - "the squared correlation between imputed and observed parental genotype is $3/4$, higher than based on best linear unbiased imputation (Supp Fig 1)." I think I see why ($1/4*1 + 1/2*3/4 + 1/4*1/2 = 3/4$), but this need to be made clearer, and the reference to Supp Figure 1 is not helpful - which subfigure? Which part of the subfigure shows this? I was looking at 1e but $3/4$ is not in there anywhere.

p.8 and Supp. p. 13, Regarding $Y1-Y2$ being orthogonal to $Y1+Y2$. These won't be orthogonal if $Y1$ and $Y2$ have different variances, such as might occur, for example, if $Y1$ is always the male phenotype and $Y2$ always the female phenotype.

p. 8 - "Letting r be the correlation of the siblings' residuals, which will be approximately equal to their phenotypic correlation for polygenic traits..." I assumed the authors meant the epsilon component, but this can't be right. The epsilon (error) components (which are independent of genes or other direct effects) may be correlated (e.g., due to neighborhoods, friends, schools, etc), but they may not be either. To say that they will be approximately equal to their phenotypic correlation is either wrong, or I'm misunderstanding what the "siblings' residual" is. Can authors clarify?

p.11 - $r_{(\Delta \beta)}$ – the authors state that this is the correlation between direct and parental effects. At first I interpreted this as “passive G-E correlation”, because that’s what the authors’ terminology suggests, but then remembered that they are calling β (which is α when $\eta_s=0$... geez) “parental effects” when it is really indirect genetic effects/genetic nurture. So then what is the interpretation of $r_{(\Delta \beta)}$? The correlation between direct effects and genetic nurture effects? And Figure 5 shows several different lines for $r_{(\Delta \beta)}$, but I’d imagine $r_{(\Delta \beta)}$ is itself a function of h^2_{β} , meaning that some parts of the lines may be impossible or unlikely. And how does assortative mating and/or stratification influence these quantities of $r_{(\Delta \beta)}$ and h^2_{β} ? Much of my confusion here could have been cleared up if the derivations of them were somewhere, but I can’t find them in the main text, Methods, or Supp. My apologies if I’ve missed it, but if they indeed are not shown, they should be somewhere.

p. 13 (top) & Supp Fig 4 – Again, I’m finding the notation confusing and inconsistent. The main text discusses $r_{(\Delta \beta)}$ and refers to Supp Figure 4, but as far as I can tell, Supp Fig 4 plots $r_{(\Delta \eta)}$. So let me see if I have this right. β is a downward estimate of the “parental effect” (i.e., genetic nurture), which equals α when η_s (the sibling indirect genetic effect)= 0. And η [with no subscript] is...?? Genetic nurture I take it? This all seems to need some work for consistency/clarity, and the authors need to have a table in the supplement with all the notations and their interpretations because it becomes mentally taxing to try to parse this, especially when the notations isn’t consistent between the text and the Supp. And main text should not refer to a figure that in turn refers to a table in the Supp that refers to quantities ($r_{(\Delta \eta)}$) never discussed in the main text.

Fig 6 – the EA/cog ability results seem awfully low, but even odder, why is the $r_{(\Delta a)}$ for height almost 1? Shouldn’t assortative mating (ASM) reduce this? It would be nice to have what the expectations of $r_{(\Delta a)}$ are for observed levels of ASM.

Also, the LDSC estimates in Fig 6 rely upon an inherent assumption of no long-range LD (i.e., the regional LD score is all that’s needed to capture the total association due to a causal variant). This is not true under ASM. Can the authors comment on how ASM might influence their LD score regression estimates?

p. 13 – the para starting “In practice...” needs work for clarity. As I understand it, there are two sources of inflation of the “ η ” (or β or α !) terms that arise from ASM. One is due to imputation: imputed parental PGS’s don’t capture the inflation of variance due to ASM (the expectation that alleles unshared between parent and offspring are drawn at random from the population is incorrect). The second is that ASM induces a correlation between the nontransmitted PGS and the transmitted PGS. Is all this correct? I felt it needed to be spelled out more clearly, and also that the relative sizes of these biases needed to be discussed. I’m guessing the second is larger. The paragraph also seems out of place. Why not describe the results first and then caveats? Finally, if we have a closed-form solution for the bias that depends only on r (the mate correlation), the estimates should be able to be back-corrected (given that we know what r is in the UKB) so that they are unbiased by ASM, right? Why not do that? I suppose the answer is because we don’t know whether equilibrium has been reached? But by comparing PGS’s across mates to those across haplotypes within person (ala the Kong Science paper), we can know this, no?

p. 13 - In that same paragraph: "Importantly, the inflation of parental effect estimates is smaller than when estimating parental effects without performing imputation. " - unclear. You mean when

estimating parental effects using PGS from nonimputed SNPs?

p. 14 - The same influences of stratification/ASM/genetic nurture are already baked into the GWAS estimates that the authors use to create the polygenic scores. Can the authors comment on how those influences affect the estimates shown in Figure 7?

p. 14 - 15 - The "parental influences" (i.e., parental indirect genetic effects) in Figure 7 for EA and IQ seem too high to have arisen from parental phenotype -> offspring phenotype effects. I see the authors make this same point on p. 15 but their treatment of this important issue is superficial. A starting place could be that, if we were to interpret this as being solely due to genetic nurture (rather than ASM or stratification), what kind of parental phenotype -> offspring phenotype effect would that imply? Balbona et al (bioarxiv) discuss how to convert estimates of genetic nurture to total parental environmental influences given the r^2 of the polygenic score. Does that comport at all with previous estimate of parental influences from extended twin methods (e.g., Devlin, Daniels, Roeder, 1997)? And how much of this can be due to ASM given known spousal correlations? The authors state that stratification may explain these findings. If this is true, it is extremely important, and therefore the authors really need to discuss this possibility more thoroughly. Why do they believe this? What evidence supports that view? Why wouldn't controlling for 10 or 20 PCs within a European-restricted sample eliminate this? (BTW, I cannot find where they discuss how many PCs were controlled for).

If there is residual stratification remaining in the data for EA or IQ, such that there are mean differences in phenotype across genetic clusters, doesn't this also lead to long-range directional LD in the same way that ASM does? If so, doesn't this lead to many of the same kinds of biases that ASM does? Can the authors discuss this?

Are there any complete genotyped trio families (including probands, and their parents) in UKB? If there are, can the author compare the genetic analysis results from complete trio in UKB to those of Iceland consortium? Alternatively, can the authors mask the Icelandic data to mimic the UKB missingness patterns and see if results change and come more in line with the UKB data? If so, maybe there are some missed confounding effects in the current Mendelian Imputation method.

In Figure 7, most of estimations of indirect genetic effects from siblings (τ) are around 0. Thus, maybe displaying the estimates without τ (providing less noisy results) would be preferable and then place the figure with τ , and differentiating father vs. mother effects, in the Supp.

Supp, p. 1 - if authors intend for the supplement to be able to be read stand-alone (which it appears to be the case, and I think is wise), then they need to introduce each new term as it comes. E.g., β is introduced on p.1 without explanation. And it's not clear why authors use η_p and η_m in equations 1 and 2 but these are never used in the main text (they are β 's and α 's there). I take it that η_p in the supp is the same as η_p in an earlier Science paper by these authors while α_p in the main text encompasses both η_p plus an effect due to stratification and/or assortative mating. But this needs to be clearer.

Supp p. 26+ - what does the "I" stand for in "PGI"? Imputed? Is another term really necessary? I thought \hat{PGS} was the imputed PGS.

The authors seek to estimate gene-environment correlation and heritability without bias from imputed genotypes. They provided some proofs of unbiasedness of their methods in the Supp and it seems

correct. However, it is hard to understand how their imputed methods could increase the power with their own derivation in Supp. section 2.3. In their equation (30), if the variance of imputed genotype is much smaller than that of observed genotypes, the variances of estimators would be larger. Thus, the authors should provide results (e.g., from simulation) comparing the variances of estimators across different missingness structures.

The parameter r in section 8 of supplementary notes seems confused with those of section 4. It would be better to change one of those parameter notations to reduce confusion.

In supplementary note pages 14 and 15, the authors described the variances of estimators with different type of methods. It is hard to determine which method has smaller variances. Since $-1 < r < 1$, no imputation method has larger variances of estimators than those of IBD and phased method, overall. I think that authors need to mention that IBD imputation method always gives more efficient estimators than unimputed method.

Author Rebuttal to Initial comments

Summary of changes

We thank both reviewers for their perspicacious remarks on the manuscript, which we have made our best efforts to fully address, and in doing so, have greatly improved the readability, impact, and rigor of the manuscript. We summarize the main changes we have made here, giving detailed responses to the reviewers' points below. Beyond the substantive changes described here, we have made some changes to the presentation of the material to make it more concise and to flow better. Due to the fact that the manuscript describes substantial novel theory, methods, software, and empirical results, there are necessarily going to be constraints on what we have the space to detail and discuss in the main text. Therefore, some material is only discussed briefly in the main text, with more details in the methods, supplementary note, and supplementary tables.

Terminology

We thank the reviewers for highlighting the confusing terminology in the original submission. We have attempted to create a terminology that is as simple as possible and does not suggest incorrect interpretations. We now express everything in terms of the parameters of model (1):

$$\begin{aligned} Y_{i1} &= \delta g_{i1} + \alpha_p g_{p(i)} + \alpha_m g_{m(i)} + \epsilon_{i1}; \\ Y_{i2} &= \delta g_{i2} + \alpha_p g_{p(i)} + \alpha_m g_{m(i)} + \epsilon_{i2}; \end{aligned} \quad (1)$$

plus the indirect effect from a sibling, η_s , meaning that model (2) is now expressed as

$$Y_{i1} = \delta g_{i1} + \eta_s g_{i2} + (\alpha_p - \eta_s/2)g_{p(i)} + (\alpha_m - \eta_s/2)g_{m(i)} + \epsilon_{i1};$$

$$Y_{i2} = \delta g_{i2} + \eta_s g_{i1} + (\alpha_p - \eta_s/2)g_{p(i)} + (\alpha_m - \eta_s/2)g_{m(i)} + \epsilon_{i2};$$

This avoids the confusion of talking about α 's and β 's. We now always refer to the α 's of model (1), which are always defined, whether any siblings are present or not.

Reviewer (2) highlighted that referring to the α parameters as 'parental' effects could be misleading. We agree with the reviewer, but we prefer not to change the terminology to something that implies the α parameters are necessarily reflecting indirect genetic effects from parents, because this is often not the case. We have decided to adopt a more neutral term, 'non-transmitted coefficient', which we abbreviate as 'NTC'. This is because the α_p and α_m parameters are the expected coefficients on the non-transmitted paternal and maternal alleles from a joint regression of offspring phenotype onto offspring genotype and non-transmitted parental alleles; and they can reflect confounding due to assortative mating and population stratification, in addition to indirect genetic effects, which we abbreviate to IGEs. This enables us to distinguish more clearly between situations where the NTCs reflect parental IGEs, and when they do not.

We have made sure that our use of terminology is consistent between the main text, methods, and supplementary note, and we have added a table of terms to the start of the Supplementary Note, where we also introduce the main models.

Comparison to other imputation methods

Reviewer #1 highlighted that we needed to situate our imputation method in relation to existing imputation methods. Reviewer 1 suggested AlphaImpute and F-impute as points of comparison. We attempted to run these imputation methods on simulated data on sets of independent siblings, but these methods produced mainly missing calls for the genotypes of the ungenotyped parents, probably because they are designed for large, complex pedigrees, rather than isolated nuclear families. We have added the following paragraph (with citations to AlphaImpute and F-impute) to the introduction to make clear the similarity to previous methods in animal breeding, and the different type of data structure our method is designed for compared to AlphaImpute and F-Impute:

“Our imputation approach is similar to methods used in animal breeding to impute ungenotyped members of a pedigree^{20–22}, which are typically designed for breeding applications in large, complex pedigrees. In contrast, our method is designed for estimation of direct effects from sets of nuclear families that are not closely related to one another, the kind of family data typical in human genetics.”

We found a recent method from the authors of AlphaImpute, AlphaFamImpute (Whalen, A., Gorjanc, G. & Hickey, J. M. AlphaFamImpute: High-accuracy imputation in full-sib families from genotype-by-sequencing data. *Bioinformatics* 36, 4369–4371 (2020).), that is designed for a similar data structure to our method: sets of nuclear families with one or both parents' genotypes missing and one or more sibling genotyped. We now compare our imputation method to AlphaFamImpute. We compare the performance of our method applied to phased and unphased data in Supplementary Table 8. We include a copy of the results here:

Data Type	AlphaFamImpute		SNIPar (unphased)		SNIPar (phased)	
	R^2	Coefficient	R^2	Coefficient	R^2	Coefficient
Sibling pair (Fig 1e)	0.6986	0.8596	0.7167	0.9994	0.7484	0.9992
Parent-offspring pair (Fig 1c,1d)	0.3743	1.0193	0.3742	0.9986	0.4999	0.9997
Parent and sibling pair (Fig 1f,1g)	0.6375	0.8897	0.6148	1.0002	0.7508	0.9998

Supplementary Table 8. Comparison of SNIPar and AlphaFamImpute. We give results from applying AlphaFamImpute²² and SNIPar to simulated data on 3,000 independent families for three different data types (Methods): siblings pairs without genotyped parents, parent-offspring pairs, and a parent with two full-sibling offspring. For imputation from a sibling pair, we compare the imputed parental genotype to the sum of the missing parents' genotypes; and for the other data types, we compare the imputed parental genotype to the missing parent's genotype. We give the squared correlation, R^2 , between the imputed and actual parental genotypes, and the coefficient from regression of actual parental genotypes onto imputed parental genotypes, which is equal to 1 when the imputation is unbiased. We show results from applying SNIPar to both unphased genotypes, and to phased haplotypes.

We found that AlphaFamImpute performs similarly to our method applied to unphased data in terms of R^2 between imputed and actual parental genotypes, but that the imputed genotypes produced by AlphaFamImpute were biased, whereas the imputed genotypes produced by our method were unbiased. When using phased data, our method improves on AlphaFamImpute in terms of R^2 between imputed and actual parental genotypes, and is unbiased. Note that for our application, bias in imputed parental genotypes would translate into biased estimates of direct effects and NTCs, making the imputed genotypes produced by AlphaFamImpute of limited utility.

We have improved our discussion of how our imputation method compares to other imputation methods and the kind of data it is best suited for. The first paragraph of the discussion now reads:

We introduce Mendelian Imputation as a tool to perform genetic association analyses. Conceptually, this is similar to multipoint linkage analysis performed with pedigrees³³, familial imputations^{34,35}, and methods in animal breeding²⁰. Our approach focuses on imputing missing

parental genotypes in a nuclear family, rather than in the large pedigrees typical in animal breeding. We found that our imputation method improves on a recent method developed for imputing missing genotypes in a nuclear family, AlphaFamImpute²², both in terms of bias and R^2 between imputed and actual parental genotypes (Methods and Supplementary Table 8). The improvement in R^2 derives from the use of pre-phased genotypes, enabling resolution of which parental alleles have been observed in certain doubly heterozygous cases. Our approach could be extended to use information on other relatives of the missing parent(s), such as siblings. However, many human genetic datasets contain genotype data on sibling pairs and/or parent-offspring pairs from families that have no known pedigree relation to one another, and for these data sets, our approach provides close to optimal recovery of the genotypes of the missing parent(s).

We note that, while our imputation method does improve on existing methods for similar data structures, we do not see the imputation in isolation as the key innovation in our manuscript; rather, we see the key contribution of the manuscript to be a new way of performing GWAS that treats the nuclear family as the fundamental unit of analysis. This perspective brings with it a new workflow for analysing human genetic data (with associated software package SNIPar): to start from a model that includes proband and parental genotypes, to impute missing parental genotypes from observed genotypes/haplotypes in the family (using inferred IBD segments between siblings), to estimate direct effects and non-transmitted coefficients, and potentially to meta-analyse the summary statistics with summary statistics from other samples and data types. This workflow maximizes power and model identifiability, and separates direct, causal, genetic effects from other factors contributing to genotype-phenotype associations.

Expanded SNP set

For the empirical analysis of direct effects, NTCs, and population effects of genome-wide SNPs, we have expanded the set of SNPs we use to include 1.1 million HapMap3 SNPs with MAF>1%, in addition to the SNPs with MAF>1% on the UK Biobank genotyping arrays. We describe this in the updated Methods section 'UK Biobank Sample':

Haplotypes for the SNPs that were present on both the UKB Axiom and the UK BiLEVE genotyping arrays and that passed quality control were provided by UK Biobank (UKB)²⁸. Phasing was performed using SHAPEIT3⁴⁰ and the 1000 Genomes Phase 3 dataset⁴¹ as a

reference panel. This resulted in phased haplotypes for a set of 658,720 autosomal SNPs with an estimated switch error rate of 0.229%²⁸.

In addition, we used SHAPEIT2 with the `-duohmm` option (with `-W 5` parameter) to phase 1.1 million HapMap3 SNPs with $MAF > 1\%$ from the imputed genotype data provided by UK Biobank. The 'duohmm' option takes advantage of parent-offspring relations to improve phasing. We merged the haplotypes provided by UKB with the haplotypes for HapMap3 SNPs using QCTOOL, giving haplotypes for 1,652,145 unique SNPs, 1,586,010 of which had $MAF > 1\%$.

We did this to improve the impact of our work, since the HapMap3 SNPs are commonly used for computation of polygenic scores and in meta-analyses. We provide full summary statistics for the effects of 1.5 million SNPs on 9 traits as Supplementary Data.

Improved IBD estimation

In response to comments about the IBD sharing statistics of siblings, we examined the IBD sharing statistics of the siblings at the SNP level. We noticed that the proportion of SNPs called IBD0 by KING greatly exceeded theoretical expectations based on Mendelian segregation for the SNPs near the ends of chromosomes and near the centromeres. This impelled us to develop a superior algorithm for IBD inference that we now include in the SNIPar software package, and which we describe in the Methods:

We developed a Hidden Markov Model (HMM) to infer IBD segments shared between siblings (Supplementary Note). The HMM models the joint distribution of a sibling pair's (unphased) genotypes at a SNP conditional on the IBD state of the siblings at that SNP. To account for linkage disequilibrium between nearby SNPs, we weight the contribution of siblings' genotypes at each SNP to overall likelihood for the chromosome by the inverse of the LD-score of the SNP. We calculated the LD-scores using the LD Score Regression⁴³ software with a 1cM window. The probability of transitioning from one IBD state to another is inferred from the genetic distance between the SNPs. We include a layer that accounts for genotyping errors, which requires a parameter γ for the probability of a genotyping error. We smooth the IBD segments inferred by the HMM to remove short segments that are improbable based on their length in cM and whose neighbouring segments have the same IBD state. This requires a parameter, m , the minimum allowed length (in cM) of an IBD segment that differs from its adjacent segments.

In order to estimate the accuracy of the inferred IBD segments and to choose optimal parameters γ and m , we used 31 families where two siblings and both of their parents have been genotyped (quads) to infer the true IBD state for a subset of SNPs: when both parents are heterozygous, the IBD state of the siblings is equal to 2 minus the absolute difference in the

siblings' genotypes, except when both siblings are heterozygous. We smoothed the true IBD inferred from the quads to account for genotyping errors: if the IBD state at a SNP differed from its neighbours, and both neighbours had the same IBD state, we changed the IBD state of the SNP to be the same as its neighbours.

To infer the IBD segments between siblings, we used the unphased SNPs on the UKB genotyping array with $MAF > 1\%$. We chose the parameters γ and m by performing a grid search over $\log_{10}(\gamma)$ from -5 to -1 in increments of 0.5, and for $\log_{10}(P_{cM}) = \log_{10}(1 - e^{-\frac{m}{100}})$ from -5 to -1 in increments of 0.5. P_{cM} is the probability of observing a segment as short or shorter than m . For each tuple (γ, m) , we calculated the probability of inferring the correct IBD state by comparing the inferred IBD state to the true IBD state for all SNPs where we could infer the true IBD state. We found that $(\gamma, m) = (10^{-4}, 0.01 \text{ cM})$ gave the highest probability of inferring the true IBD state, 99.65%. We give the proportions of SNPs with inferred IBD states 0, 1, and 2 as a function of the true IBD state in Supplementary Table 1. We compared this to IBD segments inferred by KING using the `-ibdsegs` option, which had an overall probability of inferring the true IBD state of 98.5%. Our method therefore gave around a fourfold reduction in IBD errors compared to KING. Furthermore, we found that the distribution of IBD states inferred by KING diverged substantially from the theoretical expectation near the ends of chromosomes, whereas the distribution of our IBD states was close to theoretical expectations from end to end (Supplementary Figure 5).

The improvement in the agreement between inferred IBD sharing statistics between siblings and theoretical expectation is clearly visible in Supplementary Figure 5:

Supplementary Figure 5. IBD0 proportion among sibling pairs across chromosome 1. We show the proportion of sibling pairs, out of 19,290 pairs, that are called as IBD0 for each SNP with MAF>1% on chromosome 1 on the UK Biobank genotyping array. We compare the fraction of pairs that are called IBD0 by SNIPar (black line) and KING (grey line). The theoretical expectation according to Mendelian segregation is 0.25, indicated by the red horizontal line.

Moment-based estimator of correlations between estimates

In thinking about the reviewers' points about LD-score regression (LDSC), we realized that LDSC is not the correct tool to use to answer the questions we want to answer with this study. LDSC models the expected squared Z-statistic for a SNP-effect as equal to a constant, which captures sampling variation and some confounding due to stratification, plus a heritable component that is proportional to the LD-score of the SNP, i.e. how much genetic variation that SNP tags. This has the effect of removing some of the influence of population stratification from the correlations that LDSC estimates. However, in our study, we are interested in whether confounding due to population stratification is biasing population effect estimates, so this is not appropriate. Furthermore, as reviewer 2 pointed out, the assumptions of LDSC are violated by population stratification and assortative mating, which generate long-range linkage disequilibrium between SNPs on different chromosomes. It is therefore not clear exactly what LDSC is estimating when applied to traits such as height and educational attainment, where assortative mating is strong.

In fact, the question that we are interested in answering is much simpler than what LDSC attempts to do. We are interested in the genome-wide correlation between direct effects and population effects, and between direct effects and average NTCs. If we had an infinite sample size, this would be achieved by computing the genome-wide correlation between the estimates for each SNP. However, in finite sample sizes, we

have to account for sampling variation in our estimates. We developed a simple moment-based estimator that leverages the known sampling variance-covariance matrix (which we know to be valid thanks to theory derived in the Supplementary Note) to estimate the genome-wide correlation between the underlying true effects.

We have added a section to the Methods describing this estimator:

Estimation of genome-wide correlations

To estimate correlations between different types of effect, such as direct effects and population effects, we used a moment-based estimator that accounts for the sampling errors in the effect estimates. For example, let $\hat{\delta}_k$ be the direct effect estimate for SNP k , and let \hat{a}_k be the estimated population effect. Then we have that

$$\hat{\delta}_k = \delta_k + \epsilon_{\delta k}; \hat{a}_k = a_k + \epsilon_{ak};$$

where δ_k is the true direct effect for that SNP, and $\epsilon_{\delta k}$ is the sampling error; and a_k is the true population effect for that SNP, and ϵ_{ak} is the sampling error. We assume that the variance-covariance matrix of the sampling errors at each SNP is known:

$$\text{Var} \begin{pmatrix} \epsilon_{\delta k} \\ \epsilon_{ak} \end{pmatrix} = \begin{bmatrix} \sigma_{\delta k}^2 & r_k \sigma_{\delta k} \sigma_{ak} \\ r_k \sigma_{\delta k} \sigma_{ak} & \sigma_{ak}^2 \end{bmatrix};$$

We aim to estimate the genome-wide correlation between the true effects:

$$r_{\delta}(a) = \frac{\text{Cov}(\delta_k, a_k)}{\sqrt{\text{Var}(\delta_k) \text{Var}(a_k)}}.$$

Assuming that the true effects have expectation zero across the SNPs and by applying the Law of Total Variance, we can express the correlation between the true effects as:

$$r_{\delta}(a) = \frac{\text{Cov}(\hat{\delta}_k, \hat{a}_k) - \text{E}[\text{Cov}(\epsilon_{\delta k}, \epsilon_{ak})]}{\sqrt{(\text{Var}(\hat{\delta}_k) - \text{E}[\text{Var}(\epsilon_{\delta k})])(\text{Var}(\hat{a}_k) - \text{E}[\text{Var}(\epsilon_{ak})])}}$$

We use weighted sample estimates of these quantities to obtain our estimator for $r_{\delta}(a)$:

$$\hat{r}_\delta(a) = \frac{\sum_k w_k (\hat{\delta}_k \hat{a}_k - r_k \sigma_{\delta k} \sigma_{ak})}{\sqrt{(\sum_k w_k (\hat{\delta}_k^2 - \sigma_{\delta k}^2)) (\sum_k w_k (\hat{a}_k^2 - \sigma_{ak}^2))}}$$

Similar to LD Score Regression (LDSC)⁴⁵, we use

$$w_k = \frac{f_k(1 - f_k)}{l_k}$$

as the weight for SNP k , where f_k is the allele frequency, and l_k is the LD-score of SNP k . To compute the LD scores, we used the LD Score Regression software package with a 1 centiMorgan (cM) window. The weighting in proportion to $f_k(1 - f_k)$ approximately equalizes the sampling variances across SNPs, and the weighting in proportion to $1/l_k$ accounts for correlations between SNPs due to local LD. To estimate standard errors, we use the same block-jackknife approach as LDSC⁴⁵ with 200 blocks. We excluded SNPs with MAF < 5%. We used the inferred sampling correlation between direct and population effects as a further form of quality control. Outlying values of this correlation are indicative of IBD inference errors or low genotyping quality. We excluded SNPs where, for any trait, the inferred sampling correlation between direct and population effects differed by more than 6 standard deviations from the mean across all SNPs, excluding 101 SNPs.

We assessed the performance of this estimator using updated and expanded simulation results, which we describe next.

Simulation results

We thank the reviewers for suggesting that we simulate simple scenarios to enable us to understand what results to expect as a consequence of different phenomena: assortative mating (AM), population stratification, parental indirect genetic effects (IGEs). We have performed such simulations, with the methodology reported in the new Methods section ‘Simulations using UK Biobank data’, and with results reported in Supplementary Table 2.

We found no evidence for bias in direct effect estimates for any of the scenarios, which we report in the final paragraph of the Results subsection ‘Imputing missing parental genotypes in UK Biobank’:

We also tested the performance of our method in realistic simulations based on genetic data from the UKB ‘White British’ sample (Methods). We simulated traits affected by parental IGEs, population stratification, and AM, and we did not detect any bias in estimates of direct effects across the simulated traits (Supplementary Table 2).

We also examined our estimator of the genome-wide correlation between effects for the simulated traits (Supplementary Table 2), as we describe below.

Updated empirical results with clearer interpretation

Using the new estimator for genome-wide correlations between effects, and the expanded set of SNPs, we obtained slightly different results to the initial submission. Also, for educational attainment (EA), we have used an updated coding of educational attainment that fixed an error in the coding used in previous GWAS of EA. This new coding is described in another paper currently in review at *Nature Genetics*, and we include the relevant section of the Supplementary Note of that paper as a Supplementary File.

However, the main empirical result of the initial submission that estimates of population effects from standard GWAS are poor estimates of direct effect for educational attainment (EA) and cognitive ability remains.

Further, thanks to the realistic simulations we performed, we have been able to improve the interpretation of our results.

The updated section on correlations between effects is now:

Genome-wide association analyses for 9 phenotypes

We used the sample of 39,619 individuals for which parental genotypes were observed or could be imputed (Table 4) to estimate the direct effects, NTCs, and population effects of 1,586,010 SNPs with MAF>1% on 9 phenotypes (Methods). Phenotypes were adjusted for 40 genetic principal components before SNP effects were estimated.

In our PGS analyses (above), we found no evidence for substantial indirect genetic effects from siblings. Therefore, to increase precision of estimates of direct and parental effects, we estimated effects assuming $\eta_s = 0$.

At these sample sizes, power is limited for analysis of direct effects and NTCs of individual SNPs. We therefore focused on estimating the genome-wide correlation between direct and population effects, $r_\delta(\alpha)$ (Methods). This measures the degree to which the effects estimated by standard GWAS reflect direct genetic effects. We also estimated $r_\delta(\alpha)$, the genome-wide correlation between direct effects and average NTCs. To estimate the correlations, we used a moment-based estimator that adjusts for the known sampling variance-covariance matrix of the estimates (Methods).

We estimated $r_{\delta}(\alpha)$ and $r_{\delta}(a)$ for the phenotypes simulated from genetic data from the UKB 'White British' subsample (Methods and Supplementary Table 2). For phenotypes affected by direct effects and parental IGEs in a random-mating population, $r_{\delta}(\alpha)$ corresponds to the correlation between direct effects and average parental IGEs. When there is population stratification or AM, average NTCs capture effects due to other genetic and environmental factors with which the SNP is correlated due to non-random mating, in addition to IGEs from relatives. Theory implies that AM is expected to lead to a high correlation between direct effects and average NTCs²⁵, and therefore direct and population effects, and this is what we found in our simulations. Our simulations also showed that population stratification led to correlations between direct and population effects substantially below 1.

Figure 6. Estimates of correlation between direct and population effects. The estimate is given along with the 95% confidence interval. Direct effects are causal effects due to inheritance of alleles; population effects are estimated by standard GWAS and include direct effects, indirect effects from relatives, and confounding due to population stratification and assortative mating.

We estimated the correlation between direct and population effect estimates using summary statistics derived from a sample of 39,619 individuals from the 'White British' subsample of the UK Biobank where parental genotypes were imputed, using the developed methods, or observed (Methods). Phenotypes were adjusted for 40 genetic principal components prior to analysis. We do not show the results for age at first birth in women and ever-smoked here due to their large standard errors (see Supplementary Table 5). Abbreviations: HDL, high density lipoprotein cholesterol; BMI, body mass index; FEV1, forced expiratory volume in one second.

Across the 9 phenotypes, $r_{\delta}(a)$ was not statistically distinguishable from 1 ($P > 0.05$, one-sided Z-test for $r_{\delta}(a) < 1$) except for EA ($r_{\delta}(a) = 0.739$, S.E.=0.086, $P=1.2 \times 10^{-3}$) and cognitive ability ($r_{\delta}(a) = 0.490$, S.E.=0.086, $P=1.6 \times 10^{-9}$) (Figure 6). We also estimated $r_{\delta}(a)$ (Supplementary Table 5), finding negative correlations for cognitive ability ($r_{\delta}(a) = -0.588$, S.E.=0.094, $P=3.1 \times 10^{-10}$, two-sided Z-test for $r_{\delta}(a) \neq 0$) and neuroticism ($r_{\delta}(a) = -0.421$, S.E.=0.190, $P=0.027$), and a positive correlation for height ($r_{\delta}(a) = 0.666$, S.E.=0.270, $P=0.014$). For height, the results are similar to simulation results for a phenotype affected by direct effects and AM (Supplementary Table 2), consistent with previous analyses showing that, for height, AM is strong and parental IGEs are likely to be weak^{4,29}.

Further investigation of stratification that persists after PC adjustment

The reviewers highlighted that an important potential implication of our results was substantial stratification in GWAS summary statistics for EA and cognitive ability that persists after PC adjustment. We undertook additional analyses to investigate the plausibility of this hypothesis, which we report in the main text:

To investigate whether residual population stratification that persists after adjustment for principal components (PCs)^{9,10} contributes to the lower correlations between direct and population effects for EA and cognitive ability, we adjusted those phenotypes for birth coordinates and the location where each individual was assessed, in addition to principal components (Methods). This increased the estimated correlation for EA to $r_{\delta}(a) = 0.791$ (S.E.=0.066), an increase of 0.053 (S.E.=0.045; $P=0.124$ from a one-sided Z-test for an increase); and increased the estimated correlation for cognitive ability to $r_{\delta}(a) = 0.568$ (S.E.=0.088), an increase of 0.078 (S.E.=0.064; $P=0.113$).

To further investigate the contribution of residual stratification, we computed correlations between genetic associations with birth coordinates (adjusted for PCs) and direct effects, average NTCs, and population effects for all 9 phenotypes (Methods, Supplementary Table 6). The correlations with birth coordinates reflect the degree to which SNP effects on traits, after adjustment for PCs, are correlated with the geographic structure in the population³⁰. Estimated correlations between birth coordinates and direct effects were attenuated towards zero

relative to correlations between birth coordinates and average NTCs and population effects. Furthermore, correlations between birth coordinates and average NTCs and population effects tended to line up along a south-east to north-west axis (Supplementary Figure 7), likely reflecting the phenotypes' correlations with socio-economic status (SES), genetic structure in the UK population, and geographic variation in SES across the UK^{30,31}.

We include Supplementary Figure 7 here showing that average NTCs and population effects are correlated with genetic structure that persists after PC adjustment:

Supplementary Figure 7. Correlations with north and east birth coordinates. Associations between SNPs and north and east birth coordinates that persist after principal component adjustment were assessed by performing a genome-wide association study of north and east birth coordinates in a sample of unrelated individuals from the 'White British' subsample of the UK Biobank (Methods). We estimated correlations between SNP associations with north and east birth coordinate and A) direct effects, B) average non-transmitted coefficients (NTCs), and C) population effects (Methods). Point estimates are plotted as points, with error bars giving 95% confidence intervals. See Supplementary Table 6 for numerical results.

We note that some correlation between direct effects and genetic associations with birth coordinates could be expected if frequencies of causal alleles differ between regions of the UK, as might be expected due to genetic drift, migration, and selection. However, the correlations between average NTCs and birth coordinates tend to be much larger than between direct effects and birth coordinates (which are clustered close to zero), indicating environmental stratification is likely stronger than differences in causal genetic effects between regions.

In light of these analyses and the new simulation results, we have improved our discussion of the interpretation of our results showing low correlation between direct and population effects for EA and cognitive ability:

Population effects, estimated from GWAS in unrelated individuals, capture direct effects and IGEs from relatives, in addition to confounding due to population stratification and AM³. We examined the degree to which GWAS estimates reflect direct effects by estimating the genome-wide correlation between direct and population effects, finding that population effects and

direct effects are not highly correlated (<0.9) for EA and cognitive ability, and we found evidence that residual stratification that persists after adjustment for PCs contributes to this. Another phenomenon that may contribute is ascertainment: If direct effects and NTCs are not very strongly correlated, but both are also correlated with ascertainment, then collider bias³⁷ could push the correlation estimate in the negative direction (as observed for cognitive ability and neuroticism) and reduce correlations between direct and population effects.

Comparison to independent summary statistics

Reviewer 1 highlighted that it would help to compare our results to external summary statistics. We agreed with the reviewer that this would help in interpreting our results, but we decided not to use external, publicly available summary statistics for two reasons: 1) the phenotype definitions could differ from ours, and the cohorts used could differ, which could make interpretation of correlations difficult since they could be clouded by heterogeneity; and 2) many publicly available summary statistics include UK Biobank as one of the main contributors, implying that there is sample overlap of unknown magnitude, which complicates inference. In order to provide comparison of our results to 'typical' GWAS results, we therefore performed our own GWAS using BOLT-LMM in an independent sample of unrelated, 'White British' UK Biobank participants. This way, we could use individuals of the same ancestry and with the same phenotype definitions, and we could ensure the sample was independent (i.e. no third degree or higher relationships with individuals used in our main analysis with imputed parental genotypes).

In addition to providing comparison between our direct effect estimates and summary statistics from a large, independent sample, we obtained new results demonstrating that the correlation between BOLT-LMM summary statistics and direct effects on EA and cognitive ability tends to be smaller than the correlation between BOLT-LMM summary statistics and population effects on EA and cognitive ability.

We report results from these analyses in the results in the main text:

In order to compare our results to summary statistics from an independent sample, we performed a GWAS in a sample of 276,419 unrelated individuals from the 'White British' subsample of the UKB using BOLT-LMM³² (Methods). Comparing direct effects to BOLT-LMM summary statistics, we obtained results consistent with our analysis comparing direct to population effects within the related sample (Supplementary Table 5). We found that correlations between BOLT-LMM summary statistics and direct effects on EA and cognitive ability were attenuated compared to correlations between BOLT-LMM summary statistics and population effects on EA and cognitive ability (Figure 7 and Supplementary Table 7). We found

statistically significant differences between the correlations with direct and population effects ($P < 0.05$, two-sided Z-test for non-zero difference) for 4 out of the 9 phenotypes and east birth coordinate for EA, and for 6 out of the 9 phenotypes for cognitive ability. The largest estimated difference was the correlation between BOLT-LMM summary statistics for EA and direct effects on cognitive ability (correlation=0.428, S.E. 0.070) and population effects on cognitive ability (correlation=0.747, S.E. 0.049), giving a difference of 0.319 (S.E. 0.073). This suggests either shared confounding between EA and cognitive ability, or shared IGEs that are not highly correlated with direct effects on cognitive ability, or both.

Figure 7. Correlations with educational attainment and cognitive ability direct and population effects. We computed correlations between summary statistics from our main analysis of 39,619 White British individuals in the UK Biobank (UKB) where parental genotypes were imputed or observed, and summary statistics from applying BOLT-LMM to an independent sample of 276,419 unrelated White British individuals in the UKB (Methods). In A), we show the correlation between the population effects on educational attainment (EA) estimated in our main analysis and the BOLT-LMM summary statistics for 9 phenotypes and birth coordinates on the x-axis, and the correlation between the direct effects estimated in our main analysis and the BOLT-LMM summary statistics on the y-axis. We show the same in B) for the direct and population effects on cognitive ability estimated in our main analysis. The vertical error bars give the 95% confidence intervals for the correlation with the direct effects, and the horizontal error bars give the 95% confidence intervals for the correlation with population effects. We give numerical results along with P-values for differences between correlations with direct and population effects in Supplementary Table 6. Abbreviations: AAFB, age at first birth in women; BMI, body mass index; FEV1, forced expiratory volume in one second; HDL, high density lipoprotein cholesterol.

We think these results are an important first step towards disentangling the contributions of direct and indirect effects, population stratification, and assortative mating to correlations between phenotypes, as we say in the discussion:

We found evidence that correlations between GWAS summary statistics on many of the 9 phenotypes examined here and EA and cognitive ability are inflated by factors other than direct effects. Application of the methods developed here to larger sample sizes will enable us to

tease apart the relative contribution of direct effects, IGEs, and confounding factors to relationships between phenotypes.

Improved understanding of the impact of population structure

We have added a theoretical analysis of the imputation and regression procedure in a structured population for both sibling pairs and parent-offspring pairs to the Supplementary Note. We report the main results in a new Results subsection:

Effect of population structure

In the above results, we have assumed random-mating. When population structure is present, this leads to bias in the imputed parental genotypes. We analyse the consequences of this in the Supplementary Note. In general, estimates of NTCs are biased by structure, with the bias increasing with Wright's F_{st} . Bias is introduced into estimates of direct effects when data types with different numbers of observed parental alleles are mixed together. For imputation from sibling pairs, the number of observed parental alleles differs with the IBD state of the siblings, introducing a bias into estimates of δ that is approximately equal to $F_{st}\alpha/2$ when F_{st} is small. For relatively homogeneous samples, any such bias is therefore likely to be negligible at the individual SNP level. Further, SNPs with large values of F_{st} will tend to be filtered out during quality control because they violate Hardy-Weinberg Equilibrium.

In the Supplementary Note, we derive an alternative estimator for δ that splits the regression by the number of observed parental alleles, and we prove that this estimator is not biased by population structure. While this estimator is more robust, it is less precise than the estimator described above, which performs a single regression using all individuals, irrespective of the number of parental alleles observed. However, the alternative estimator for δ is still more precise than the estimator based on genetic differences between siblings, having an effective sample size $1 + \frac{1-r}{6(1+r)} \geq 1$ times higher, with a maximum of 7/6 when $r = 0$ (assuming $r \geq 0$).

Since F_{st} is around 10^{-3} between neighbouring countries in Europe, it is likely to be even smaller within the White British sample of the UK Biobank, and therefore any bias in direct effects in our empirical analysis will be negligible. For analysis of populations exhibiting greater levels of structure, the estimator we derive in the Supplementary Note will be useful, and shows that the imputation we propose can be used in arbitrarily structured populations to increase power without introducing bias into direct effect estimates.

Reviewer #1

Remarks to the Author:

This paper is a timely contribution to the reconciliation of genotype-phenotype associations estimated from population-based GWAS samples and within-family effects. It is exciting to see the return of families after recent emphasis on the study of 'unrelated' individuals.

We thank the reviewer for his comments, and we share the reviewer's excitement for the return of families to human genetics.

The paper makes conducts three main analyses (1) imputation of parental and/or sibling genotypes based on nuclear families, (2) deconstruction of population-based GWAS estimates of SNP effects for 9 traits in the UKB into direct and indirect (sibling, maternal, paternal) effects, and (3) estimation of the direct and indirect effects of an education polygenic score on 9 traits in the UKB. The authors estimate a low correlation between GWAS and direct effects for EA, cognitive ability and HDL cholesterol. They also find EA polygenic score to have a significant direct effect on all traits (EA, age at first birth, cognitive ability, BMI, ever-smoked, FEV1, HDL, neuroticism, height) and indirect effect on all traits except ever-smoked and neuroticism.

There is much emphasis in the paper on the imputation part of the manuscript, but little comment on the advancement that this new method brings. There is extensive (slightly overwhelming) literature on imputation, and some related to the genotype imputation of ungenotyped individuals with pedigree links – mostly from livestock. Currently there is no comparison of the new method to other methods, e.g. F-impute and AlphaImpute. As I understand these programs use a rules-based approach and segregation analysis. Is this – in your terminology – equivalent to linear imputation or something else? Other authors have used linear models in complex pedigrees, e.g. Gengler, Mayeres & Szydlowski (2007) Animal and related references. Further, programs such as F-impute (and possibly beagle) use both pedigree and population-based information for imputation of SNP genotypes for ungenotyped individuals. As far as I can see, the method proposed by the authors does not utilise population-based haplotype information for imputation (as it resorts to the allele frequency in the population). Is the current 'IBD' method equivalent to a long-range phasing / haplotype approach? Accuracy of imputation increases with the number of relatives in a family but there are no clear take-away messages regarding data structure. Greater context – in the introduction, up front – for this aspect of the study would help. The authors should give a clearer examples of when their method is superior, possible modification of Fig 3 to show the case with and without IBD information. The paper needs to better articulate the novelty of this approach, given many of the other approaches already developed for

imputation. The relevant literature suggests that either population information (short haplotypes) or information on other close (genotyped) relatives (e.g. uncles, or cousins) may also be useful.

We thank the reviewer for highlighting the need to compare our imputation method to other methods used in animal breeding. As we outline in the summary of changes, we have put some discussion of different methods in both the introduction and discussion, and we now make a direct comparison between our method of imputation and AlphaFamImpute, a recent method from the authors of AlphaImpute for imputing missing genotypes in a nuclear family. We found that our method provides an improvement in terms of bias, and, when using pre-phased genotypes, an improvement in terms of R^2 between imputed and actual parental genotypes. This answers the reviewer's question about whether we are using population-based haplotype information: when pre-phased genotypes are input, then we are indeed using population-based haplotype information to resolve uncertainty about which parental genotypes have been observed in cases where siblings or parent-offspring pairs are both heterozygous and in IBD1. This is not exactly the same as using long-range phasing: long-range phasing can only be performed on datasets with certain properties, whereas local phasing, such as performed by SHAPEIT, can be performed more generally. For the type of imputation we are doing, only the local phase around the target SNP is needed (see Figure 3).

As we now discuss in the discussion, our approach is optimized for the typical kinds of family data found in human genetics: sibling pairs, and parent-offspring pairs, from sets of families with no known pedigree relations between one another. The methods, like AlphaImpute, designed for large, complex pedigrees, do not perform well for these kinds of data, which is presumably why the authors of AlphaImpute released AlphaFamImpute.

Our method can utilize any number of sibling pairs optimally, with or without a genotyped parent, to impute the missing parent(s). While further improvements could be made to leverage other relatives of the missing parent when such pedigree relations are known or could be inferred accurately, for most contemporary human genetic datasets, such improvements are likely to be marginal.

The subsequent analysis are novel, building on the authors previous publications in this area. My main concern is the high degree of collinearity between the sources of information (proband, parental, sibling) combined with a relatively small sample size. If there are four main effects (direct, sibling, maternal, paternal), there are 10 potential pairwise interactions? The authors take a number of steps throughout the manuscript to fix or remove certain parameters, maybe because trying to estimate all these correlated

effects simultaneously results in high uncertainty. The methods are not clear to me, p. 20 for example, how are the SNP effects calculated. They seem to be estimated from GBLUP – what are the estimates of the variance components? The authors assume knowledge of LD score regression on p. 21, it should contain a brief explanation.

We thank the reviewer for pointing out the lack of clarity on the methods and the potential issues of collinearity. First, we point out that the SNP effects are not estimated by GBLUP. We have added a sentence to the first paragraph of the Results section on 'Estimating effects using imputed genotypes' to make this clear:

In general, we replace unobserved parental genotypes with their imputed values in order to estimate the parameters of models (1) and (2). We estimate the direct effect and NTCs of each SNP as fixed effects in a linear mixed model, implemented in SNIPar, that models mean differences between families as a random effect, thereby accounting for phenotypic correlations between siblings (Methods).

The primary goal of our LMM approach was to provide a computationally efficient way to estimate direct effects and non-transmitted coefficients (NTCs) [see section on terminology in summary of changes] that accounts for phenotypic correlations between siblings. We give details in the Methods section 'Estimation of SNP effects', which we have improved to make clearer since the initial submission:

Estimation of SNP effects

Phenotype observations of siblings are correlated through both shared genetic factors and shared environmental factors. In order to obtain efficient estimates of SNP effects from phenotypic observations on multiple siblings from the same family, the phenotypic correlations between siblings should be modelled. We implemented a linear mixed model in SNIPar that achieves this by modelling the mean phenotype within each family as a random effect. Let Y_{ij} be the phenotype of sibling j in family i , then, assuming the overall mean of the phenotype is zero,

$$Y_{ij} = X_{ij}\theta + \mu_i + \epsilon_{ij}; \mu_i \sim N(0, \sigma_F^2); \epsilon_{ij} \sim N(0, \sigma_\epsilon^2);$$

where X_{ij} are the mean-centered (observed or imputed) genotypes; θ is the corresponding vector of parameters; μ_i is the mean in family i , which we model as a mean-zero normally distributed random effect with variance σ_F^2 , independent for each family; and ϵ_{ij} is the residual for individual j in family i , independent for each individual. This implies that, conditional on X , the phenotypic correlation of siblings is $\sigma_F^2 / (\sigma_F^2 + \sigma_\epsilon^2)$.

The columns of X and θ depend upon the data type and model being estimated. The default is for the columns of X to be the individual's genotype, the individual's father's imputed or observed genotype, and the individual's mother's imputed or observed genotype, with $\theta = [\delta, \alpha_p, \alpha_m]^T$. When only sibling genotypes are available, in order to prevent collinearity, the columns of X reduce to the individual's genotype and the imputed sum of maternal and paternal genotypes, and $\theta = [\delta, \alpha = (\alpha_p + \alpha_m)/2]^T$. We also provide an option in SNIPar to add the proband's siblings' genotypes to the regression in order to fit indirect effects from siblings.

For estimation of the effects of genome-wide SNPs, SNIPar first infers the variance components σ_F^2 and σ_ϵ^2 by maximum likelihood for a null model without any SNP effects, which can be done in $O(n)$ computations (Supplementary Note). We then fix the variance components at their maximum likelihood estimate for estimation of the SNP effects. Given the variance components, the vector of effect estimates for a SNP can be obtained analytically in $O(n)$ computations (Supplementary Note). Our software package, SNIPar, estimates the direct effect and NTCs of a SNP by performing a single regression that uses all genotyped and phenotyped individuals for whom imputed and/or observed parental genotypes are available.

This also gets to the reviewer's concern about collinearity. Performing a family based GWAS, even with fully observed genotypes, leads to a regression where the covariates have relatively high correlations: 0.5 between proband and the proband's first degree relatives. The imputation procedure leads to higher correlations between the covariates in the regression, but this does not typically create any issues in terms of numerical stability. The exception is when parental genotypes are imputed from siblings alone, which creates perfect collinearity between imputed paternal and maternal genotypes: in this case, we advise, and provide an option in the software, to regress onto the proband genotype and sum of imputed paternal and maternal genotypes, which estimates the direct effect and average NTC.

In the revised manuscript, we no longer use LD-score regression for inference of correlations between effects (see summary of changes), so we do not describe it in any detail, although we reference it at certain points. We give a description of the moment-based estimator we now use in the Methods.

Is Supplementary Figure 4 the genetic correlation between direct and indirect-parental effects? There is not much comment on these results – are these 95% CI? There seems a significant negative correlation for cognitive ability, and positive correlation for height. Under what conditions can a negative (or positive) correlation be generated

between a probands direct genetic value and its mid-parent value? The expectation should be zero? Can the authors clarify.

We have removed the supplementary figure from the revised manuscript. Using the new moment-based estimator, the standard errors on the correlation between direct effects and average NTCs increased substantially for certain traits compared to using LD Score-Regression. While the standard errors are larger, the interpretation of what we are estimating is clear, unlike for LDSC. The larger standard errors make a figure displaying the results difficult to inspect meaningfully. We report the estimates in Supplementary Table 5, and we discuss the results in the Results section on 'Genome-wide association analyses for 9 phenotypes':

We also estimated $r_{\delta}(\alpha)$ (Supplementary Table 5), finding negative correlations for cognitive ability ($r_{\delta}(\alpha) = -0.588$, S.E.=0.094, $P=3.1 \times 10^{-10}$, two-sided Z-test for $r_{\delta}(\alpha) \neq 0$) and neuroticism ($r_{\delta}(\alpha) = -0.421$, S.E.=0.190, $P=0.027$), and a positive correlation for height ($r_{\delta}(\alpha) = 0.666$, S.E.=0.270, $P=0.014$). For height, the results are similar to simulation results for a phenotype affected by direct effects and AM (Supplementary Table 2), consistent with previous analyses showing that, for height, AM is strong and parental IGEs are likely to be weak^{4,29}.

While we are able to provide a clear interpretation of the results for height as being consistent with both theory and simulations of a trait affected by direct effects and assortative mating, we are not able to give a clear interpretation of the negative estimates for cognitive ability and neuroticism. As we say in the Results,

For phenotypes affected by direct effects and parental IGEs in a random-mating population, $r_{\delta}(\alpha)$ corresponds to the correlation between direct effects and average parental IGEs. When there is population stratification or AM, average NTCs capture effects due to other genetic and environmental factors with which the SNP is correlated due to non-random mating, in addition to IGEs from relatives. Theory implies that AM is expected to lead to a high correlation between direct effects and average NTCs²⁵, and therefore direct and population effects, and this is what we found in our simulations. Our simulations also showed that population stratification led to correlations between direct and population effects substantially below 1.

It is possible that non-random mating has led to a negative correlation between the direct effects of SNPs and other genetic and environmental factors affecting both cognitive ability and neuroticism, and this is what we are observing. In the Discussion, we also posit that collider bias, induced by trait related study participation, could influence these results:

Another phenomenon that may contribute is ascertainment: If direct effects and NTCs are not very strongly correlated, but both are also correlated with ascertainment, then collider bias³⁷ could push the correlation estimate in the negative direction (as observed for cognitive ability and neuroticism) and reduce correlations between direct and population effects.

We believe that it will be up to future studies, examining both representative and non-representative samples, to determine what exactly generates negative correlations between direct effects and NTCs. These empirical results demonstrate the power of our novel methodology, and they imply there are potentially serious issues with the way that GWAS has been conducted for traits such as cognitive ability, but we think it is beyond the scope of our study to identify all processes contributing to the empirical results we obtained.

The main findings from the results, Fig 6, suggest a low correlation between the direct and population effects for education, cognitive ability and HDL. However, we want to know the correlation between the population effects and the true (unobserved) direct effects. Is this what was done in simulation, or the correlation between the inferred direct effects, it is not clear. Can the authors show that their method produces unbiased estimates of the direct effects? Is Fig. 6 the correlation between population effect (direct + avg. parental) and direct from this population? A more useful parameter would be the correlation with the population effect from the latest GWAS using unrelated individuals. Is the population effect bias from indirect effects decreased with increasing sample size?

Thank you for raising these important points, which we have addressed.

We showed theoretically that our method produces unbiased estimates of direct genetic effects in a random-mating population, and we derive what bias to expect due to population structure, which is negligible for relatively homogeneous populations as examined here. We did not find any detectable bias in our estimates of direct effects in our simulations examining different scenarios: parental indirect genetic effects, assortative mating, and population stratification, as described above.

We also show that our estimator (see summary of changes) for the correlation between direct and population effects and between direct and average NTCs produces the correct answer in the simulations where the true value is known, i.e. the simulations of a random-mating population with parental indirect genetic effects. This estimator accounts for the sampling variation in the estimates, so is estimating the correlation between the true unobserved effects, not the estimates. Figure 6 does indeed show the correlation from estimates within the same sample. The estimator accounts for the known sampling

variance-covariance matrix between the different effect estimates, however, so this should not be an issue.

In response to your point about comparing to external summary statistics, we have included a comparison to BOLT-LMM summary statistics computed in a large, independent sample of unrelated individuals from UKB (see summary of changes), finding results consistent with our analysis of the related sample. We did not compare to publicly available summary statistics in part because they are likely to have overlapping samples with our analysis, and the unknown sampling correlation this induces makes inference tricky.

Supplementary Note Table 8 makes it clearer as to which type of parameters can be estimated from different types of data and where the information is coming from. This is presented in the context of a meta-analysis but may be more useful (in some form) in the main text. It would make interpretation of the UKB results easier, as it is clear that there are few trios or parent-offspring pairs, thus little power to distinguish paternal / maternal effects.

Thank you for the suggestion. We have now put such a table in the results section on 'Combining different missing data types' in the main text:

Observed genotypes	Y_{i1} regressed on	$E[\hat{\theta}]$
Proband (Fig 1h)	g_{i1}	$\delta + (\alpha_p + \alpha_m)/2$
Sibling pairs (Fig 1e)	$g_{i1}, g_{i2}, \hat{g}_{\text{par}(i)}$	δ η_s $(\alpha_p + \alpha_m - \eta_s)/2$
Father-child pairs (Fig 1c)	$g_{i1}, g_{p(i)}, \hat{g}_{m(i)}$	δ α_p α_m
Mother-child pairs (Fig 1d)	$g_{i1}, \hat{g}_{p(i)}, g_{m(i)}$	δ α_p α_m
Trios (Fig 1b)	$g_{i1}, g_{p(i)}, g_{m(i)}$	δ α_p α_m
Quads (Fig 1a)	$g_{i1}, g_{i2}, g_{p(i)}, g_{m(i)}$	δ η_s $\alpha_p - \eta_s/2$ $\alpha_m - \eta_s/2$

Table 2. Examples of regressions and expected regression coefficients for different data types. In the first column, we give the data type in terms of the observed genotypes in the nuclear family, referencing the relevant subfigure of Figure 1; in the second column, we give an example of a regression that could be performed using that data type and parental genotypes imputed from the observed genotypes; in the third column, we give the expected column vector of regression coefficients from performing the regression. Terms: Y_{i1} is the phenotype of sibling 1 in family i ; g_{ij} is the genotype of sibling j in family i ; $g_{p(i)}$ is the paternal genotype, and $\hat{g}_{p(i)}$ is the imputed paternal genotype; $g_{m(i)}$ is the maternal genotype, and $\hat{g}_{m(i)}$ is the imputed maternal genotype; $\hat{g}_{par(i)}$ is the imputed sum of maternal and paternal genotypes; δ is the direct effect; η_s is the indirect sibling effect; and α_p and α_m are, respectively, the paternal and maternal non-transmitted coefficients (NTCs).

Detailed comments:

Why are ‘assortative mating’ effects classified as confounding but not real/causal effects? (introduction, line 10 para 2). Assortative mating creates gametic phase disequilibrium, i.e. correlations between trait-increasing alleles across chromosomes. It is a real effect, and generates real genetic variance. If the analysis is one SNP at a time then the effects of the correlated SNP are captured but it is still a genetic effect. Traits under assortative mating, such as height or EA, have higher (true) genetic variance in the current population than that under random mating.

We agree that the inflation of the SNP’s correlation with a phenotype due to assortative mating is genetic in origin, i.e. the marginal effect is capturing the effect of that SNP plus effects of other SNPs through linkage disequilibrium induced by assortative mating. However, this inflation of correlation is a confounding effect in the sense that it does not reflect the causal effect of inheritance of one allele vs. the other at that SNP, so the marginal effects estimated by GWAS are biased as estimates of causal effects of the SNP in question. In the introduction, we make clear that confounding is the result of correlations, induced by non-random mating, between the SNP in question and other genetic and environmental factors affecting the trait:

Confounding can occur due to population stratification and assortative mating (AM), which lead to correlations between the allele and other genetic and environmental factors affecting the phenotype.

I find the ‘linear’ and ‘non-linear’ description of imputation distracting. Non-linear means using haplotype information? – non-linear implies (to me) imputation that does not follow the normal rules of inheritance.

We thank the reviewer for highlighting this potential source of confusion. We intended to make clear that the kind of imputation we are doing differs from typical types of imputation methods in statistics, which are often linear. We have added some clarification of what we mean in the first paragraph of the Results section ‘Missing data framework’:

Non-linear imputation refers to imputation that leverages knowledge about which particular alleles have been transmitted from parent(s) to offspring, leading to imputed genotypes that are not linear functions of observed genotypes. Linear imputation predicts an unobserved

genotype using a linear combination of the observed genotypes with coefficients that minimize mean squared error (MSE), and is performed using the known variance-covariance matrix, assuming random mating, of the four genotypes (g_{i1} , g_{i2} , $g_{p(i)}$, $g_{m(i)}$) (Supplementary Figure 1a and Supplementary Note).

We derive the relevant linear imputations in Appendix D of the Supplementary Note.

We make clear why we think this distinction is important in the discussion:

Mendelian Imputation, used appropriately, produces unbiased estimates of parameters along with valid sampling errors (Supplementary Note). This makes it rather unique among single imputation methods in situations where the amount of missing information is substantial³⁶. These properties derive from the fact that missing genotypes are imputed as a non-linear function of observed ones, using known laws of Mendelian Inheritance, thereby adding information for parameter estimation, without introducing additional noise.

There are well documented effects for assortative mating for human height, and yet the results in Fig 6 suggest a strong correlation between direct and population effects?

The results for height fit fairly well the simulation results for a phenotype with direct effects and assortative mating at equilibrium: a moderately high correlation between direct effects and average NTCs, and a high correlation between direct and population effects. These results are supported by theory (Nagylaki, T. Assortative mating for a quantitative character. *J. Math. Biol.* 16, 57–74 (1982).) showing that the inflation of the marginal correlation between genotype and phenotype under assortative mating at equilibrium is related to the variance that SNP would explain in a random-mating population, implying a correlation between direct effects and average NTCs. We have stated an interpretation of the height results:

For height, the results are similar to simulation results for a phenotype affected by direct effects and AM (Supplementary Table 2), consistent with previous analyses showing that, for height, AM is strong and parental IGEs are likely to be weak^{4,29}.

Why the low correlation for HDL? (Fig 6).

In the revised analysis, we no longer find a correlation between direct and population effects for HDL that is statistically distinguishable from 1.

What is the frequency of the 15 different missing information imputation patterns in the UKB? A straight statement of these cases/number of families would be useful

(statements on pg 11 are unclear). Most (>88%) are case e. Are the configurations for the ungenotyped proband (last 2 lines Fig 1) relevant? Top of page 11 sounds like only analysed sibling pairs but I don't think this was the case. The results should state the sample size for each analysis and be clearer as to what type of information is contributing to each estimate.

Thank you for pointing out that we did not clearly state which sample was used for which analysis. We have taken steps to make this clear in the revised manuscript. At the start of the main empirical results section ('Genome-wide association analyses for 9 phenotypes'), we now state:

We used the sample of 39,619 individuals for which parental genotypes were observed or could be imputed (Table 4) to estimate the direct effects, NTCs, and population effects of 1,586,010 SNPs with MAF>1% on 9 phenotypes (Methods).

We also include a table (4) that gives the number of probands from each data type:

We applied our methods to the 'White British' subsample of the UKB²⁷. Using KING²⁸, we identified a sample of 39,619 individuals for which parental genotypes were observed or could be imputed (Table 4). We inferred IBD segments for all 19,290 sibling pairs using a Hidden Markov Model implemented in SNIPar (Methods and Supplementary Note).

Data Type	N (probands)
Proband and sibling(s) but no parents (Fig 1e)	35,197
Proband and parent (Fig 1c, 1d)	3,216
Proband, parent, and sibling(s) (Fig 1f, 1g)	312
Proband and both parents (Fig 1b)	832
Proband, both parents, and sibling (Fig 1a)	62
Total	39,619

Table 4. Summary of data types in the 'White British' UKB subsample.

There seem missing co-efficients in the eq. on the bottom of page 7.

Thanks for pointing this out. We were using the *R* syntax for a regression equation, but, on reflection, this is not appropriate for a research paper, so we have changed this to the full regression equation.

There are many places throughout the manuscript and supplementary material where the terms are not defined where they should be. e.g. 'PGS' (pg. 1, line 3 after

introduction), 'f' and 'X' legend to Supplementary Figure 1 (both) and 2 ('f' only), η_s and δ in legend of Supplementary Figure 3, etc. Not an exhaustive list.

Thank you for pointing these out. We have fixed the specific instances you point out, and we have gone through and checked that terms are defined where they should be. We now include a Table at the start of Supplementary Note that lists all terms where the models are introduced.

Ref [2] is incorrect. Should be Meuwissen, Hayes & Goddard.

Thank you. We have corrected this.

It seems that there is also no significant effect of the parental education polygenic score on FEV1 also (p. 14 Figure 7).

We apologise for the confusion here. We had put the figure for the analysis including the indirect effect from the sibling in the main text, where the power is lower, but primarily discussed results from the analysis assuming the indirect sibling effect is zero. In response to this point and reviewer 2, we have changed main text figure to the analysis assuming the indirect sibling effect is zero (now Figure 5), where a significant average NTC of the education PGS on FEV1 can clearly be seen:

Figure 5. Decomposition of the EA polygenic score effect assuming no indirect sibling effects. The standardized effect estimate (SD change in phenotype per SD change in PGS) is given along with the 95% confidence interval. Estimates were derived from a sample 39,619 individuals with imputed and/or observed parental genotypes. Effects were estimated by joint regression of individuals' phenotypes onto their own PGS, and their mother's and father's (imputed/observed) PGS (Methods). We give the average of the estimated maternal and paternal non-transmitted coefficients (NTCs), adjusted for bias due to imputation in the presence of assortative mating (Methods), as the 'average NTC' here, and the difference between maternal and paternal NTCs as 'maternal-paternal'. Phenotypes were adjusted for 40 genetic principal components prior to analysis. Abbreviations: HDL, high density lipoprotein cholesterol; BMI, body mass index; FEV1, forced expiratory volume in one second.

The figure with the model including the indirect sibling effect is now Supplementary Figure 6.

I'm not sure that being deceased is a precondition for multi-point linkage analysis (p. 14, discussion).

e agree. We have changed this to "Conceptually, this is similar to multipoint linkage analysis performed with pedigrees."

Methods: UKB Sample. No detail on number of SNP, MAF, chip etc etc for UKB. How did they phase their genotypes? What do you mean by 'family based gwas' – not explained. sample size? Did the mean and variance in IBD sharing match expectations for sib-pairs?

Thank you for pointing out the lack of detail. The relevant methods section has been updated to include more detail on the genotype data:

Haplotypes for the SNPs that were present on both the UKB Axiom and the UK BiLEVE genotyping arrays and that passed quality control were provided by UK Biobank (UKB)²⁸. Phasing was performed using SHAPEIT3⁴⁰ and the 1000 Genomes Phase 3 dataset⁴¹ as a reference panel. This resulted in phased haplotypes for a set of 658,720 autosomal SNPs with an estimated switch error rate of 0.229%²⁸.

In addition, we used SHAPEIT2 with the `-duohmm` option (with `-W 5` parameter) to phase 1.1 million HapMap3 SNPs with $MAF > 1\%$ from the imputed genotype data provided by UK Biobank. The `'duohmm'` option takes advantage of parent-offspring relations to improve phasing. We merged the haplotypes provided by UKB with the haplotypes for HapMap3 SNPs using QCTOOL, giving haplotypes for 1,652,145 unique SNPs, 1,586,010 of which had $MAF > 1\%$.

We state the number of SNPs with $MAF > 1\%$ that we estimated direct effects and non-transmitted coefficients from in the main text:

We used the sample of 39,619 individuals for which parental genotypes were observed or could be imputed (Table 4) to estimate the direct effects, NTCs, and population effects of 1,586,010 SNPs with $MAF > 1\%$ on 9 phenotypes (Methods).

Table 4 lists the number of probands by data type.

We have removed the phrase 'family based GWAS' from the revision.

We thank the reviewer for questioning whether the statistics of our inferred IBD segments match theoretical expectations. We now include a Supplementary Figure (5) that shows the close agreement between the proportion of SNPs called IBD0 by our IBD algorithm and the theoretical expectation of 0.25 across chromosome 1. This Figure also shows a clear improvement over the IBD segments provided by KING (used in the initial submission). Furthermore, the mean genome-wide relatedness between the sibling pairs 0.502, and the standard deviation in relatedness is 0.040, which is close to the theoretical expectation (see Visscher 2006, Assumption-free estimation of heritability from genome-wide identity-by-descent sharing between full siblings.). We also test the accuracy of our IBD inference algorithm using parent-sib quads, where the true IBD state can be inferred when both parents (and not both sibs) are heterozygous. We found that our IBD algorithm obtained the correct IBD state 99.65% of the time, as we detail in the Methods section 'IBD inference'. We show the proportion of SNPs called with each IBD state as a function of the true IBD state in

Supplementary Table 1. While our IBD inference is not perfect, the rate of error is low enough that it will have a negligible impact upon our imputations, as reflected in the tests we perform using families with genotyped parents, as we detail in the Methods section 'Imputation of missing parental genotypes':

Using the inferred IBD segments (above), we imputed missing parental genotypes for 1,586,010 SNPs (the union of the genotyping array SNPs and the HapMap3 SNPs with MAF>1%) for which we had phased haplotypes. We examined the bias in the imputed parental genotypes by performing the imputation for families with observed parental genotypes as if the parental genotypes were missing. If the imputation is unbiased, then the regression coefficient of the observed parental genotypes onto the imputed parental genotypes should be 1. This is because the covariance between the imputed parental genotypes and the observed parental genotypes should be equal to the variance of the imputed parental genotypes (Supplementary Note). Based on data from 31 families with two siblings and two parents genotyped, we obtained a regression coefficient of 0.9997 for regression of the sum of observed parental genotypes onto the imputed sum of parental genotypes. Based on data from 894 families with at least one parent genotyped, we obtained a regression coefficient of 0.9989 for regression of the observed parent's genotype onto the imputed parental genotype. These results show there is negligible bias in the imputed parental genotypes.

Reviewer #2:

Remarks to the Author:

This paper has two goals. The first is to show how genomic data of the nuclear family (sibs and parents) can be imputed using IBD information in a way that causes no biases in downstream analyses, even when the vast majority of the data is missing. This is important for the second goal, which is to estimate direct and indirect genetic influences in large biobanks (here, the UKB) that are not oversampled for close relatives. The authors demonstrate that their imputation method (instantiated in software SNIPar) produces unbiased estimates of these influences and that (perhaps more surprisingly) those estimates have the correct standard errors. By applying the proposed method to an educational attainment (EA) and cognitive ability (which I'll call "IQ") PGS, the authors observed a lower direct/(direct+indirect) effect ratio than was observed in their previous study with Iceland cohort. To explain these differences, the authors claimed that the combined influence of indirect effects, population stratification, and assortative mating is stronger in the UK than in Iceland.

We thank the reviewer for accurately summarizing some of the main results of the paper. We agree that the fact that the imputation method produces both unbiased estimates and valid standard errors is surprising: as we say in the discussion, this is rather unique among single imputation methods in statistics with substantial amounts of missing data, making this of interest to the statistics community as well as the genetics community.

This is an excellent idea and represents a tremendous amount of work on the authors' part. However, the manuscript really needs work in multiple areas. The three biggest problems are, first, parts of it are overly confusing, use poor terminology, and are inconsistent. More work needs to be put into clarity. Second, the surprisingly large and potentially important result that most of the population effect of EA and IQ are not direct, and that this may be due to stratification, is not adequately explored. Third, there are a lot of different influences on the results (e.g., assortative mating alone influences Mendelian imputation, the GWAS estimates, LDSC estimates, and the correlation between transmitted and non-transmitted PGSs). These various factors influence the results the authors describe in complicated and sometimes competing ways. It would be nice to see a simulation that shows that the method works as advertised under simplifying scenarios, and to understand how big the biases can be under other scenarios in order to better gauge the relative importance of these factors. I describe these and other points below as they arose in the manuscript.

We are glad that the reviewer thinks this is an excellent idea, and we thank the reviewer for giving constructive criticisms. We have made considerable efforts to fully address all of the reviewer's concerns, as we outline in the summary of changes and in our responses to specific points below. We believe the revised manuscript is much stronger thanks to these revisions.

p.3 - alpha's referred to as "parental effects" is poor terminology, misleading, and confusing. Words matter. First, the caveat the authors provide after describing their choice of terminology could be easily forgotten or missed. Second, and more importantly, the alphas represent only those influences of parental genes mediated via the parental phenotype. Unless the trait is completely heritable (in which case there would be no genetic nurture), there must be additional parental influences that are environmental in nature and thus going into the epsilon terms in the model. Thus, "parental effects" as used by the authors are only a potentially small portion of the actual parental phenotype -> offspring phenotype influences. Part of the problem in my view is that the models used by the authors don't make this explicit (e.g., these parental phenotype -> offspring phenotype influences, as well as the sib -> sib influences, are buried in the epsilon term). At the least, I'd suggest authors describe "alpha" differently. They used to call it "genetic nurture," and while I can see that this is inflated by AM and its estimate can be biased by stratification,

“genetic nurture” or “parental indirect genetic effects” or “residual parent-offspring genetic association” aren't great, but they all seem much preferable to the misleading “parental influences” term.

We agree with the reviewer that the original terminology was potentially misleading and confusing. As we outline in the summary of changes, we have changed the term for ‘parental effects’ to ‘non-transmitted coefficients’ (NTCs), since these are the expected coefficients on the non-transmitted parental alleles from a regression of proband phenotype onto proband genotype and the non-transmitted parental alleles. We prefer this to a term that specifies parental indirect genetic effects (IGEs) since these coefficients can capture factors other than parental IGEs, including assortative mating and stratification, as the reviewer points out. This allows to more clearly specify when we mean parental IGEs as opposed to other phenomena that may contribute to the NTCs. We also believe the ‘non-transmitted coefficient’ term will not mislead people into thinking that we are capturing all (or even any by necessity) parental influences on the offspring.

p.3 - when first introduced, authors should explain what “non-linear imputation” is and why it is relevant.

In the first paragraph of the ‘Missing data framework’, we now state:

Non-linear imputation refers to imputation that leverages knowledge about which particular alleles have been transmitted from parent(s) to offspring, leading to imputed genotypes that are not linear functions of observed genotypes. Linear imputation predicts an unobserved genotype using a linear combination of the observed genotypes with coefficients that minimize mean squared error (MSE), and is performed using the known variance-covariance matrix, assuming random mating, of the four genotypes (g_{i1} , g_{i2} , $g_{p(i)}$, $g_{m(i)}$) (Supplementary Figure 1a and Supplementary Note).

We believe this is an important distinction to make since most imputation methods in statistics are linear, and linear imputation in most of the cases we examine would lead to a regression design matrix that is collinear, and therefore not enable identification of the model parameters: Mendelian Imputation, by contrast, often leads to imputed genotypes that are non-linear functions of observed genotypes, thereby adding information for parameter estimation in a multiple regression context. This is reflecting in paragraph 2 of the Discussion:

Mendelian Imputation, used appropriately, produces unbiased estimates of parameters along with valid sampling errors (Supplementary Note). This makes it rather unique among single imputation methods in situations where the amount of missing information is substantial³⁶.

These properties derive from the fact that missing genotypes are imputed as a non-linear function of observed ones, using known laws of Mendelian Inheritance, thereby adding information for parameter estimation, without introducing additional noise.

A few suggestions for interpretability of Figure 2:

- would be easier to understand if "?" replaced the values of the parental genotypes that are unknown, given that the values provided are just arbitrary numerical examples.

Thank you for the suggestion. We have changed the unobserved parental alleles to '?'.

- at top, write "Stretches where siblings are IBD0", "Stretches where siblings are IBD1", etc.

Thank you for the suggestion. While we can see that the suggestion would help relate what we are doing at the SNP level to the level of haplotype sharing between siblings, we wanted this figure to make concrete what happens at the individual SNP level. If we wrote 'stretches...' at the top of the figure, this may introduce confusion about what we are doing at the SNP level since the figure displays the sibling and parental genotypes at the SNP level, and these would vary from SNP to SNP within a IBD segment.

- rewrite last sentence a bit. "...and IBD segments can be determined by finding segments of chromosomes where siblings' genotypes always have at least one allele the same (are IBS1 or IBS2), in which case the segment is IBD1, or always have both alleles the same (IBS2), in which case the segment is IBD2".

Thank you for the suggestion. We have changed the legend to state that IBD segments can be inferred from dense genotype data on siblings, referencing the Methods and Supplementary Note, where we discuss the IBD inference algorithm we use in the revised manuscript. We also have a sentence that makes clear that the IBD state cannot be determined from a single SNP alone, in general. After making the changes, the figure and legend are now:

Figure 2 Imputation of parental genotype from sibling genotype. Here we illustrate how, given knowledge of the IBD state between the siblings' alleles (alleles coded by '0' and '1'), the sum of the maternal and paternal genotypes can be imputed. If the siblings do not share any alleles identical-by-descent (IBD), then all four parental alleles are observed (IBD0). If the siblings share one allele by descent from their parents, then three parental alleles are observed, and one allele is unobserved (IBD1). If the siblings share both alleles by descent from their parents, then two parental alleles are observed and two are unobserved (IBD2). When parental alleles are unobserved, we impute them with the population frequency of allele '1', f . The IBD state between siblings changes with the recombination events that occurred during meiosis in the parents of the siblings and can be inferred from dense, genome-wide genotype data on siblings (Methods and Supplementary Note). The IBD state at a SNP cannot be determined from sibling genotypes at that SNP alone, other than when siblings are discordant homozygotes ('00' and '11'), which implies IBD0, as shown here.

p. 6 - "the squared correlation between imputed and observed parental genotype is $3/4$, higher than based on best linear unbiased imputation (Supp Fig 1)." I think I see why ($1/4 \cdot 1 + 1/2 \cdot 3/4 + 1/4 \cdot 1/2 = 3/4$), but this need to be made clearer, and the reference to Supp Figure 1 is not helpful - which subfigure? Which part of the subfigure shows this? I was looking at 1e but $3/4$ is not in there anywhere.

Thank you for the suggestion to make this clearer. We have changed this section of text to now read:

the squared correlation between imputed and observed parental genotype is $3/4$. This is because we observe 2 parental alleles in IBD2, with probability $1/4$; 3 parental alleles in IBD1, with probability $1/2$; and 4 parental alleles in IBD0, with probability $1/4$; giving an average of 3 observed parental alleles. This is higher than based on best linear unbiased imputation, $2/3$ (Supplementary Note).

We refer to Supplementary where we now derive the variances of the linear imputations in Appendix D.

p.8 and Supp. p. 13, Regarding $Y_1 - Y_2$ being orthogonal to $Y_1 + Y_2$. These won't be orthogonal if Y_1 and Y_2 have different variances, such as might occur, for example, if Y_1 is always the male phenotype and Y_2 always the female phenotype.

We have removed this from the main text in the revised manuscript, in part to save space, and in part because we do not explicitly consider a regression where we use both $Y_1 - Y_2$ and $Y_1 + Y_2$ as orthogonal variables in the main text. We have changed the text in the supplement to say that these two variables will be approximately uncorrelated if siblings have approximately the same residual variances.

p. 8 - "Letting r be the correlation of the siblings' residuals, which will be approximately equal to their phenotypic correlation for polygenic traits..." I assumed the authors meant the epsilon component, but this can't be right. The epsilon (error) components (which are independent of genes or other direct effects) may be correlated (e.g., due to neighborhoods, friends, schools, etc), but they may not be either. To say that they will be approximately equal to their phenotypic correlation is either wrong, or I'm misunderstanding what the "siblings' residual" is. Can authors clarify?

The epsilon component for the single SNP model includes all factors influencing the phenotype that are uncorrelated with the proband and parental genotypes at that SNP. For complex human traits, individual SNPs typically explain only a tiny fraction of the phenotypic variance, so the variance of the epsilon term will be approximately equal to the phenotypic variance, and the correlation between siblings' epsilon components will be approximately equal to the phenotypic correlation of the siblings. We have added text to make this clearer:

Letting r be the correlation of the siblings' residuals (which will be approximately equal to their phenotypic correlation when the SNP explains a small fraction of the phenotypic variance, as is typical for complex human traits)

p.11 - $r_{(\Delta\beta)}$ – the authors state that this is the correlation between direct and parental effects. At first I interpreted this as "passive G-E correlation", because that's what the authors' terminology suggests, but then remembered that they are calling β (which is α when $\eta_s=0$... geez) "parental effects" when it is really indirect genetic effects/genetic nurture. So then what is the interpretation of $r_{(\Delta\beta)}$? The correlation between direct effects and genetic nurture effects? And Figure 5 shows several different lines for $r_{(\Delta\beta)}$, but I'd imagine $r_{(\Delta\beta)}$ is itself a function of h^2_{β} , meaning that some parts of the lines may be impossible or unlikely. And how does assortative mating and/or stratification influence these quantities of $r_{(\Delta\beta)}$ and h^2_{β} ? Much of my confusion here could have been cleared up if the

derivations of them were somewhere, but I can't find them in the main text, Methods, or Supp. My apologies if I've missed it, but if they indeed are not shown, they should be somewhere.

We apologise for the confusion in the initial submission. Some of this confusion was due to our use of a tool, LD Score Regression (LDSC), to infer parameters that it was not originally designed to infer. As we outline in the summary of changes, we have derived a new estimator for the genome-wide correlation between direct effects and population effects, and between direct effects and average NTCs. This differs from LDSC, which attempts to separate stratification from causal genetic effects, and which makes assumptions about the relationship between marginal effects and joint effects that are violated by assortative mating and population stratification. The goal of our analysis is to answer the question: in an infinite sample size, how correlated would direct effects and population effects be? We do not make any assumptions about the process that generates direct or population effects in our estimator. Indirect effects, stratification, and assortative mating can all contribute to differences between direct and population effects, and between direct effects and non-transmitted coefficients, as we demonstrate in our new simulation results (Supplementary Table 2). Our analysis puts a number on how much bias the combined influence of these phenomena generates in marginal population effect estimates, which is what standard GWAS reports and forms the basis of current methods of polygenic prediction. While separating out the contribution of indirect genetic effects from assortative mating and population stratification would help interpret these results, doing so would require unbiased estimates of indirect genetic effects at the SNP level, which is beyond the scope of this paper. In light of these changes in the manuscript, we have removed Figure 5 and the discussion of Figure 5, since Figure 5 only applies to situations where the non-transmitted coefficients are capturing indirect genetic effects alone.

Furthermore, we define the quantities in the new section of the main text describing the simulation results:

At these sample sizes, power is limited for analysis of direct effects and NTCs of individual SNPs. We therefore focused on estimating the genome-wide correlation between direct and population effects, $r_{\delta}(\alpha)$ (Methods). This measures the degree to which the effects estimated by standard GWAS reflect direct genetic effects. We also estimated $r_{\delta}(\alpha)$, the genome-wide correlation between direct effects and average NTCs. To estimate the correlations, we used a moment-based estimator that adjusts for the known sampling variance-covariance matrix of the estimates (Methods).

We believe that introducing these quantities here, along with simulation results showing what these quantities are under different scenarios, help in interpretation of the empirical results.

Our simulation results show that stratification is expected to decrease the correlation between direct and population effects, while assortative mating is expected to lead to a moderately high correlation between direct effects and average NTCs, and a high correlation between direct and population effects.

For height, our results fit expectations from theory and simulations for a trait affected by direct effects and assortative mating. However, for traits such as educational attainment and cognitive ability, we are dealing with multiple complex phenomena (assortative mating, stratification, indirect effects, and interactions between these forces) that can influence genotype-phenotype association beyond direct effects. While we provide analyses that show the presence of residual stratification in population effects for these and other traits, and provide evidence that adjusting for geographic factors beyond principal components increases the correlation between direct and population effects, providing a full account of all the forces that influence genotype-phenotype association for these traits is beyond the scope of this paper.

p. 13 (top) & Supp Fig 4 – Again, I’m finding the notation confusing and inconsistent. The main text discusses $r_{\Delta\beta}$ and refers to Supp Figure 4, but as far as I can tell, Supp Fig 4 plots $r_{\Delta\eta}$. So let me see if I have this right. Beta is a downward estimate of the “parental effect” (i.e., genetic nurture), which equals alpha when η_s (the sibling indirect genetic effect) = 0. And eta [with no subscript] is...?? Genetic nurture I take it? This all seems to need some work for consistency/clarity, and the authors need to have a table in the supplement with all the notations and their interpretations because it becomes mentally taxing to try to parse this, especially when the notations isn’t consistent between the text and the Supp. And main text should not refer to a figure that in turn refers to a table in the Supp that refers to quantities ($r_{\Delta\eta}$) never discussed in the main text.

We thank the reviewer for pointing out the confusion and inconsistency in our terminology. As we outline in the summary of changes, we have simplified the terminology so that we refer only to one set of parameters giving the direct effect, indirect sibling effect, and the paternal and maternal non-transmitted coefficients, and their average, along with the population effect. We have changed the notation in the Supplementary Note to be consistent with the main text, and we have include a table of terms at the start of the Supplementary Note where we introduce the models.

The relevant Supplementary Table (now Supplementary Table 5) now refers to $r_{\delta}(\alpha)$ as the correlation between direct effects and average NTCs, which is now described in detail in the main text and the methods. Furthermore, we have made sure all notation is consistent

between the main text, supplementary tables, supplementary figures, and supplementary note. We believe the new simulation results should help the reader in interpreting what we should expect these parameters to be under different simple scenarios.

Fig 6 – the EA/cog ability results seem awfully low, but even odder, why is the $r_{\Delta a}$ for height almost 1? Shouldn't assortative mating (ASM) reduce this? It would be nice to have what the expectations of $r_{\Delta a}$ are for observed levels of ASM.

Also, the LDSC estimates in Fig 6 rely upon an inherent assumption of no long-range LD (i.e., the regional LD score is all that's needed to capture the total association due to a causal variant). This is not true under ASM. Can the authors comment on how ASM might influence their LD score regression estimates?

We thank the reviewer for the suggestion of examining what to expect under assortative mating. As we outline in the summary of changes, we have included new simulation results that give expectations of these correlations for a phenotype affected by direct effects with random-mating heritability 0.5 and spousal phenotypic correlation 0.5 going to equilibrium. For this trait, we observe a moderately high correlation between direct effects and average NTCs, and a high correlation between direct and population effects, similar to the empirical results for height.

We agree that the results for educational attainment and cognitive ability are striking, and they suggest that standard GWAS methodology that adjusts for PCs gives estimates of SNP effects that are substantially biased. A model of direct effects and assortative mating cannot explain the results for educational attainment and cognitive ability: both the simulation and height results imply that we should expect a high correlation between direct and population effects under this kind of model.

We also agree with the reviewer that the assumption of LDSC that there is no long-range LD is violated for traits affected by assortative mating and population stratification. In part because of this, we decided to change our estimator to one that does not attempt to model how marginal effects relate to joint-fit effects, or remove stratification (as LDSC does), but that simply estimates how correlated (marginal) direct effects are with (marginal) population effects, adjusting for sampling error. Furthermore, the LDSC method removes some of the contribution of stratification to bias in population effects, which is not appropriate when the influence of stratification on GWAS summary statistics is of interest.

In the revision, we have expanded the set of SNPs we use, and we have updated the coding of EA to fix an error made in previous GWAS of EA, in addition to using a more appropriate

estimator for the genome-wide correlations. The estimate for the correlation between direct and population effects for EA is now somewhat higher than in the initial submission, and the estimate for cognitive ability somewhat lower, but qualitatively the results are similar:

Across the 9 phenotypes, $r_{\delta}(a)$ was not statistically distinguishable from 1 ($P > 0.05$, one-sided Z-test for $r_{\delta}(a) < 1$) except for EA ($r_{\delta}(a) = 0.739$, S.E.=0.086, $P=1.2 \times 10^{-3}$) and cognitive ability ($r_{\delta}(a) = 0.490$, S.E.=0.086, $P=1.6 \times 10^{-9}$) (Figure 6).

While the results for cognitive ability are rather extreme, they are supported by the new analyses we perform comparing the direct effects for cognitive ability to BOLT-LMM summary statistics from a large, independent sample of unrelated individuals, where we found a correlation of 0.706 (S.E.=0.110) for the correlation between direct effects in the related sample and BOLT-LMM summary statistics in the unrelated sample. We also observe statistically significant differences between correlations between BOLT-LMM summary statistics and direct and population effects on cognitive ability for 6 out of 9 phenotypes, as we describe in the Results:

In order to compare our results to summary statistics from an independent sample, we performed a GWAS in a sample of 276,419 unrelated individuals from the 'White British' subsample of the UKB using BOLT-LMM³² (Methods). Comparing direct effects to BOLT-LMM summary statistics, we obtained results consistent with our analysis comparing direct to population effects within the related sample (Supplementary Table 5). We found that correlations between BOLT-LMM summary statistics and direct effects on EA and cognitive ability were attenuated compared to correlations between BOLT-LMM summary statistics and population effects on EA and cognitive ability (Figure 7 and Supplementary Table 7). We found statistically significant differences between the correlations with direct and population effects ($P < 0.05$, two-sided Z-test for non-zero difference) for 4 out of the 9 phenotypes and east birth coordinate for EA, and for 6 out of the 9 phenotypes for cognitive ability. The largest estimated difference was the correlation between BOLT-LMM summary statistics for EA and direct effects on cognitive ability (correlation=0.428, S.E. 0.070) and population effects on cognitive ability (correlation=0.747, S.E. 0.049), giving a difference of 0.319 (S.E. 0.073). This suggests either shared confounding between EA and cognitive ability, or shared IGEs that are not highly correlated with direct effects on cognitive ability, or both.

This shows that the difference between direct and population effects we observe in the related sample translates into different relationships with independent summary statistics for multiple other traits.

p. 13 – the para starting “In practice...” needs work for clarity. As I understand it, there are two sources of inflation of the “eta” (or beta or alpha!) terms that arise from ASM. One is due to imputation: imputed parental PGS’s don’t capture the inflation of variance due to ASM (the expectation that alleles unshared between parent and offspring are drawn at random from the population is incorrect). The second is that ASM induces a correlation between the nontransmitted PGS and the transmitted PGS. Is all this correct? I felt it needed to be spelled out more clearly, and also that the relative sizes of these biases needed to be discussed. I’m guessing the second is larger. The paragraph also seems out of place. Why not describe the results first and then caveats?

The reviewer is correct about the two sources of bias, and that we needed to be clearer about the different way assortative mating creates bias. We have rewritten this section to make this clearer:

In practice, linkage disequilibrium (LD) between some of the SNPs is expected. However, if many SNPs from across the genome contribute to the PGS, only a small fraction of the pairs of SNPs will have non-negligible correlations due to local LD, and the effect on the imputations and estimates would be negligible. However, for a phenotype with AM, contributing SNPs can become correlated regardless of their relative physical positions^{3,24,25}. Since each SNP is imputed individually without conditioning on other SNPs that contribute to the PGS, the imputed PGSs are not exactly the conditional expectations given the observed PGSs. Consider a model for the association between phenotype and a PGS:

$$Y_{i1} = \delta PGS_{i1} + \eta_s PGS_{i2} + \alpha_p PGS_{p(i)} + \alpha_m PGS_{m(i)} + \epsilon_{i1},$$

where PGS_{ij} is the PGS of sibling j in family i , $PGS_{m(i)}$ is the PGS of the mother in family i . We show in the Supplementary Note that using imputed parental PGSs in place of observed parental PGS does not introduce bias to estimates of δ and η_s , even when AM is present, when the number of SNPs, L , is large. However, a slight bias in estimates of NTCs is introduced (Supplementary Note). For example, if using parental genotypes imputed from sibling pairs with phased data, the estimate of the average NTC, $\alpha = (\alpha_p + \alpha_m)/2$, would be inflated by a factor of $(1 + r_{am})/(1 + r_{am}/2)$, where r_{am} is the equilibrium correlation between maternal and paternal PGS. We note that, even with fully observed genotypes, AM implies that α_p and α_m capture confounding due to the parental PGS being correlated with the genetic component of the phenotype that would be uncorrelated with the PGS under random-mating, as described previously^{4,26}.

We decided not to move the paragraph after the results because we wanted to make clear upfront that, even under AM, the estimates we obtain for the direct and indirect sibling effects of the PGS are unbiased. Instead, we have split the polygenic score analysis section in two: a first section that describes the theoretical results, which goes with the rest of the theoretical results; and a section that describes the empirical results, which follows the theoretical results.

Finally, if we have a closed-form solution for the bias that depends only on r (the mate correlation), the estimates should be able to be back-corrected (given that we know what r is in the UKB) so that they are unbiased by ASM, right? Why not do that? I suppose the answer is because we don't know whether equilibrium has been reached? But by comparing PGS's across mates to those across haplotypes within person (ala the Kong Science paper), we can know this, no?

Thank you for suggesting that we adjust the estimates. We did not do that in the initial submission because any such adjustment would necessarily be approximate. However, on reflection, an approximate adjustment is better than no adjustment at all. We added this to the paragraph describing the PGS results

We estimated that the correlation between maternal and paternal PGS is 0.114, indicating strong assortative mating. We therefore adjusted the NTC estimates for the bias introduced by imputation of parental PGS when assortative mating is present (Methods).

We added the following paragraph to the Methods:

We adjusted average NTC estimates for bias introduced by imputation when AM is present (Supplementary Note). The expected bias introduced depends on the data structure used to perform the imputation. In this case, parental genotypes were imputed from sibling genotypes (without observed parental genotypes) in 88.8% of the sample, and for 92.8% of those individuals, the parental genotypes were imputed from a single sibling pair. This suggests that the formula derived in the Supplementary Note for the bias in average NTC estimates derived from samples with parental genotypes imputed from sibling pairs should provide a good approximation, provided that AM has reached an approximate equilibrium. The formula implies that parental effect estimates are inflated by a factor of $(1 + r_{am}) / (1 + r_{am}/2)$, where r_{am} is the correlation between the maternal and paternal PGS at equilibrium. To compute r_{am} , we took advantage of the fact that the correlation between siblings' PGS values is equal to $(1 + r_{am})/2$ at equilibrium (see Nagylaki²⁵). We estimated the correlation between siblings' PGS to be 0.557, giving an estimate of 0.114 for r_{am} , implying that average NTC estimates are inflated

by a factor of around 1.054. We therefore divided the average NTC estimates by 1.054 to produce adjusted NTC estimates.

While it is hard to be sure that equilibrium assumptions hold approximately, we expect that the adjustment will be approximately correct even out of equilibrium conditions. This also shows that, even with strong AM leading to a high value for r_{am} , any inflation of estimates is likely to be slight.

We could not perform the type of adjustment performed in Kong et al. because we did not have access to large samples where parent-of-origin, and therefore paternal and maternal non-transmitted alleles, could be determined.

p. 13 - In that same paragraph: "Importantly, the inflation of parental effect estimates is smaller than when estimating parental effects without performing imputation." - unclear. You mean when estimating parental effects using PGS from nonimputed SNPs?

We have removed this since it was confusing. We are referring to the fact that not imputing does not allow you to escape bias induced by assortative mating. For example, if you regress phenotype onto PGS, you get an estimate of $\delta + (1 + r_{am})\alpha$. If you subtracted the direct effect from this, you get $(1 + r_{am})\alpha$, which is more biased than the estimate you get with imputation of parental PGS: $\frac{2(1+r_{am})}{2+r_{am}}\alpha$. We have moved discussion of this more subtle point to the Supplementary Note.

p. 14 - The same influences of stratification/ASM/genetic nurture are already baked into the GWAS estimates that the authors use to create the polygenic scores. Can the authors comment on how those influences affect the estimates shown in Figure 7?

Thank you for the suggestion. We have added a couple of sentences to the discussion on this:

The PGS was constructed from standard GWAS summary statistics, so the average NTCs of the PGS could reflect bias and/or IGEs in the original GWAS summary statistics. Future studies could examine prediction using PGS constructed from direct effect estimates, which do not have these biases.

p. 14 - 15 - The "parental influences" (i.e., parental indirect genetic effects) in Figure 7 for EA and IQ seem too high to have arisen from parental phenotype -> offspring phenotype effects. I see the authors make this same point on p. 15 but their treatment of this important issue is superficial. A starting place could be that, if we were to interpret this as being solely due to genetic nurture (rather than ASM or stratification),

what kind of parental phenotype -> offspring phenotype effect would that imply? Balbona et al (bioarxiv) discuss how to convert estimates of genetic nurture to total parental environmental influences given the r^2 of the polygenic score. Does that comport at all with previous estimate of parental influences from extended twin methods (e.g., Devlin, Daniels, Roeder, 1997)? And how much of this can be due to ASM given known spousal correlations?

We agree with the reviewer that these are all important points. However, our ability to address them satisfactorily within the bounds of this manuscript and the UK Biobank data is fairly limited.

We agree that the average NTCs are likely too large to be explained by parental IGEs alone. As described in Kong et al. 2018, these coefficients capture confounding due to correlation with the genetic component of the trait that would be uncorrelated with the PGS in a random-mating population. Given that AM appears to be strongly influencing the PGS here, it is likely that this phenomenon contributes substantially to the average NTC estimates. We are careful not to interpret this as solely due to parental IGEs in the discussion:

By applying the proposed method to an EA PGS, we observed a lower direct/(direct+average NTC) ratio for EA than was observed in Iceland⁴. This implies that the combined influence of IGEs, population stratification, and AM is stronger in the UK than in Iceland.

Disentangling these different influences is beyond the scope of this paper, and may involve obtaining unbiased estimates of parental IGEs through multi-generational models.

We thank the reviewer for the reference to Balbona et al. However, based on our understanding of the paper, we would need either parental phenotypes and/or estimates of the random-mating heritability in the UK to perform the estimation of overall parental influences. Neither of these are available based on UKB data. We inserted a reference to Balbona et al. at the end of the paragraph on PGS analysis with assortative mating since it explains the issue of having an incomplete/noisy PGS well:

We note that, even with fully observed genotypes, AM implies that α_p and α_m capture confounding due to the parental PGS being correlated with the genetic component of the phenotype that would be uncorrelated with the PGS under random-mating, as described previously^{4,32}.

The authors state that stratification may explain these findings. If this is true, it is extremely important, and therefore the authors really need to discuss this possibility more thoroughly. Why do they believe this? What evidence supports that view? Why wouldn't controlling for 10 or 20 PCs within a European-restricted sample eliminate this? (BTW, I cannot find where they discuss how many PCs were controlled for).

We agree that the role of stratification in explaining these findings is important, and we have performed several new analyses to further investigate this hypothesis, as outlined in the summary of changes. Our new analyses show that stratification persists in standard GWAS summary statistics after PC adjustment (see Supplementary Figure 7), even in a relatively homogeneous sample such as the UKB White British ancestry individuals, and that this stratification likely contributes to the low correlation between direct and population effects for EA and cognitive ability. Our simulation of a trait affected by stratification shows that, even after PC adjustment, stratification substantially lowers the correlation between direct and population effects. Perhaps this is not surprising since there is no theoretical reason to suppose that PC adjustment, in general, removes all stratification: see recent work such as Zaidi, A. A. & Mathieson, I. Demographic history mediates the effect of stratification on polygenic scores. *Elife* 9, e61548 (2020).

We apologise for not making clear that traits were adjusted for 40 principal components. We state that at the start of the results section 'Genome-wide association analyses for 9 phenotypes':

We used the sample of 39,619 individuals for which parental genotypes were observed or could be imputed (Table 4) to estimate the direct effects, NTCs, and population effects of 1,586,010 SNPs with MAF>1% on 9 phenotypes (Methods). Phenotypes were adjusted for 40 genetic principal components before SNP effects were estimated.

We also state it in the legend of Figure 6, and in the methods section 'UK Biobank phenotypes'.

If there is residual stratification remaining in the data for EA or IQ, such that there are mean differences in phenotype across genetic clusters, doesn't this also lead to long-range directional LD in the same way that ASM does? If so, doesn't this lead to many of the same kinds of biases that ASM does? Can the authors discuss this?

Yes, we believe that the effects of population stratification and assortative mating are similar in some ways. The simulation results, however, show that stratification would tend to reduce the correlation between direct and population effects more than assortative mating, as we discuss in the interpretation of our results in the main text.

Are there any complete genotyped trio families (including probands, and their parents) in UKB? If there are, can the author compare the genetic analysis results from complete trio in UKB to those of Iceland consortium? Alternatively, can the authors mask the Icelandic data to mimic the UKB missingness patterns and see if results change and come more in line with the UKB data? If so, maybe there are some missed confounding effects in the current Mendelian Imputation method.

In agreement with the reviewer, we would like to compare our results directly to results from the Icelandic data. However, none of the authors of this study are currently affiliated with deCODE Genetics/Amgen, who own the data, and deCODE were not responsive to requests for collaboration. There are 832 individuals with both parents genotyped included in our PGS analysis in the UKB. However, this sample is too small to perform any kind of meaningful comparison to results from the rest of the sample used for PGS analysis.

To test if the imputation method is leading to some unknown confounding, we also estimated the direct effect of the PGS on EA by regressing phenotypic differences between siblings onto PGS differences between siblings, and we compared that to the 'population effect' you get from regression of phenotype onto PGS, and we obtained a ratio between direct and population effect of 0.52, which is very close to the 0.50 we obtained by performing regression with imputed parental PGS and comparing direct to (direct+average NTC). Therefore, we see no evidence for an unknown bias of meaningful magnitude affecting our analyses with imputed PGS.

In Figure 7, most of estimations of indirect genetic effects from siblings (τ) are around 0. Thus, maybe displaying the estimates without τ (providing less noisy results) would be preferable and then place the figure with τ , and differentiating father vs. mother effects, in the Supp.

Thank you for the suggestion. We have swapped the figures around to show the more precise estimates of direct and average NTCs in the main text (Figure 5) and the Figure with the indirect sibling effects in the Supplement (Supplementary Figure 6). We cannot display the maternal-paternal estimates on the same plot as indirect sibling effect estimates since they are derived from the analysis assuming the indirect sibling effect is zero, with a larger sample size and more precise estimates.

Figure 5. Decomposition of the EA polygenic score effect assuming no indirect sibling effects. The standardized effect estimate (SD change in phenotype per SD change in PGS) is given along with the 95% confidence interval. Estimates were derived from a sample 39,619 individuals with imputed and/or observed parental genotypes. Effects were estimated by joint regression of individuals' phenotypes onto their own PGS, and their mother's and father's (imputed/observed) PGS (Methods). We give the average of the estimated maternal and paternal non-transmitted coefficients (NTCs), adjusted for bias due to imputation in the presence of assortative mating (Methods), as the 'average NTC' here, and the difference between maternal and paternal NTCs as 'maternal-paternal'. Phenotypes were adjusted for 40 genetic principal components prior to analysis. Abbreviations: HDL, high density lipoprotein cholesterol; BMI, body mass index; FEV1, forced expiratory volume in one second.

Supp, p. 1 - if authors intend for the supplement to be able to be read stand-alone (which it appears to be the case, and I think is wise), then they need to introduce each new term as it comes. E.g., beta is introduced on p.1 without explanation. And it's not clear why authors use η_p and η_m in equations 1 and 2 but these are never used in the main text (they are beta's and alpha's there). I take it that η_p in the supp is the same as η_p in an earlier Science paper by these authors while α_p in the main text encompasses both η_p plus an effect due to stratification and/or assortative mating. But this needs to be clearer.

Thank you for pointing out the inconsistencies. We have rewritten the first section of the Supplement to introduce the models in a way that is consistent with the main text, along with a table of terms.

Supp p. 26+ - what does the “I” stand for in “PGI”? Imputed? Is another term really necessary? I thought PGS_hat was the imputed PGS.

Thank you for noticing this. This was an error due to using notation from a different project where we refer to ‘polygenic indices’ rather than ‘polygenic scores’. We have modified the supplement to refer to ‘PGS’ everywhere, to be consistent with the main text.

The authors seek to estimate gene-environment correlation and heritability without bias from imputed genotypes. They provided some proofs of unbiasedness of their methods in the Supp and it seems correct. However, it is hard to understand how their imputed methods could increase the power with their own derivation in Supp. section 2.3. In their equation (30), if the variance of imputed genotype is much smaller than that of observed genotypes, the variances of estimators would be larger. Thus, the authors should provide results (e.g., from simulation) comparing the variances of estimators across different missingness structures.

We are not claiming that our imputation method produces unbiased estimators with smaller variance than estimators using fully observed parental genotypes, which is not possible. We are claiming that our imputation method increases power for estimation of direct effects and non-transmitted coefficients compared to not performing imputation (for example, regression of phenotypic differences between siblings onto genetic differences) when parental genotypes are unobserved. We derive the variances of the estimators in Supplementary Section 4.2, and they are summarized in a table there. We display the relative efficiencies of the different methods (no imputation, imputation without IBD, with IBD and unphased data, with IBD and phased data) in Figure 4 in the main text, which shows that the imputation method always beats not performing imputation, and that imputation with phased data is always the best.

Supporting these theoretical results, we performed simulations with different levels of phenotypic correlation between siblings, and we confirmed the theoretical results are accurate. We added text to the Results section ‘Estimating effects using genotypes imputed from siblings’:

We confirmed the theoretical result using simulated data (Methods and Supplementary Figure 3).

Supplementary Figure 3. Confirmation of theoretical result for direct effects. Here we compare theoretical predictions to results from simulated data for the effective sample size for estimation of direct genetic effects using parental genotypes imputed from both phased and unphased sibling genotype data relative to estimating direct genetic effects using genetic differences between siblings. For 3,000 independent families, we simulated three different traits affected by direct, paternal, and maternal effects of 1,000 SNPs with minor allele frequency 0.5 (Methods). The overall variance explained by the combined direct, paternal, and maternal effects varied between the simulations, leading to different correlations between siblings' phenotypes. We computed theoretical expectations based on formulae derived in the Supplementary Note, which are drawn as the red curve (theoretical expectation for imputation from unphased data) and the black curve (theoretical expectation for imputation from phased data). The simulation results for unphased data are given by the black circles, and the simulation results for the phased data are given by black triangles. The relative effective sample size is given by the ratio between the sampling variance for estimation of direct effects when using differences between siblings to the sampling variance when using parental genotypes imputed from phased or unphased data.

For imputation from different data structures, such as parent-offspring pairs, we cannot compare to not performing imputation, since one cannot obtain unbiased estimates of direct effects without performing imputation. However, we show that our method has $\frac{1}{2}$ of the effective sample size compared to fully observed parental genotypes for estimation of direct effects when using phased data, and we derive the relative efficiency compared to full observed parental genotypes for different allele frequencies when using unphased data in the Supplementary Note, and we display this in Supplementary Figure 2:

Supplementary Figure 2. Relative effective sample size for estimating direct genetic effects when imputing a missing parent's genotype from a parent-offspring pair using unphased data. We derive that the relative effective sample is $\frac{1-3f(1-f)}{2-2f(1-f)}$ compared to using fully observed parental genotypes (Supplementary Note). There is a penalty for heterozygosity when using un-phased data since the allele that is shared with the observed parent cannot be determined when both parent and offspring are heterozygous. In contrast, the relative efficiency when using phased data is $1/2$, independent of allele frequency.

The parameter r in section 8 of supplementary notes seems confused with those of section 4. It would be better to change one of those parameter notations to reduce confusion.

Thank you for pointing this out. We have changed the name of that parameter to r_{am} , and we use this name in the main text and methods too.

In supplementary note pages 14 and 15, the authors described the variances of estimators with different type of methods. It is hard to determine which method has

smaller variances. Since $-1 < r < 1$, no imputation method has larger variances of estimators than those of IBD and phased method, overall. I think that authors need to mention that IBD imputation method always gives more efficient estimators than unimputed method.

Thank you for the suggestion. We have added such as sentence to the end of results section on 'Estimating effects using genotypes imputed from siblings':

Using parental genotypes imputed from phased data always results in estimates of direct effects and average NTCs with smaller variance than using unphased data or genetic differences between siblings (Supplementary Note).

Decision Letter, first revision:

8th Oct 2021

Dear Dr Young,

Your Technical Report, "Mendelian imputation of parental genotypes for genetic association analyses" has now been seen by 2 referees. You will see from their comments below that while they find your work of interest, some important points are raised. We are interested in the possibility of publishing your study in Nature Genetics, but would like to consider your response to these concerns in the form of a revised manuscript before we make a final decision on publication.

We therefore invite you to revise your manuscript taking into account all reviewer comments. Please highlight all changes in the manuscript text file. At this stage we will need you to upload a copy of the manuscript in MS Word .docx or similar editable format.

*2) If you have not done so already please begin to revise your manuscript so that it conforms to our Technical Report format instructions, available [here](http://www.nature.com/ng/authors/article_types/index.html). Refer also to any guidelines provided in this letter.

*3) Include a revised version of any required Reporting Summary:

[REDACTED]

We hope to receive your revised manuscript within eight to twelve weeks. If you cannot send it within this time, please let us know.

Sincerely,

Wei

Wei Li, PhD
Senior Editor
Nature Genetics
New York, NY 10004, USA
www.nature.com/ng

Reviewers' Comments:

Reviewer #1:

Remarks to the Author:

Overall the manuscript has improved significantly from the first submission. My primary concerns about contextualizing the advances in the paper and co-linearity have been adequately addressed. Clarification of the independence of the UKB 350K sample is required (see below), and there are some further issues raised around clarity.

- The authors chose to define a 'population effect' as the overall genotype-phenotype in the population. OK but this is not defined explicitly. Can I assume the 'population effects' are from a standard study design of unrelated individuals with the analysis of one SNP at a time? If so these SNP effects may have direct + population stratification (PS) + indirect (IGE) + assortative mating (AM) effects included in the estimate. From this point there are a range of experimental designs/types of analyses that provide conditional estimates and attempt to remove or reduce some of the cofounders (PS, IGE, AM). At the moment all these designs are treated equally/ambiguously but they include different effects in their estimates. It is not clear what the authors think (ideally or in practice) BLOT-LMM estimates. A clearer distinction of methods in terms of the various parameters the paper is estimating would help, especially when the authors' ultimate aim seems to be to 'tease apart' the relative contribution of all these different effects. For example:
 - a. Mixed linear model (e.g. BLOT-LMM) = AM + IGE + direct, i.e. it accounts for population stratification through the relationship matrix but not correlations between SNP (such as caused by AM) as it only fits one SNP at a time.
 - b. Multiple regression (e.g. COJO) = PS + IGE + direct.
 - c. Bayesian Multiple Regression (e.g. BayesR) = IGE + direct.
 - d. Within-family based study designs (e.g. SNIPar) = direct effect, NTC.
- P. 18, para 2. 'If parental IGEs are substantial and not highly correlated with direct genetic effects, then genetic prediction models could be improved by including polygenic predictors based on parental genotypes and NTCs'. This is not clear. Do you mean 'polygenic predictors of direct effects'? A polygenic predictor of an individual from a population (i.e. no relatives) based on 'population effects' (i.e. direct + IGE etc) could feasibly have a higher accuracy than one which only includes direct effects. This seems fine if the objective is to create the best predictor of phenotype. Problems arise when making inferences about these effects (e.g. heritability) or trying to apply them within families. It is already well established that using 'population effects' (as defined above) are not optimal for prediction, hence the development of other various methods (e.g. multiple regression).
- Can the authors make clearer which UKB sample they are referring to (line 4, para 2, pg. 13). 350K unrelated individuals seems like the number of unrelated individuals in the UKB. These individuals are unrelated to each other but would include one member/family, and therefore some of the individuals with sibs (for example) in these analyses. In the methods under 'GWAS in unrelated individuals' they refer to a sample of ~280K which excludes all individuals with a third-degree or closer relative – this sounds more appropriate but it is unclear if the correct sample was used. This smaller subsample seems to have been used for some analysis.

Minor comments / suggestions:

- Small typo 3rd last line abstract
- Missing 'common' environmental – line 4, para 1, pg. 3
- Remove 'For $r > 0$ ', line 4, para 5, pg. 8 and line 2, para 6. Awkward text.
- Also suggest topic sentences through this section to help guide reader; could move current paragraph 5 before para 4. At the moment it switches from estimating direct/sibling effects (para 2/3), then setting sibling effects to zero (para 4), then back to estimating direct/sibling effects (para 5).

Reviewer #2:

Remarks to the Author:

Review of Young et al resubmission

This is an impressive paper. The authors did a nice job in revising their MS based on reviewer feedback. It is clearer and reads much better now. I think they've adequately addressed the major concerns from the last reviews as well. I think it is close but I do have a few bigger picture comments I think the authors should consider. I follow these with several specific ones meant to help the writing/clarity (which the authors can heed or not as they see fit).

MAJOR COMMENTS:

1) The simulation performed in the UKB is an important addition but I see a few issues with it. Most importantly, the authors have simulated data following the setup of their own model; that's useful in its own right (to ensure the model isn't broken) but it can lead to false sense of security in how well the model actually performs in real situations. I think a major overhaul of the simulation approach is not called for here, but I do suggest the authors perform three additional simulations that might improve the realism and informativeness of the simulations, but I place these in order of importance because I do not feel strongly they all have to be done, esp. if they would add substantially to the turnaround time of the MS:

a) the simulation varying $r_{\Delta}(\alpha)$ is quite artificial. SNPs do not come in nature with two effects (direct and IGE) – the direct effects are the starting place and IGEs arise as a natural consequence of vertical (or horizontal) transmission. I think it's fine to do it the way the authors did in order to have control over $r_{\Delta}(\alpha)$, thereby ensuring that their model works well, but I think it is also important to do this in a realistic way, by simulating vertical transmission and letting $r_{\Delta}(\alpha)$ fall out however it may (similar to how the authors treated AM and population stratification). One reason this is important is that it would be nice to know if vertical transmission, like AM, typically leads to levels of $r_{\Delta}(\alpha)$ that aren't substantially low (e.g., $< .50$). If so, then low observations of $r_{\Delta}(\alpha)$ really point to population stratification as being the most plausible cause.

b) The authors later suggest that collider bias based on ascertainment of NTCs and direct effects could lead to otherwise puzzling results they observe ($r_{\Delta}(\alpha) < 0$). Perhaps the authors could simulate this to test if this is likely? As noted below, I'm not sure what realistic mechanism would lead to ascertainment on NTCs separate from ascertainment on deltas.

c) the authors do not simulate AM and vertical transmission (leading to IGEs) at the same time. These two effects (AM & vertical transmission) feed into one another and create a greater effect than

predicted from either of them alone. I think the authors should simulate both effects co-occurring.

So, I would suggest that, in addition to the 6 simulations in Supp Table 2 the authors have already conducted, they perform 3 final ones: (1) realistic vertical transmission; (2) realistic ascertainment leading to collider bias; (3) realistic vertical transmission + AM co-occurring.

2) p. 15-16: The paper by Zaidi & Mathieson that the authors refer to suggests that controlling for rare + common PCs can deal with stratification due to recent demographic events (unless they're highly localized). UKB has data imputed down to low frequencies. I recommend that the authors control for rare PCs in addition to common ones to see if the apparent effects of stratification are reduced.

3) In the authors' simulation, other than the artificial situation where $r_{\Delta}(\alpha)$ was set to 0, the lowest $r_{\Delta}(\alpha)$ that was observed was .865 – and this was for what I think would be strong population stratification (center effects $r^2 = .30$). However, for cognitive ability, authors observe a value of $r_{\Delta}(\alpha)$ substantially lower than this ($\sim .49$). What are the likely causes for this? Even larger stratification effects (which seems unlikely)?

4) Along these lines, (and also along the lines of doing the simulation (a) in point 1): what processes would lead to a low (e.g. $\leq .50$) or negative value of $r_{\Delta}(\alpha)$? What processes could lead to a highly significant cognitive ability $r_{\Delta}(\alpha) = -.58$ (SE = .094)? Can authors discuss the possible causes for this in a bit more depth? The authors suggest collider bias whereby both NTCs and direct effects are correlated with the ascertained trait. I'm trying to wrap my head around this. So NTCs and direct effects both influence some trait more or less independently because they have to be low to begin with in order for $r_{\Delta}(\alpha)$ to eventually go < 0 . Then strong ascertainment on this trait creates a collider bias/negative $r_{\Delta}(\alpha)$. Do I have that right? Can the authors unpack this a bit, perhaps trying to provide some perspective on how big this ascertainment would have to be to lead to the observed results (or perhaps choosing to do simulation 1b suggested above)? Or are there other types of model misspecification that could lead to these negative estimates?

MINOR COMMENTS:

1) p. 3 – I greatly appreciate the authors taking seriously the issue about what to call the effects of alpha from my comments in the previous review. NTC is preferable to “parental effects” from the last time but it’s statistical parlance and not very descriptive of the mechanism(s). That aside, the authors can define terms as they’d wish, but when they first introduce NTC (or perhaps earlier when they introduce IGE), I believe it is important that they describe how these terms are related to the earlier “genetic nurture” concept from Kong et al (2019), given that “genetic nurture” parlance has taken hold in the field. Interestingly, the term “genetic nurture” shows up nowhere in this paper. I understand it that NTC is the general term and that IGE is a component of this. Is “genetic nurture” then synonymous with IGE?

2) p. 3 – authors call η_s the “IGE” from siblings but alpha the “NTC”. I take it that authors are making an intentional point that η_s does not include the effects of population stratification and AM whereas alpha does. Should this be made explicit? I can buy this but it would be nice for the authors to explain why that's the case.

- 3) p. 4 – put the 3 equivalence classes in the order you'll cover them
- 4) Fig 2 is great but I do have a couple of minor suggestions – take or leave. Where “ $0+0+1+1=2$ ” etc are, it might be clarifying to say “ $g_{\text{par}}=0+0+1+1$ ” etc. and remove the “Imputed parental genotype” which is small and easily lost and instead put that g_{par} = “imputed parental genotype” in the caption. Perhaps g_{par} needs a hat over it here too?
- 5) Given that the Supplementary material is 50 pages long with 273 equations (!), it would be useful for authors to refer readers to the particular *section* of the Supp – or to specific equations if possible – rather than to just “Supplementary Note” throughout the main text.
- 6) For the only equation on p. 6, at first blush it seems to contradict the 3 equations on p. 5 and perhaps needs a bit more explanation. E.g., authors could note that from above, this is equal to either $(2+f)$ or $(1+f)$ depending on whether the unshared allele is 1 or 0, and that averaged across those two possibilities, the expectation is $1+2f$ (see Supp Note equation 45).
- 7) p. 6 – should it read “Given the proband’s MATERNALLY inherited allele,...” ?
- 8) Table 1 is very helpful. I would make the $E[\theta_{\text{hat}}]$ column of the same form as the middle column – i.e., put the parameters on the same line and separate them by commas to make it clear that the parameters on the right are respectively related to the predictors on the left.
- 9) I’m having a difficult time making the numbers in Table 2 and the text line up. For sibling pairs, are both siblings being counted as “probands”? But $19290*2 = 38580$. I’m not seeing a way to make that number align with the probands & sib data.
- 10) Fig 7 – it is awfully hard to tell the colors apart (e.g., “EA” vs. east) – consider using 11 different symbols for each trait as well as colors.
- 11) In Fig 7, I was expecting the correlation with education direct effects (the y-axis of Fig 7A) for EA to be around .75, as it is in Fig 6. Instead it appears to be $\sim .95$. I know they are independent estimates and that the SEs are wide, but it might be helpful to point out that these two are estimating the same quantity (if that's right).
- 12) Fig 7 – the labels on the x- and y-axes are not very descriptive – e.g., “Correlation with education population effects” – correlation between what and what? Suggest instead something like “correlation of a^{hat} for EA in main model with a^{hat} from BOLT-LMM for 9 traits” for the x-axis and “correlation of δ^{hat} for EA in main model with a^{hat} from BOLT-LMM for 9 traits” Similarly for the y-axis in Fig 6.
- 13) Related to the point above, in general, in both figures and text and supplement, I think authors should be re-using the terms they've already coined (e.g., $r_{\text{delta}}(a)$) rather than switching between the term and what it stands for (e.g., $r_{\text{delta}}(a)$ sometimes and “correlation between direct and population effects” at others). Alternatively, if they want to write out the text for what the term stands for, at least put in parentheses the term.

Author Rebuttal, first revision:

Response to reviews of ‘Mendelian Imputation of parental genotypes for genetic association analyses’

We thank the reviewers again for their helpful and insightful comments. We have addressed all the points raised by the reviewers, which has improved the clarity of the manuscript and enabled clearer interpretation of our empirical results. The revised manuscript has been provided with changes from the last version tracked.

In response to reviewer 2’s suggestion of additional simulations, we discovered a bug in the way we were simulating recombination in the original simulations. Although this bug did not appear to have much, if any, impact upon the simulation results, we decided to redo the simulations completely after fixing the bug. We changed the simulation design slightly to simplify analysis: after simulating parent and offspring generations, we set all parental genotypes to missing and imputed the parental genotypes from the sibling pairs in the offspring generation. We also increased the sample size for the assortative mating simulation from 50,000 to 100,000. In addition to redoing the existing simulations, we have added simulations with vertical transmission and vertical transmission and assortative mating (Supplementary Table 2). We have also used our simulations to assess what happens to the correlations between effects under ascertainment (Supplementary Table 9).

We simulated vertical transmission where the parental phenotype that affects the offspring phenotype is the same as the offspring phenotype. Under a single generation of vertical transmission, this is equivalent to a model with perfectly correlated direct and indirect genetic effects. However, vertical transmission leads to a dependence of proband phenotype on the phenotype of all of the proband’s ancestors. After several generations of vertical transmission, the phenotype distribution reaches an approximate equilibrium (see Cavalli-Sforza, L. L. & Feldman, M. W. Cultural versus biological inheritance: Phenotypic transmission from parents to children). We simulated 20 generations of vertical transmission with and without assortative mating to reach an approximate equilibrium and examine correlations between direct and population effects at equilibrium. The new simulation results show that vertical transmission alone does not lead to low correlations between direct and population effects, or negative correlations between direct effects and non-transmitted coefficients (NTCs). However, when vertical transmission and assortative mating are strong, our simulations show that the correlation between direct and population effects is substantially reduced, reaching 0.883 in our simulation. This implies that the combined forces of vertical transmission and assortative mating likely contribute to the low correlation we found between direct and population effects for educational attainment (EA) and possibly also for cognitive

ability. Our simulation results also show that ascertainment tends to lower correlations between direct and population effects, and between direct effects and NTCs, even producing a negative correlation between direct effects and NTCs when the true correlation in the unascertained sample is zero.

Beyond the simulations, we also examined the correlations for EA and cognitive ability after adjusting for principal components derived from the IBD relatedness matrix. We found that adjusting for the IBD-PCs significantly increased these correlations over adjusting for common variant PCs alone, implying the recent structure in the population not well-captured by PCs based on common variants partly explains the low correlations for EA and cognitive ability.

We are therefore able to conclude that a combination of vertical transmission, assortative mating, stratification due to recent structure, and ascertainment likely explain the low correlations for EA and cognitive ability.

Reviewers' Comments:

Reviewer #1:

Remarks to the Author:

Overall the manuscript has improved significantly from the first submission. My primary concerns about contextualizing the advances in the paper and co-linearity have been adequately addressed. Clarification of the independence of the UKB 350K sample is required (see below), and there are some further issues raised around clarity.

We are glad the reviewer thinks that the manuscript has improved significantly from the initial submission, and we thank the reviewer for their effort in helping us improve the clarity of our manuscript.

- The authors chose to define a 'population effect' as the overall genotype-phenotype in the population. OK but this is not defined explicitly. Can I assume the 'population effects' are from a standard study design of unrelated individuals with the analysis of one SNP at a time? If so these SNP effects may have direct + population stratification (PS) + indirect (IGE) + assortative mating (AM) effects included in the estimate. From this point there are a range of experimental designs/types of analyses that provide conditional estimates and attempt to remove or reduce some of the cofounders (PS, IGE, AM). At the moment all these designs are treated equally/ambiguously but they include different effects in their estimates. It is not clear what the authors think (ideally or in practice) BLOT-LMM estimates. A clearer distinction of methods in terms of the various parameters the paper is estimating would help, especially when the authors' ultimate aim seems to be to 'tease

apart' the relative contribution of all these different effects. For example:

- a. Mixed linear model (e.g. BLOT-LMM) = AM + IGE + direct, i.e. it accounts for population stratification through the relationship matrix but not correlations between SNP (such as caused by AM) as it only fits one SNP at a time.
- b. Multiple regression (e.g. COJO) = PS + IGE + direct.
- c. Bayesian Multiple Regression (e.g. BayesR) = IGE + direct.
- d. Within-family based study designs (e.g. SNIPar) = direct effect, NTC.

We thank the reviewer for highlighting the need for greater clarity on what the population effect captures under different regression designs. We agree that different methods, such as BOLT-LMM, or principal component adjustment, will change the population effect by changing the relative contribution of different factors to the NTC. We expect that, in general, all methods that do not control for parental genotype will pick up indirect genetic effects as part of the population effect. In addition, while linear mixed models (LMMs) or Bayesian Multiple Regression will remove some of the confounding due to assortative mating, unless the model captures all of the heritability, including from rare variants, confounding due to assortative mating will remain. We have added some sentences to the end of paragraph 2 of the introduction to clarify this:

By modelling the effects of many genome-wide SNPs, LMMs can reduce confounding due to AM⁵. However, unless all of the heritability is captured, some confounding due to AM will remain. Depending on the regression design (PC adjustment, LMM, LMM & PC adjustment, etc.), the amount of confounding due to population stratification and AM can differ^{5,8}.

We also define the population effect in the paragraph following model (1):

We refer to α_p and α_m as 'non-transmitted coefficients' (NTCs), since they are the expected coefficients on the alleles not transmitted to the proband in a regression of proband phenotype on proband genotype and non-transmitted alleles⁴. The NTCs capture IGEs from relatives, in addition to confounding due to population stratification and AM^{3,4}. The residuals ϵ_{i1} and ϵ_{i2} are uncorrelated with the genotypes of the siblings and parents, but may be correlated with each other. Note that standard GWAS methods that regress proband phenotype onto proband genotype estimate the 'population effect': $a = \delta + (\alpha_p + \alpha_m)/2$, which is the direct effect plus the average NTC, $\alpha = (\alpha_p + \alpha_m)/2$.

• P. 18, para 2. 'If parental IGEs are substantial and not highly correlated with direct genetic effects, then genetic prediction models could be improved by including polygenic predictors based on parental genotypes and NTCs'. This is not clear. Do you

mean 'polygenic predictors of direct effects'? A polygenic predictor of an individual from a population (i.e. no relatives) based on 'population effects' (i.e. direct + IGE etc) could feasibly have a higher accuracy than one which only includes direct effects. This seems fine if the objective is to create the best predictor of phenotype. Problems arise when making inferences about these effects (e.g. heritability) or trying to apply them within families. It is already well established that using 'population effects' (as defined above) are not optimal for prediction, hence the development of other various methods (e.g. multiple regression).

We agree that this sentence was not clear. The point we were trying to make was that if the true model is a direct effect component (a function of the proband genotype) plus and indirect genetic effect component (a function of the parental genotype), and direct and indirect effects are not perfectly correlated, then a prediction model that uses separate direct and indirect genetic effect estimates to predict separately the direct and indirect effect components (using both proband and parental genotypes) should, in theory, be able to outperform methods that use only population effects and proband genotypes, or only direct effects and proband genotypes.

To make this clearer, we have modified the sentence:

If parental IGEs are substantial and imperfectly correlated with direct genetic effects, then genetic prediction models could be improved by including polygenic predictors based on the indirect genetic effects of parental genotypes in addition to polygenic predictors based on the direct effects of proband genotypes.

This issue is somewhat different to the issue of how best to translate summary statistics from marginal regressions into the best polygenic predictors, where one has to deal with local LD. The issue of local LD applies whether one uses marginal direct effect estimates or marginal population effect estimates. The point we are making here is that separating out the direct and indirect effects could, in theory, help improve prediction models when those (true) effects are not perfectly correlated.

- Can the authors make clearer which UKB sample they are referring to (line 4, para 2, pg. 13). 350K unrelated individuals seems like the number of unrelated individuals in the UKB. These individuals are unrelated to each other but would include one member/family, and therefore some of the individuals with sibs (for example) in these analyses. In the methods under 'GWAS in unrelated individuals' they refer to a sample of ~280K which excludes all individuals with a third-degree or closer relative – this sounds more appropriate but it is unclear if the correct sample was used. This smaller subsample seems to have been used for some analysis.

We apologize for the confusion here, which is due to the fact that we used two different procedures for selecting an independent, unrelated sample: one for generating summary statistics for the polygenic score, and one for performing BOLT-LMM GWAS. The summary statistics for the polygenic score were produced by modifying the code from the meta-analysis used to produce the summary statistics released as part of Lee et al. 2018. For this, as we state in the Methods section: “We used summary statistics from a GWAS of EA⁵ modified to remove the individuals in this study and their relatives, up to the third-degree, from the summary statistics.” This resulted in a modified version of the meta-analysis used in Lee et al. where the UKB sample is reduced to ~350k individuals unrelated (<3rd degree) to the related sample used in the main analysis (but still including some other related pairs not related to the sample in our main analysis), the summary statistics of which were used to construct a polygenic score using LD-pred.

For the analysis using BOLT-LMM, which was done at a later date in response to reviews, we used a different approach: the UK Biobank provided an indicator variable in the sample QC file for whether an individual was present in the kinship table distributed as part of the UK Biobank data release. We restricted the sample used for the BOLT-LMM analysis to those samples that were not in the kinship table provided by UKB, implying they have no third degree or higher relatives in the genotyped UKB sample. This is a smaller sample than the ~350k used in the meta-analysis since it does not include any individuals with genotyped relatives (3rd degree or higher), whether the relative was used in our main analysis sample or not. We appreciate in hindsight that it would have been more consistent to use the same sample for the BOLT-LMM analysis as was used to compute the summary statistics for the polygenic score. However, both samples are unrelated (<3rd degree) to the related sample used for the main analysis in the paper, so there should not be any issues with dependence with the results from the main analysis.

Minor comments / suggestions:

- Small typo 3rd last line abstract

Fixed. Thank you.

- Missing ‘common’ environmental – line 4, para 1, pg. 3

If you mean the sentence “Sibling genotypes are conditionally independent of environmental effects given parental genotypes.”, then I don’t think ‘common environmental’ effects is accurate. Since sibling genotypes are randomly assigned given parental genotypes, they are independent of both environmental effects that are shared between siblings and those that are not.

- Remove 'For $r > 0$ ', line 4, para 5, pg. 8 and line 2, para 6. Awkward text.

Thank you for the suggestion. We have removed the text.

- Also suggest topic sentences through this section to help guide reader; could move current paragraph 5 before para 4. At the moment it switches from estimating direct/sibling effects (para 2/3), then setting sibling effects to zero (para 4), then back to estimating direct/sibling effects (para 5).

Thank you for the suggestions. We have made some modifications to this part of the manuscript to improve clarity.

Reviewer #2:

Remarks to the Author:

Review of Young et al resubmission

This is an impressive paper. The authors did a nice job in revising their MS based on reviewer feedback. It is clearer and reads much better now. I think they've adequately addressed the major concerns from the last reviews as well. I think it is close but I do have a few bigger picture comments I think the authors should consider. I follow these with several specific ones meant to help the writing/clarity (which the authors can heed or not as they see fit).

We thank the reviewer for the kind words and we are glad that we managed to address the concerns from the first round of reviews.

MAJOR COMMENTS:

1) The simulation performed in the UKB is an important addition but I see a few issues with it. Most importantly, the authors have simulated data following the setup of their own model; that's useful in its own right (to ensure the model isn't broken) but it can lead to false sense of security in how well the model actually performs in real situations. I think a major overhaul of the simulation approach is not called for here, but I do suggest the authors perform three additional simulations that might improve the realism and informativeness of the simulations, but I place these in order of importance because I do not feel strongly they all have to be done, esp. if they would add substantially to the turnaround time of the MS:

a) the simulation varying $r_{\Delta}(\alpha)$ is quite artificial. SNPs do not come in nature with two effects (direct and IGE) – the direct effects are the starting place and IGEs

arise as a natural consequence of vertical (or horizontal) transmission. I think it's fine to do it the way the authors did in order to have control over $r_{\Delta}(\alpha)$, thereby ensuring that their model works well, but I think it is also important to do this in a realistic way, by simulating vertical transmission and letting $r_{\Delta}(\alpha)$ fall out however it may (similar to how the authors treated AM and population stratification). One reason this is important is that it would be nice to know if vertical transmission, like AM, typically leads to levels of $r_{\Delta}(\alpha)$ that aren't substantially low (e.g., $< .50$). If so, then low observations of $r_{\Delta}(\alpha)$ really point to population stratification as being the most plausible cause.

b) The authors later suggest that collider bias based on ascertainment of NTCs and direct effects could lead to otherwise puzzling results they observe ($r_{\Delta}(\alpha) \ll 0$). Perhaps the authors could simulate this to test if this is likely? As noted below, I'm not sure what realistic mechanism would lead to ascertainment on NTCs separate from ascertainment on deltas.

c) the authors do not simulate AM and vertical transmission (leading to IGEs) at the same time. These two effects (AM & vertical transmission) feed into one another and create a greater effect than predicted from either of them alone. I think the authors should simulate both effects co-occurring.

So, I would suggest that, in addition to the 6 simulations in Supp Table 2 the authors have already conducted, they perform 3 final ones: (1) realistic vertical transmission; (2) realistic ascertainment leading to collider bias; (3) realistic vertical transmission + AM co-occurring.

We agree with the reviewer that vertical transmission is a more realistic model. We have added simulations with vertical transmission with and without assortative mating. Since vertical transmission, even without assortative mating, creates a dynamical system that tends towards equilibrium (Cavalli-Sforza and Feldman, 1973), we have simulated vertical transmission (for 20 generations, reaching an approximate equilibrium) both with and without assortative mating.

We have added the results of these two new simulations to Supplementary Table 2. We note that the estimation of direct genetic effects remained unbiased for the additional simulations. We discuss these results in paragraph 4 of the Results section 'Genome-wide association analyses for 9 phenotypes':

We estimated $r(\delta, a)$ and $r(\delta, \alpha)$ for the phenotypes simulated from genetic data from the UKB 'White British' subsample (Methods and Supplementary Table 2). For phenotypes affected by direct effects and parental IGEs in a random-mating population,

$r(\delta, \alpha)$ is the correlation between direct effects and average parental IGEs, which our simulation results confirmed. A plausible model for indirect genetic effects is vertical transmission²⁹, where a phenotype of the parent(s) affects the phenotype of the offspring through the environment. We simulated a phenotype where the phenotype of the offspring is determined by the direct effect of its own genotype, and the phenotype of its parents, in addition to a random environmental component, and we iterated this for 20 generations of random-mating, reaching an approximate equilibrium (Methods). For this phenotype, $r(\delta, \alpha)$ was 0.953 (S.E.=0.009) (Supplementary Table 2). When there is population stratification or AM, average NTCs (and therefore population effects) capture effects due to other genetic and environmental factors with which the SNP is correlated due to non-random mating, in addition to IGEs from relatives. We simulated 20 generations of AM for the same vertical transmission phenotype model (Methods), reaching an approximate equilibrium. For this phenotype, $r(\delta, \alpha)$ was 0.883 (S.E.=0.009). For a phenotype affected by direct effects and a random environmental component (no indirect effects or vertical transmission), $r(\delta, \alpha)$ was 0.949 (S.E.=0.008) after 20 generations of assortative mating. We also simulated a phenotype affected by direct effects and population stratification (Methods), for which $r(\delta, \alpha)$ was 0.917 (S.E.=0.007). These results imply that population stratification, assortative mating, and vertical transmission, along with their interactions, can lead to $r(\delta, \alpha)$ substantially below 1.

We have also modified the discussion to reflect what we have learned from the new simulation results, adding a sentence “Our simulation results (Supplementary Table 2) suggest that a combination of vertical transmission and assortative mating^{30,40} also contribute to the low correlation between direct and population effects.”

In relation to the reviewer’s point about ascertainment bias, we agree with the reviewer that suggesting this without providing any support was incomplete. Although we believe a complete treatment of ascertainment is beyond the scope of this paper, we have now added ascertainment to the simulated phenotypes with direct and indirect genetic effects with varying levels of correlation (0, 0.5, and 1). We simulated ascertainment by analysing only the upper half of the phenotype distribution for the simulated phenotypes. We added Supplementary Table 9, which compares the estimated correlations between direct effects and average NTCs and between direct and population effects for these traits before and after ascertainment. Ascertainment appeared to reduce the correlation between direct effects and average NTCs, consistent with the collider bias hypothesis. In particular, for the phenotype with uncorrelated direct and indirect genetic effects, ascertainment results in a negative correlation. We have added a paragraph on these results to the discussion:

Another phenomenon that may contribute is ascertainment: If direct effects and NTCs are not very strongly correlated, but both are also correlated with ascertainment, then collider bias⁴¹ could push the correlation estimate in the negative direction (as observed for cognitive ability and neuroticism) and reduce correlations between direct and population effects. Analysis of simulated phenotypes under ascertainment support this hypothesis, where strong ascertainment reduced $r(\delta, \alpha)$ to -0.264 (S.E.=0.091) for the phenotype with uncorrelated direct effects and parental IGEs (true $r(\delta, \alpha) = 0$). This may explain why we observed negative $r(\delta, \alpha)$ for cognitive ability and neuroticism, since observations for these phenotypes are ascertained for higher education and lower neuroticism⁴².

2) p. 15-16: The paper by Zaidi & Mathieson that the authors refer to suggests that controlling for rare + common PCs can deal with stratification due to recent demographic events (unless they're highly localized). UKB has data imputed down to low frequencies. I recommend that the authors control for rare PCs in addition to common ones to see if the apparent effects of stratification are reduced.

We thank the reviewer for this suggestion. Zaidi & Mathieson suggest that PCs extracted from IBD sharing can also detect recent structure. Due to the lack of whole genome-sequence data (at the time of doing the analysis) and the relative ease of detecting IBD segments, we decided to use the IBD approach suggested by Zaidi & Mathieson, rather than the rare variant approach. We have added a paragraph to the Results describing the outcome of this analysis:

PCs based on rare variants or IBD sharing capture recent structure better than PCs based on common variants¹⁰, which we used to adjust the phenotypes in our primary analysis. In order to better adjust for recent structure, we adjusted EA and cognitive ability for the top 40 PCs of the IBD relatedness matrix in addition to the 40 common variant PCs used originally (Methods). This increased the estimated correlation for EA to $r(\delta, \alpha) = 0.785$ (S.E.=0.076), an increase of 0.048 (S.E.=0.019; $P=5.7 \times 10^{-3}$ from a one-sided Z-test for an increase); and increased the estimated correlation for cognitive ability to $r(\delta, \alpha) = 0.621$ (S.E.=0.058), an increase of 0.131 (S.E.=0.041; $P=6.5 \times 10^{-4}$).

Using the PCs of the IBD relatedness matrix did improve control for population stratification, especially for cognitive ability. This suggests that recent structure in the UK population contributes to population stratification for EA and cognitive ability, which we now mention in the discussion: "We found evidence that residual stratification that persists after adjustment for common variant PCs contributes to this, and that some of this

stratification is likely due to recent structure in the population that is better captured by PCs of the IBD relatedness matrix¹⁰.”

However, we note that the correlations even after adjusting for IBD based PCs were still not that close to 1, indicating residual stratification may still be present.

3) In the authors' simulation, other than the artificial situation where $r_{\Delta}(\alpha)$ was set to 0, the lowest $r_{\Delta}(\alpha)$ that was observed was .865 – and this was for what I think would be strong population stratification (center effects $r^2 = .30$). However, for cognitive ability, authors observe a value of $r_{\Delta}(\alpha)$ substantially lower than this (~.49). What are the likely causes for this? Even larger stratification effects (which seems unlikely)?

4) Along these lines, (and also along the lines of doing the simulation (a) in point 1): what processes would lead to a low (e.g. $\leq .50$) or negative value of $r_{\Delta}(\alpha)$? What processes could lead to a highly significant cognitive ability $r_{\Delta}(\alpha) = -.58$ (SE = .094)? Can authors discuss the possible causes for this in a bit more depth? The authors suggest collider bias whereby both NTCs and direct effects are correlated with the ascertained trait. I'm trying to wrap my head around this. So NTCs and direct effects both influence some trait more or less independently because they have to be low to begin with in order for $r_{\Delta}(\alpha)$ to eventually go < 0 . Then strong ascertainment on this trait creates a collider bias/negative $r_{\Delta}(\alpha)$. Do I have that right? Can the authors unpack this a bit, perhaps trying to provide some perspective on how big this ascertainment would have to be to lead to the observed results (or perhaps choosing to do simulation 1b suggested above)? Or are there other types of model misspecification that could lead to these negative estimates?

Combined response to 3) and 4): adjusting cognitive ability for PCs from the IBD relatedness matrix increased the direct to population correlation to 0.621 (S.E.=0.058) and the direct to average NTC correlation to -0.318 (S.E.=0.120). This is still lower than the correlation observed in our simulations for a phenotype affected by stratification, and the direct to average NTC correlation remains negative. We agree that it is unlikely that stratification alone can explain these results. There can be many factors contributing to the results and interacting with each other: stratification, assortative mating, vertical transmission from an imperfectly correlated parental phenotype, and ascertainment. We have shown that residual stratification not captured by common variant PCs contributes, and we've shown in simulations that ascertainment can push the direct to NTC correlation in the negative direction. However, giving a complete accounting of all the factors leading to a low observed correlation is beyond the scope of this paper and is probably not possible in general. The fact that the results for height fit our expectations for a trait affected by direct effects and assortative mating indicates

that, whatever is going on for the cognitive ability phenotype in UKB, it is not some general issue with our method applied to traits affected by assortative mating (as cognitive ability is), but something specific to this phenotype that can be further investigated in future work.

MINOR COMMENTS:

1) p. 3 – I greatly appreciate the authors taking seriously the issue about what to call the effects of alpha from my comments in the previous review. NTC is preferable to “parental effects” from the last time but it’s statistical parlance and not very descriptive of the mechanism(s). That aside, the authors can define terms as they’d wish, but when they first introduce NTC (or perhaps earlier when they introduce IGE), I believe it is important that they describe how these terms are related to the earlier “genetic nurture” concept from Kong et al (2019), given that “genetic nurture” parlance has taken hold in the field. Interestingly, the term “genetic nurture” shows up nowhere in this paper. I understand it that NTC is the general term and that IGE is a component of this. Is “genetic nurture” then synonymous with IGE?

Thank you for the suggestion. We have now added a reference to the genetic nurture term in the second paragraph of the introduction where we define indirect genetic effects:

Indirect genetic effects (IGEs) – effects of genetic variants in one individual that affect the phenotype of another through the environment – from relatives, such as parents (genetic nurture⁴), also contribute to population effect estimates.

We wanted to talk about indirect genetic effects from relatives more generally than genetic nurture, which more properly applies to indirect genetic effects from parents to offspring alone. This modification specifies the difference we want to make in use of terminology.

2) p. 3 – authors call η_s the “IGE” from siblings but alpha the “NTC”. I take it that authors are making an intentional point that η_s does not include the effects of population stratification and AM whereas alpha does. Should this be made explicit? I can buy this but it would be nice for the authors to explain why that’s the case.

In the text following model (2), we state:

Since both proband and sibling genotypes are conditionally independent of environment given parental genotype, estimates of δ and η_s from fitting model (2) are unbiased⁴.

And we give a reference to Kong et al. 2018, which discusses this in more detail.

3) p. 4 – put the 3 equivalence classes in the order you’ll cover them

Thank you for the suggestion, which we have followed.

4) Fig 2 is great but I do have a couple of minor suggestions – take or leave. Where “0+0+1+1=2” etc are, it might be clarifying to say “g_par=0+0+1+1” etc. and remove the “Imputed parental genotype” which is small and easily lost and instead put that g_par = “imputed parental genotype” in the caption. Perhaps g_par needs a hat over it here too?

Thank you for the helpful suggestion, which we have followed:

5) Given that the Supplementary material is 50 pages long with 273 equations (!), it would be useful for authors to refer readers to the particular *section* of the Supp – or to specific equations if possible - rather than to just “Supplementary Note” throughout the main text.

Thanks for the helpful suggestion. We’ve gone through and referenced the relevant Supplementary Note (sub)-sections throughout the manuscript.

6) For the only equation on p. 6, at first blush it seems to contradict the 3 equations on p. 5 and perhaps needs a bit more explanation. E.g., authors could note that from

above, this is equal to either (2+f) or (1+f) depending on whether the unshared allele is 1 or 0, and that averaged across those two possibilities, the expectation is 1+2f (see Supp Note equation 45).

Thank you for the suggestion. We have modified the paragraph to make it clearer what is going on here and referenced the relevant subsection of the Supplementary Note:

When in IBD1 (the siblings share one allele IBD), the alleles not shared are known unless both siblings are heterozygous. When both are heterozygous, utilizing information from neighbouring SNPs through phasing can be used to resolve the uncertainty (Figure 3). However, imputation can proceed by averaging over the two possibilities (shared allele is '0' vs shared allele is '1'), giving

$$E[g_{\text{par}(i)} | g_{i1} = 1, g_{i2} = 1, \text{IBD}_i = 1] = 1 + 2f,$$

at the cost of lower correlation with the unobserved parental genotype (Supplementary Note Section 3.2).

7) p. 6 – should it read “Given the proband’s MATERNALLY inherited allele,...” ?

Here, we meant that if we know which of the proband’s alleles was inherited from the father, we have determined half of the father’s alleles. We have modified the sentence to make this clearer:

Given the proband’s paternally inherited allele, one half of the paternal genotype is determined

8) Table 1 is very helpful. I would make the E[theta_hat] column of the same form as the middle column – i.e., put the parameters on the same line and separate them by commas to make it clear that the parameters on the right are respectively related to the predictors on the left.

Thank you for a good suggestion, which we have followed. Table 1 now looks like:

Observed genotypes	Y_{i1} regressed on	$E[\hat{\theta}]$
Proband (Fig 1h)	g_{i1}	$\delta + (\alpha_p + \alpha_m + \eta_s)/2$
Sibling pairs (Fig 1e)	g_{i1}	δ
	g_{i2}	η_s

	$\hat{g}_{\text{par}(i)}$	$(\alpha_p + \alpha_m - \eta_s)/2$
Father-child pairs (Fig 1c)	g_{i1}	δ
	$g_{p(i)}$	α_p
	$\hat{g}_{m(i)}$	α_m
Mother-child pairs (Fig 1d)	g_{i1}	δ
	$\hat{g}_{p(i)}$	α_p
	$g_{m(i)}$	α_m
Trios (Fig 1b)	g_{i1}	δ
	$g_{p(i)}$	α_p
	$g_{m(i)}$	α_m
Quads (Fig 1a)	g_{i1}	δ
	g_{i2}	η_s
	$g_{p(i)}$	$\alpha_p - \eta_s/2$
	$g_{m(i)}$	$\alpha_m - \eta_s/2$

9) I'm having a difficult time making the numbers in Table 2 and the text line up. For sibling pairs, are both siblings being counted as "probands"? But $19290 * 2 = 38580$. I'm not seeing a way to make that number align with the probands & sib data.

This is because the number of pairs is not exactly the same as one half of the number of individuals with genotyped siblings. If you have a sibship of size n , then there are $n(n-1)/2$ pairs, so there is no way to directly convert number of pairs to number of probands without conditioning on the numbers of sibships of each size. To make this clearer, we now state

There were 19,290 sibling pairs from 17,289 sibships including 913 sibships of size greater than 2, with a maximum size of 6.

10) Fig 7 – it is awfully hard to tell the colors apart (e.g., "EA" vs. east) – consider using 11 different symbols for each trait as well as colors.

Thanks for the suggestion. We have updated the figure to use different symbols as well as different colours. We used a set of 4 simple shapes rather than a larger set of more complex shapes that may be hard to distinguish. Although the shapes are repeating, the fact each phenotype has a unique shape-colour representation helps in visually processing the plot.

11) In Fig 7, I was expecting the correlation with education direct effects (the y-axis of Fig 7A) for EA to be around .75, as it is in Fig 6. Instead it appears to be ~.95. I know

they are independent estimates and that the SEs are wide, but it might be helpful to point out that these two are estimating the same quantity (if that's right).

We do not think that they are quite independent estimates since the direct effect summary statistics are the same for the two estimates. We may expect that BOLT-LMM population effects to have somewhat lower confounding due to assortative mating and population stratification, so this could explain the higher estimate, but with the size of the standard errors, it is impossible to draw this conclusion with any confidence. With the notation change described below, we make the connection between the quantities in Figure 7 and Figure 6 clear.

12) Fig 7 – the labels on the x- and y-axes are not very descriptive – e.g., "Correlation with education population effects" – correlation between what and what? Suggest instead something like "correlation of \hat{a} for EA in main model with \hat{a} from BOLT-LMM for 9 traits" for the x-axis and "correlation of $\hat{\delta}$ for EA in main model with \hat{a} from BOLT-LMM for 9 traits" Similarly for the y-axis in Fig 6.

13) Related to the point above, in general, in both figures and text and supplement, I think authors should be re-using the terms they've already coined (e.g., $r_{\delta}(a)$) rather than switching between the term and what it stands for (e.g., $r_{\delta}(a)$ sometimes and "correlation between direct and population effects" at others). Alternatively, if they want to write out the text for what the term stands for, at least put in parentheses the term.

Thank you for the suggestions (points 12 and 13). We have changed the symbols used to represent these correlations to allow them to transmit more information. We have changed $r_{\delta}(a)$ to $r(\delta, a)$ and $r_{\delta}(\alpha)$ to $r(\delta, \alpha)$. We have added $r(\delta, a)$ to the y-axis for Figure 6, and we have referenced $r(\delta, a)$ in the legend at the relevant point. To make things clearer, we have changed the title for Figure 7 to 'Correlations between BOLT-LMM population effects from an independent sample and direct and population effects on educational attainment and cognitive ability'. To denote the correlation between direct effects on EA and BOLT-LMM summary statistics, we have used the notation $r(\delta_{EA}, a_{BOLT})$, which makes the connection to the $r(\delta, a)$ quantities estimated in the main analysis clear. We have used a similar notation for the other quantities: $r(a_{EA}, a_{BOLT})$ and $r(a_{cog}, a_{BOLT})$ for the correlation between population effects on EA and cognitive ability from the main analysis and BOLT-LMM summary statistics; and $r(\delta_{cog}, a_{BOLT})$ for the correlation between direct effects on cognitive ability and BOLT-LMM summary statistics. We have also modified the main text paragraph where we discuss these results so that we use this notation, which has shortened and clarified the text.

Decision Letter, second revision:

Our ref: NG-TR56884R1

16th Feb 2022

Dear Dr. Young,

Thank you for submitting your revised manuscript "Mendelian imputation of parental genotypes for genetic association analyses" (NG-TR56884R1). It has now been seen by the original referees and their comments are below. The reviewers find that the paper has improved in revision, and therefore we'll be happy in principle to publish it in Nature Genetics, pending minor revisions to satisfy the referees' final requests and to comply with our editorial and formatting guidelines.

Sincerely,
Wei

Wei Li, PhD
Senior Editor
Nature Genetics
New York, NY 10004, USA
www.nature.com/ng

Reviewer #1 (Remarks to the Author):

The authors have improved the clarity surrounding the issues I raised in my previous review. I have two further minor clarifications related to assortative mating below.

(1) 'bias' introduced into the NTC under assortative mating (top pg. 14 + methods)
Is the 'bias' that the authors identify the genetic variance generated by LD (i.e. genetic variance from the correlation between SNPs generated by assortative mating)? If it is then this should be clearly stated. If it is not, then where does this covariance end up in your statement 'only 1/4 of the variance explained by an educational attainment polygenic score can be attributed to direct effects alone'? I am unsure because the covariance is not important in statements about a single locus but once it moves

genome-wide (PRS) then it should be accounted for.

(2) Paragraph 3, under 'simulations using UK Biobank data' (p. 29)

The simulation of assortative mating are poorly worded. The manuscript states 'we then simulated subsequent generations by pairing parents according to their ranks in an ordered list where the ordering was determined by the value of their phenotype plus a Gaussian noise term'. When I first read this I interpreted the 'noise term' to be the environmental variance (part of the phenotype) but I assume this is not the case. Changing the variance of the environmental effects would be problematic. In addition to this clarification can the authors add clear statements about the heritability under random mating (0.5?) and that achieved under equilibrium conditions? Are these in line with theoretical expectations given the parental correlation?

Reviewer #2 (Remarks to the Author):

The authors have done an admirable job throughout in addressing reviewer concerns and improving their paper at each step. I have no further comments or suggestions.

Author Rebuttal, second revision:

Reviewer #1 (Remarks to the Author):

The authors have improved the clarity surrounding the issues I raised in my previous review. I have two further minor clarifications related to assortative mating below.

(1) 'bias' introduced into the NTC under assortative mating (top pg. 14 + methods)

Is the 'bias' that the authors identify the genetic variance generated by LD (i.e. genetic variance from the correlation between SNPs generated by assortative mating)? If it is then this should be clearly stated. If it is not, then where does this covariance end up in your statement 'only $\frac{1}{4}$ of the variance explained by an educational attainment polygenic score can be attributed to direct effects alone'? I am unsure because the covariance is not important in statements about a single locus but once it moves genome-wide (PRS) then it should be accounted for.

(2) Paragraph 3, under 'simulations using UK Biobank data' (p. 29)

The simulation of assortative mating are poorly worded. The manuscript states 'we then simulated subsequent generations by pairing parents according to their ranks in an ordered list where the ordering was determined by the value of their phenotype plus a Gaussian noise term'. When I first read this I interpreted the 'noise term' to be the environmental variance (part of the phenotype) but I assume this is not the case. Changing the variance of the environmental effects would be problematic. In addition to this clarification can the authors add clear statements about the heritability under random mating (0.5?) and that achieved under equilibrium conditions? Are these in line with theoretical expectations given

the parental correlation? (1) Due to space constraints, we have removed discussion of the assortative mating (AM) bias from the main text.

Author's Response:

In the Methods section, we state that the imputed parental PGS is not exactly the conditional expectation given the offspring PGS because of the LD induced by AM. This is not exactly the same as bias induced by an increase in the genetic variance: both the bias and the increase in genetic variance are consequences of the LD induced by AM. The increase in variance due to direct genetic effects is captured by the direct effect of the PGS. However, as we state in the Methods (and cite Kong et al. 2018 and the Supplementary Note) for a more detailed discussion, the NTCs do capture some of the direct effect of the part of the genetic component that would orthogonal to the PGS under random-mating (due to the PGS not capturing all heritability) but becomes correlated due to AM. This is also true when using observed parental genotypes, as explained in Kong et al.

(2) We have added extra detail to this section to make the procedure we used clear. We found close agreement between the observed change in genetic variance and that predicted from theory, which we now detail in that section. Note that we have now moved that section to Supplementary Note Section 12.2.

Final Decision Letter:

In reply please quote: NG-TR56884R2 Young

28th Apr 2022

Dear Dr. Young,

I am delighted to say that your manuscript "Mendelian imputation of parental genotypes improves estimates of direct genetic effects" has been accepted for publication in an upcoming issue of Nature Genetics.

Your paper will be published online after we receive your corrections and will appear in print in the next available issue. You can find out your date of online publication by contacting the Nature Press Office (press@nature.com) after sending your e-proof corrections. Now is the time to inform your Public Relations or Press Office about your paper, as they might be interested in promoting its publication. This will allow them time to prepare an accurate and satisfactory press release. Include your manuscript tracking number (NG-TR56884R2) and the name of the journal, which they will need when they contact our Press Office.

Please note that *Nature Genetics* is a Transformative Journal (TJ). Authors may publish their research with us through the traditional subscription access route or make their paper immediately open access through payment of an article-processing charge (APC). Authors will not be required to make a final decision about access to their article until it has been accepted. [Find out more about Transformative Journals](https://www.springernature.com/gp/open-research/transformative-journals)

Authors may need to take specific actions to achieve [compliance with funder and institutional open access mandates](https://www.springernature.com/gp/open-research/funding/policy-compliance-faqs). If your research is supported by a funder that requires immediate open access (e.g. according to [Plan S principles](https://www.springernature.com/gp/open-research/plan-s-compliance)) then you should select the gold OA route, and we will direct you to the compliant route where possible. For authors selecting the subscription publication route, the journal's standard licensing terms will need to be accepted, including [self-archiving-and-license-to-publish](https://www.nature.com/nature-portfolio/editorial-policies/self-archiving-and-license-to-publish). Those licensing terms will supersede any other terms that the author or any third party may assert apply to any version of the manuscript.

Please note that Nature Portfolio offers an immediate open access option only for papers that were first submitted after 1 January, 2021.

If you have not already done so, we invite you to upload the step-by-step protocols used in this manuscript to the Protocols Exchange, part of our on-line web resource, natureprotocols.com. If you complete the upload by the time you receive your manuscript proofs, we can insert links in your article that lead directly to the protocol details. Your protocol will be made freely available upon publication of your paper. By participating in natureprotocols.com, you are enabling researchers to more readily reproduce or adapt the methodology you use. [Natureprotocols.com](http://natureprotocols.com) is fully searchable, providing your protocols and paper with increased utility and visibility. Please submit your protocol to <https://protocolexchange.researchsquare.com/>. After entering your [nature.com](http://www.nature.com) username and password you will need to enter your manuscript number (NG-TR56884R2). Further information can be found at <https://www.nature.com/nature-portfolio/editorial-policies/reporting-standards#protocols>

Sincerely,
Wei

Wei Li, PhD
Senior Editor
Nature Genetics
New York, NY 10004, USA
www.nature.com/ng